



**Cloud climatologies from the InfraRed Sounders AIRS and IASI:**
**Strengths, Weaknesses and Applications**
*Claudia J. Stubenrauch[1,2], Artem G. Feofilov[1,2], Sofia E. Protopapadaki[1,2],*
*Raymond Armante[1,2]*
[1]Laboratoire de Météorologie Dynamique / Institute Pierre-Simon Laplace, (LMD/IPSL), CNRS,
Sorbonne Universities, University Pierre and Marie Curie (UPMC) Paris, University of Paris 06, Paris,
France
[2]Laboratoire de Météorologie Dynamique / Institute Pierre-Simon Laplace, (LMD/IPSL), CNRS, Ecole
Polytechnique, Université Paris-Saclay, Palaiseau, France
Correspondence to: C. J. Stubenrauch (stubenrauch@lmd.polytechnique.fr)
**Abstract**
The cloud retrieval scheme developed at the Laboratoire de Météorologie Dynamique (LMD) can now
be easily adapted to any Infrared (IR) sounder: the CIRS (Clouds from IR Sounders) retrieval applies
improved radiative transfer, as well as an original method accounting for atmospheric spectral
transmissivity changes associated with $CO_2$ concentration. The latter is essential when considering long-
term time series of cloud properties. For the 13-year and 8-year global climatologies of cloud properties
from observations of the Atmospheric IR Sounder (AIRS) and of the IR Atmospheric Interferometer
(IASI), respectively, we used the latest ancillary data (atmospheric profiles, surface emissivities and
atmospheric spectral transmissivities). The A-Train active instruments, lidar and radar of the CALIPSO
and CloudSat missions, provide a unique opportunity to evaluate the retrieved AIRS cloud properties
such as cloud amount and height as well as to explore the vertical structure of different cloud types.
CIRS cloud detection agreement with CALIPSO-CloudSat is about 84% - 85% over ocean, 79% - 82%
over land and 70% - 73% over ice / snow, depending on atmospheric ancillary data. Global cloud
amount has been estimated to 67% - 70%. CIRS cloud height coincides with the middle between the
cloud top and the ⹂apparent⹂ cloud base (real base for optically thin clouds or height at which the cloud
reaches opacity) independent of cloud emissivity, which is about 1 km below cloud top for low-level
clouds and about 1.5 km to 2.5 km below cloud top for high-level clouds, slightly increasing because the
apparent vertical cloud extent is slightly larger for large cloud emissivity. IR sounders are in particular
advantageous for the retrieval of upper tropospheric cloud properties, with a reliable cirrus identification
down to an IR optical depth of 0.1, day and night. Total cloud amount consists of about 40% high-level



clouds and about 40% low-level clouds and 20% mid-level clouds, the latter two only detected when not hidden by upper clouds. Upper tropospheric clouds are most abundant in the tropics, where high opaque clouds make out 7.5%, thick cirrus 27.5% and thin cirrus about 21.5% of all clouds. The asymmetry in upper tropospheric cloud amount between Northern and Southern hemisphere with annual mean of 5% has a pronounced seasonal cycle with a maximum of 25% in boreal summer, which can be linked to the shift of the ITCZ peak latitude. Comparing tropical geographical change patterns of high opaque clouds with that of thin cirrus as a function of changing tropical mean surface temperature indicates that their response to climate change may be quite different, with potential consequences on the atmospheric circulation.

## 1    Introduction

Clouds cover about 70% of the Earth's surface and play a key role in the energy and water cycle of our planet. The Global Energy and Water Exchanges (GEWEX) Cloud Assessment (Stubenrauch *et al.,* 2013) has highlighted the value of cloud properties derived from space observations for climate studies and model evaluation and has identified reasons for discrepancies in the retrieval of specific scenes (especially thin cirrus, alone or with underlying low-level clouds). Compared to other passive remote sensing instruments, the high spectral resolution of IR vertical sounders leads to especially reliable properties of cirrus with IR optical depth as low as 0.1, day and night. $CO_2$ sensitive channels of IR vertical sounders allow the determination of height and emissivity of a single cloud layer, which corresponds to the uppermost cloud layer in the case of multiple cloud layers. While measured radiances near the center of the $CO_2$ absorption band are only sensitive to the upper atmosphere, radiances from the wing of the band are emitted from successively lower levels in the atmosphere. Spaceborne instruments have been observing our planet since the 1980's: the High Resolution Infrared Radiation Sounders (HIRS) aboard the National Oceanic and Atmospheric Administration (NOAA) polar satellites provide data since 1979, the Atmospheric InfraRed Sounder (AIRS) aboard the National Aeronautics and Space Administration (NASA) Earth Observation Satellite Aqua since 2002, the IR Atmospheric Sounding Interferometers (IASI) aboard the European Organisation for the Exploitation of Meteorological Satellites (EUMETSAT) Meteorological Operation (MetOp) since 2006 and the Cross-track Infrared Sounder (CrIS) aboard the Suomi National Polar-orbiting Partnership (NPP) satellite since 2011, while a next generation of IR sounders (IASI-NG) is foreseen as part of the EUMETSAT Polar System – Second Generation (EPS-SG) program for 2021 (Crevoisier *et al.,* 2014).



Active sensors are part of the A-Train satellite formation (Stephens *et al.*, 2002), synchronous with
Aqua, since 2006: The CALIPSO lidar and CloudSat radar, together, determine the cloud vertical
structure (Stephens *et al.*, 2008). Whereas the lidar is highly sensitive and can detect sub-visible cirrus, its
beam only reaches the cloud base of clouds with an optical depth less than 3. For larger optical depth the
radar is providing a cloud base location.
Our goal to establish a coherent long-term cloud climatology from different IR sounders has led to the
evolution of the cloud retrieval method developed at the Laboratoire de Météorologie Dynamique
(Stubenrauch *et al.*, 1999, 2006, 2008, 2010) towards an operational and modular cloud retrieval
algorithm suite (CIRS, Feofilov and Stubenrauch, 2017) which has so far been applied to AIRS and
IASI data as well as to HIRS data (Hanschmann *et al.,* 2017). The cloud property retrieval employs
radiative transfer and atmospheric and surface ancillary data (atmospheric temperature and water vapour
profiles, surface temperature and surface emissivity, identification of snow and ice). Compared to the
initial method, the CIRS retrieval applies an improved radiative transfer and an original calibration
method to adjust the atmospheric spectral transmissivities from look-up tables, computed once for a
fixed atmospheric gaseous composition, according to latitudinal, seasonal and interannual atmospheric
$CO_2$ variations.
Compared to the 6-year AIRS-LMD cloud climatology (Stubenrauch *et al.,* 2010), which participated in
the GEWEX Cloud Assessment, results of i) an updated and extended 13-year AIRS cloud climatology
(2003 ó 2015), using two different sets of the latest ancillary data (originating from retrievals and from
meteorological reanalyses), and ii) a new 8-year IASI cloud climatology (2008 ó 2015) are presented in
this article. After the description of data and methods in section 2, section 3 is dedicated to the evaluation
of cloud detection and cloud height using the unique A-Train synergy of synchronous passive and active
measurements. Section 4 presents average cloud properties and their regional, seasonal, inter-annual and
long-term variability, in comparison with other datasets, as well as uncertainty estimates with respect to
the used ancillary data. Section 5 concentrates on the variability of the upper tropospheric clouds with
respect to changes in atmospheric conditions in order to illustrate how these data may be used for climate
studies. Conclusions and an outlook are given in section 6.
**2   Data and methods**
**2.1   AIRS Data**
The AIRS instrument (Chahine *et al.*, 2006) provides very high spectral resolution measurements of
Earth emitted radiation in 2378 spectral bands in the thermal infrared (3.74-15.40 μm). The spatial





resolution of these measurements varies from 13.5 km x 13.5 km at nadir to 41 km x 21 km at the scan
extremes. The polar orbiting Aqua satellite provides observations at 1:30 and 13:30 local time (LT).
Nine AIRS measurements (3 x 3) correspond to one footprint of the Advanced Microwave Sounder Unit
(AMSU) and are grouped as a ‒golf ball‖. The CIRS cloud retrieval uses measured radiances around the
15 μm $CO_2$ absorption band. We have chosen AIRS channels closely corresponding to the five channels
used in the TIROS-N Operational Vertical Sounder (TOVS) Path-B cloud retrieval, at wavelengths of
14.19, 14.00, 13.93, 13.28 and 10.90 μm, and three additional channels at 14.30, 14.09 and 13.24 μm
(with peaks in the weighting function at 235, 255, 375, 565, 415, 755, 885 hPa and surface, respectively).
The cloud property retrieval (section 2.4) is applied to all data, after which an ‒a posteriori‖ cloud
detection based on the spectral coherence of cloud emissivities, obtained by using the retrieved cloud
pressure, decides whether the AIRS footprint is cloudy or mostly clear (section 2.5). For the latter,
radiances in the atmospheric window between 9 and 12 μm are used, at six wavelengths of
11.85, 10.90, 10.69, 10.40, 10.16, 9.12 μm. The NASA Science Team provides L2 standard products
(Version 6 (V6); Olsen *et al.,* 2016), which include atmospheric temperature and water vapour profiles
as well as surface skin temperature. These are necessary ancillary data for the CIRS cloud retrieval. They
were retrieved from cloud-cleared AIRS radiances within each AMSU footprint. The methodology is
essentially unchanged from that described in (Susskind *et al.,* 2003). Compared to Version 5, the most
significant changes are: i) V6 uses an IR‒microwave neural network solution (Blackwell *et al.*, 2014) as
a first guess for the retrieval of atmospheric temperature and water vapour profiles as well as for surface
skin temperature, instead of the previously used regression approach (Susskind *et al.*, 2014). This leads
to physical solutions for many more cases than in Version 5 (V5). ii) The retrieval of surface skin
temperature only uses shortwave window channels (Susskind *et al.*, 2014). These modifications have
resulted in significant improvement of accurate temperature profiles and surface skin temperatures under
partial cloud cover conditions (Van T. Dang *et al.*, 2012). Compared to V5, the surface skin temperature
is larger over land in the afternoon (especially over desert) and over maritime stratocumulus regions.
We also use the microwave identification of snow or ice covered surface from the NASA L2 data.
Since the retrieved cloud pressure should be within the troposphere to lower stratosphere, we have
determined the tropopause pressure from the atmospheric profiles, using the ideology described in
(Reichler *et al.*, 2003, Feofilov and Stubenrauch, 2017) and allow clouds to be up to 30 hPa above the
tropopause.
**2.2    IASI data**



IASI, developed by CNES in collaboration with EUMETSAT, is a Fourier Transform Spectrometer
based on a Michelson interferometer covering the IR spectral domain from 3.62 to 15.5 m. As a cross-
track scanner, the swath corresponds to 30 ground fields per scan, each of these measures a $2 \times 2$ array of
footprints (12-km diameter at nadir). IASI raw measurements are interferograms that are processed to
radiometrically calibrated spectra on board the satellite. Two instruments were launched so far onboard
the European Platforms Metop-A and Metop-B (in October 2006 and September 2012, respectively),
with measurements of at 9:30 / 21:30 LT and 10:30 / 22:30 LT (local equator crossing time). IASI has
been providing water vapor and temperature sounding profiles for operational meteorology (accuracy
requirements respectively of 1 K and 10% in the troposphere), while observing simultaneously a whole
suite of trace gases, surface and atmospheric properties, including aersols and clouds (Hilton *et al.*,

11   2012).

At the time we started incorporating IASI data to the cloud retrieval, two data sets of retrieved
atmospheric profiles and surface temperature were available: one provided by EUMETSAT (Version 5)
and one by NOAA. EUMETSAT L2 temperature and water vapour Version 5 products were only
available for clear and partly cloudy scenes, leaving atmospheric and surface retrievals in only 9% of all
cases, while the recent Version 6 has extended the retrieval of thermodynamical parameters (such as
temperature and water vapor) to cloudy scenes.
Therefore we first used IASI L2 ancillary data provided by NOAA. However, a comparison with cloud
amounts deduced from AIRS via CIRS has demonstrated that the amount of low-level clouds over
ocean was underestimated (Feofilov *et al.*, 2015a), probably due to underestimated sea surface
temperature (SST) linked to cloud contamination. In addition, the comparison with collocated
temperature profiles of the Analyzed RadioSoundings Archive (ARSA, available at the French data
centre AERIS) has revealed that the AIRS-NASA and IASI-NOAA L2 atmospheric profiles were quite
different. This brought us to the conclusion, that one needs ancillary data from the same source, if one
wants to make use of the AIRS ó IASI synergy to explore the cloud diurnal cycle. Therefore, we also
implemented ancillary data from the European Centre for Medium-Range Weather Forecasts
(ECMWF) meteorological reanalyses.
**2.3    ERA-Interim meteorological reanalyses**
ECMWF provides the meteorological reanalyses ERA-Interim, covering the period from 1989 until
now. Dee *et al.* (2011) give a detailed description of the model approach and the assimilation of data.
The data assimilation scheme is sequential: at each time step, it assimilates available observations to
constrain the model built with forecast information obtained in the previous step. The analyses are then




used to make a short-range model forecast for the next assimilation time step. Gridded data products (at a
spatial resolution of 0.75° latitude x 0.75° longitude) include 6-hourly surface temperature, atmospheric
temperature and water vapor profiles, as well as dynamical parameters such as horizontal and vertical
large-scale winds. A common proxy for the intensity of the vertical motions in the atmosphere is the
vertical pressure velocity at 500 hPa level, ω500 (e.g. Bony and Dufresne, 2005; Martins *et al.,* 2011).
To match these data, given at universal time of 0:00, 6:00, 12:00 and 18:00, to the AIRS and IASI
observations, we interpolate them to the corresponding local time, using a cubic spline function, as in
(Aires *et al.*, 2004).
To avoid uncertainties in atmospheric and surface ancillary data in the analysis of the diurnal cycle of
upper tropospheric clouds from AIRS and IASI retrievals, we use ERA-Interim as ancillary data
(Feofilov *et al.*, 2015a). By using different sets of ancillary data in the cloud retrieval we are also able to
estimate uncertainties in cloud amounts (sections 3 and 4).
**2.4     CIRS cloud property retrieval**
The cloud property retrieval is based on a weighted $\chi^2$ method using channels around the 15
μm $CO_2$ absorption band (Stubenrauch *et al.*, 1999). Cloud pressure and effective emissivity
are determined by minimizing $\chi^2(p_k)$, computed at different atmospheric pressure levels by
summation over $N$ wavelengths $\lambda_i$ within the $CO_2$ absorption band and atmospheric window:

$$\chi^2(\boldsymbol{p_k}) = \sum_{i=1}^{N} \left[ (\boldsymbol{I}_{cld}(\boldsymbol{p_k}, \lambda_i) - \boldsymbol{I}_{clr}(\lambda_i)) \cdot \varepsilon_{cld}(\boldsymbol{p_k}) - (\boldsymbol{I}_m(\lambda_i) - \boldsymbol{I}_{clr}(\lambda_i)) \right]^2 * \boldsymbol{W}^2(\boldsymbol{p_k}, \lambda_i) \qquad (1)$$

where $I_m$ corresponds to the measured radiance. $I_{clr}$ is the simulated radiance the IR Sounder would
measure in the case of clear sky, and $I_{cld}(p_k)$ is the radiance emitted by a homogeneous opaque single
cloud layer, calculated for 42 levels $p_k$ above surface (from 984 hPa to 86 hPa) and for the corresponding
viewing zenith angle. In general, five (for HIRS) to eight channels (AIRS and IASI) around the 15μm
$CO_2$ band (regularly spaced) are sufficient, as a sensitivity study has shown. Doubling the number of
channels in the retrieval did not change the results.
By introducing empirical weights $W(p_k, \lambda_i)$, the method takes into account the vertical
weighting of the different channels, the growing uncertainty in the computation of $\varepsilon_{cld}$ with
increasing $p_k$ and uncertainties in atmospheric profiles. These weights are determined for each
of five typical air mass classes (tropical, midlatitude summer and winter, polar summer and
winter) as in (Stubenrauch *et al.*, 1999; Feofilov and Stubenrauch, 2017), using the spread of
clear sky radiances within these air mass classes obtained from the Thermodynamic Initial
Guess Retrieval (TIGR) data base (Chédin *et al.*, 1985; Chevallier *et al.*, 1998; Chédin *et al.*,



2003). Minimizing $\chi^2$ in Eq. 1 is equivalent to $d\chi^2/d\varepsilon_{cld} = 0$, from which one can extract $\varepsilon_{cld}$
as:

$$\varepsilon_{cld}(p_k) = \frac{\sum_{i=1}^{N}\left[I_m(\lambda_i) - I_{clr}(\lambda_i)\right] \cdot \left[I_{cld}(p_k,\lambda_i) - I_{clr}(\lambda_i)\right] \cdot W^2(p_k,\lambda_i)}{\sum_{i=1}^{N}\left[I_{cld}(p_k,\lambda_i) - I_{clr}(\lambda_i)\right]^2 \cdot W^2(p_k,\lambda_i)} \tag{2}$$

In general, the $\chi^2(p)$ profiles have a more pronounced minimum for high-level clouds than for low-level
clouds. We stress here that for the identification of low-level clouds it is important to allow values larger
than 1 for $\varepsilon_{cld}$, because at larger pressure $I_{clr}$ and $I_{cld}$ become very similar and their uncertainties may lead
to values larger than 1 (Stubenrauch *et al.*, 1999). Therefore, only pressure levels leading to $\varepsilon_{cld} > 1.5$ are
excluded from the solution. Typical $p_{cld}$ uncertainties have been estimated from a statistical analysis of
the $\chi^2(p)$ profiles: they range from 30 hPa for high-level clouds to 120 hPa for low-level clouds,
corresponding to about 1.2 km in altitude, $z_{cld}$.
In the case of atmospheric temperature inversions in the lower troposphere, for which temperature first
increases with height before decreasing, with $T(z_{inv}) > T_{surf}$, the cloud is moved to the inversion layer $z_{inv}$.
To detect these cases, the inversion strength, defined by $T(z_{inv}) - T_{surf}$, has to be larger than 2 K.
Depending on the ancillary data, these cases arise about 7 to 15 %. $\varepsilon_{cld}$ as defined in Eq. (2) does not
have a physical meaning in the case of an inversion, since $I_{cld}(p_{cld})$ will be greater than $I_{clr}$. Therefore, we
scale $\varepsilon_{cld}$ and the spectral emissivities in accordance with the ratio $p_{inv} / p_{cld}$.
Cloud temperature, $T_{cld}$, is determined from $p_{cld}$, using the ancillary temperature profile similar to the
observed situation (see section 2.4.1). Cloud types are distinguished according to $p_{cld}$ and $\varepsilon_{cld}$. High-level
clouds are defined by $p_{cld} < 440$ hPa, midlevel clouds by 440 hPa $< p_{cld} < 680$ hPa and low-level clouds
by $p_{cld} > 680$ hPa. High-level clouds may be further distinguished into opaque ($\varepsilon_{cld} > 0.95$), cirrus ($0.95 >$
$\varepsilon_{cld} > 0.50$) and thin cirrus ($\varepsilon_{cld} < 0.50$). $p_{cld}$ is transformed to cloud altitude, $z_{cld}$, using a hydrostatic
conversion, with the virtual temperature profile accounting for humidity, again from ancillary data
similar to the observed situation.
The retrieval is applied to all footprints. In a second step, an ʻa posterioriʼ cloud detection is applied
(section 2.5). When sufficient channels are available in the atmospheric window, as for the high spectral
resolution IR sounders like AIRS, CrIS and IASI, a test based on the spectral coherence of retrieved
cloud emissivities decides whether the footprint is cloudy (overcast or mostly cloudy) or clear (or not
cloudy enough to determine reliable cloud properties). Thresholds have been established using the A-





Train synergy (section 3). In the case of HIRS, other methods have been developed to decide if the scene
is cloudy (e. g. Stubenrauch *et al.*, 2006; Hanschmann *et al.*, 2017).
For the computation of $I_{clr}$ and $I_{cld}$ in Eq. (1), we need surface skin temperature and spectral surface
emissivities as well as atmospheric temperature and spectral transmissivity profiles for the atmospheric
situation of the measurements. The atmospheric spectral transmissivity profiles were calculated using the
4A radiative transfer model (Scott and Chédin, 1981), separately for each satellite viewing zenith angle
(up to 50°) and for about 2300 representative clear sky atmospheric temperature and humidity profiles of
the TIGR data base.
In the cloud retrieval, the TIGR data base is searched for the atmospheric profile corresponding best to
the observational conditions by applying a proximity recognition which compares the atmospheric
temperature and water vapour profiles from the ancillary data with those from TIGR as in (Stubenrauch
*et al.*, 2008). The preparation and evaluation of these ancillary data is presented in 2.4.1.
*2.4.1 Preparation and comparison of atmospheric / surface ancillary data*
Over land we use monthly spectral surface emissivity climatological values at a spatial resolution of
0.25° x 0.25°, retrieved from IASI measurements (Paul *et al.*, 2012) and spectrally interpolated to the
AIRS channels. Over ocean the surface emissivity is set to 0.99 for wavelengths less than and 0.98 for
wavelengths larger than 10 μm (Wu and Smith, 1997). Over snow and ice, the spectral surface
emissivities are taken from (Hori *et al.*, 2006), and since these depend in this case on the viewing zenith
angle, they are corrected as in (Smith *et al.*, 1996).
Since IR sounders, in combination with microwave sounders, were originally designed for the retrieval
of atmospheric temperature and humidity profiles, the atmospheric clear sky situation can then be
directly described by simultaneous L2 atmospheric profiles of good quality (when the situation is not too
cloudy). When these are not available, we use atmospheric profiles, surface skin temperature and
tropopause of good quality, averaged over 1° latitude x 1° longitude, and if there are still no data
available we interpolate these averages in time (inversely proportional to distance within maximal ±15
days) and then in space (inversely proportional to distance within maximal 3° longitude, considering the
same surface type).
To define atmospheric temperature and humidity profiles as well as surface temperature of good quality,
one has to find a compromise between an acceptable quality and enough statistics.
This led to the following quality criteria in the case of ancillary data from AIRS-NASA (V6) :





• Surface temperature is of good quality, if the provided retrieval error is smaller than 3 K / 6 K / 7 K for
ocean / land / ice or snow, respectively. It should also be larger than 180 K and smaller than 400 K.
• Atmospheric temperature profiles are of bad quality, when three consecutive layers have large retrieval
errors, with thresholds in the upper part (70 hPa to 500 hPa) / lower part of the troposphere (500 hPa to
surface) / near surface of 2 K / 2K / 2K over ocean, 2.5 K / 2.5 K / 3 K over land and 2.5 K / 2.5 K / 5
K over ice or snow, respectively.
• For atmospheric water vapour profiles the quality criteria of NASA were kept.
Nevertheless, when comparing SST of good quality from AIRS with SST from ERA-Interim, AIRS
values were slightly colder. Since this effect is most probably due to a slight underestimation of the
AIRS SST linked to cloud contamination, we applied a small correction to SST by adding the minimum
between 0.5 K and the retrieval error. Since the behaviour over land is more complex, we left the surface
temperature ($T_{surf}$) values as they are.
When we use time interpolated ERA-Interim atmospheric profiles and surface temperature as ancillary
data in the cloud retrieval, these data are always available. However, since the time interpolated ERA-
Interim SST did not show a diurnal cycle (most of the amplitudes are less than 0.2 K), which is not
consistent with observations (e.q. Webster *et al.*, 1996), we applied a simple parameterized correction,
based on Fig. 11 of (Webster *et al.*, 1996). This parameterization links the SST diurnal cycle to peak
insolation. The coefficient between the maximal solar flux at given latitude, longitude, solar zenith angle
and local time and the SST diurnal amplitude was adjusted to 0.005 K/Wm$^{-2}$ to make the latter consistent
with that of recent observations (e.g. Seo *et al.*, 2014). Without this correction, cloud amount over ocean
was larger during night (78%) than in the early afternoon (71%), while now cloud amount is more
similar (76% / 73%), in better agreement with results using AIRS ancillary data (71% / 71%). The
behaviour over land is more complex, so we left the $T_{surf}$ values as they are, leading to CA of 62% / 56%,
with ERA-Interim, and 56% / 58%, with AIRS-NASA, at 1:30AM / 1:30PM.
Figure 1 presents comparisons of $T_{surf}$, as used in the cloud retrieval, deduced from NASA AIRS
retrievals and from ERA-Interim, with collocated surface air temperature, $T_{surf}^{air}$, from the ARSA data
base. One would expect that over land $T_{surf}$ is colder than $T_{surf}^{air}$ during night and warmer than $T_{surf}^{air}$ in
the afternoon; this effect should be stronger for temperate and warmer temperatures, especially if the
climate is dry. SST should be similar to $T_{surf}^{air}$ in the tropics, slightly warmer in midlatitudes and colder
in polar regions. Considering Figure 1, the distributions reflect the expectations, with peaks for AIRS-
NASA and ERA-Interim corresponding to similar differences with ARSA. When looking more in





detail, the land distributions are slightly larger for AIRS-NASA than for ERA-Interim, and they are
shifted towards colder values for colder $T_{surf}$ and at night for warmer $T_{surf}$. For warmer $T_{surf}$ in the
afternoon, AIRS-NASA $T_{surf}$ is slightly larger than ERA-Interim $T_{surf}$. Colder AIRS-NASA values
might still indicate some cloud contamination, whereas the colder values of ERA-Interim over warm
land in the afternoon might indicate an underestimation, especially over desert, as has already been
pointed out by Trigo *et al.* (2015). The effect of $T_{surf}$ on cloud amount will be further investigated in
section 3.2.
*2.4.2 Calibration for changes in atmospheric $CO_2$ concentration*
The TIGR data base of atmospheric spectral transmissivities was created for an atmosphere with a fixed
$CO_2$ volume mixing ratio of 372 ppmv. However, the atmospheric $CO_2$ concentration varies
latitudinally, seasonally and with time. While both the increase during the last ten years and the seasonal
variability in the Northern hemisphere (NH) are of the order of ~20 ppmv, the latitudinal gradient in the
NH varies from  0.1 ppmv / ° to +0.1 ppmv / °. Seasonal variability in the NH is related to the
vegetation and fossil fuel burning seasonality. The difference between an averaged value and actual $CO_2$
volume mixing ratio can easily reach 10%, which is a noticeable change since the concentration enters
the power of the exponent in calculating the transmissivity, $\tau$. To avoid errors in the radiative transfer
associated with $CO_2$ changes, we rescale the transmissivity according to the following rule:

18                  $\tau = \exp(-\beta\ \acute{o}\ \alpha\ CO_2^{current})$                  (3)

with $\alpha = -k \log (\tau^{ref})/ CO_2^{ref}$ and $\beta = \alpha\ CO_2^{ref} (1-k)/k$, where k is the relative $CO_2$ contribution to the
opacity of the channel. Details are described in (Feofilov and Stubenrauch, 2017). The $CO_2$
concentrations are taken from (GLOBALVIEW-CO2, 2013).
This correction also removes long-term biases due to increasing $CO_2$ in the atmosphere from
anthropogenic $CO_2$ emissions, which introduced an artefact in the time series of cloud amount. Applying
the correction of equation (3) has eliminated this bias (see section 4).
*2.4.3 Summary of changes compared to the previous version of the AIRS-LMD cloud climatology*
Compared to the six-year AIRS-LMD cloud climatology (Stubenrauch *et al.*, 2010), the following
changes have been implemented into the CIRS algorithm:
• Minimum cloud pressure has been extended from 106 hPa to 86 hPa.
• Ancillary atmospheric and surface data have been updated from NASA V5 to NASA V6.



• To fill gaps in atmospheric and surface ancillary data of good quality, the interpolation method has
slightly changed.
• In the case of atmospheric temperature inversions, the cloud is moved to the inversion layer and $\varepsilon_{cld}$ is
scaled accordingly.
• The radiative transfer to determine the TIGR atmospheric spectral transmissivities has been improved.
• The atmospheric spectral transmissivity of the TIGR data base near the surface was adjusted to the
surface pressure of the observed situation.
• The improved radiative transfer computations of clear sky radiances led to a decreased threshold on
the variability of the cloud spectral emissivities between 9 and 12 μm, used in the cloud detection, (see
section 2.5).
• Only one cloud detection test, based on the coherence of cloud spectral emissivity, is applied.
• Simulated clear sky atmospheric spectral transmissivities have been corrected for variability in
atmospheric $CO_2$ concentration.
The impact of these changes, however, is in general small, as can be seen in the latitudinal averages of
total, high, midlevel and low-level cloud amounts presented in section 4.
**2.5 A posteriori cloud detection**
Once the cloud properties are retrieved, we use the same cloud detection strategy as in (Stubenrauch *et*
*al.*, 2010), based on the spectral coherence of retrieved cloud emissivities between 9 and 12 μm,
wavelengths in the IR atmospheric window. For each footprint, cloud emissivities $\varepsilon_{cld}$ are
determined at six wavelengths, $\lambda_i$, as:

$$\varepsilon_{cld}(\lambda_i) = \frac{I_m(\lambda_i) - I_{clr}(\lambda_i)}{I_{cld}(p_{cld}, \lambda_i) - I_{clr}(\lambda_i)} \tag{4}$$

where $I_{cld}$ is now determined for $p_{cld}$ which has been retrieved by the $\chi^2$ method (see above). When $p_{cld}$
is well determined, these spectral cloud emissivities should only slightly differ. The variability should be
larger, when the footprint is partly cloudy or clear and hence $p_{cld}$ is not well determined. In that case, the
footprint is declared as not cloudy.
To determine thresholds, we make use of the A-Train synergy: by comparing distributions of the
standard deviation $\sigma(\varepsilon)$ over these wavelengths divided by the retrieved $\varepsilon_{cld}$, separately for cloudy scenes
and for clear sky scenes as determined by CALIPSO (see section 3.1). Overcast / clear sky scenes are





situations for which all three CALIPSO samples within the AIRS golf ball are cloudy / clear,
respectively, and partly cloudy scenes include a mix of cloudy and clear sky within the three samples.
Figure 2 presents these distributions, separately over ocean, land and ice / snow, when AIRS ancillary
data and when ERA-Interim ancillary data are used in the AIRS cloud retrieval. First of all, we observe
that the distributions are in general narrower for cloudy scenes than for clear sky, as expected. The large
tails of the clear sky distributions are presented as a large peak at $(\varepsilon)/\varepsilon_{cld} = 0.59$, the maximum value to
which $(\varepsilon)/\varepsilon_{cld}$ was set. The separation between cloudy and clear is best over ocean, followed by land
and then ice / snow. Distributions are similar over ocean and land between both ancillary data, whereas
the distinction between cloudy and clear sky over ice / snow is slightly better when ERA-Interim is used.
This might be explainable by the fact that the retrieval of atmospheric profiles with good quality is
challenging over ice / snow. According to these figures and by experimenting with thresholds to obtain a
good agreement in the identification of cloudy and clear sky scenes with CALIPSO-CloudSat (see
section 3.2), we perform the following tests for the AIRS-CIRS cloud detection.
The footprint is identified as cloudy if the following conditions are fulfilled:
$(\varepsilon)/\varepsilon_{cld} < 0.17 / 0.20 / 0.30$         for ocean / land / snow or ice and AIRS ancillary data
$(\varepsilon)/\varepsilon_{cld} < 0.17 / 0.20 / 0.20$         for ocean / land / snow or ice and ERA-Interim ancillary data
For IASI we do not have the possibility to test these distributions with CALIPSO-CloudSat. However,
the overall distributions of $(\varepsilon)/\varepsilon_{cld}$ are similar for AIRS and IASI, comparing retrievals both based on
ERA-Interim ancillary data. Therefore we use the same thresholds for the IASI cloud detection.
To reduce noise, we declare footprints with a cloud of $\varepsilon_{cld} < 0.10$, corresponding to a visible (VIS) optical
depth of about 0.2, as not cloudy.

## 22    3      Evaluation of cloud properties using the A-Train synergy

The A-Train active instruments, lidar and radar of the CALIPSO and CloudSat missions, provide a
unique opportunity to evaluate the retrieved AIRS cloud properties such as cloud amount and cloud
height, as well as to explore the vertical structure of the AIRS cloud types (Stubenrauch *et al.*, 2010).
These results can then be transposed to cloud types determined by CIRS retrieval method using other IR
sounders.

### 28    3.1     Collocated AIRS Ë CALIPSO Ë CloudSat data

We use the same colocation procedure as in (Feofilov *et al.*, 2015b): all satellites of the A-Train follow
each other within a few minutes. First, each AIRS footprint is collocated with NASA CALIPSO L2



cloud data averaged over 5 km (version 3, Winker *et al.*, 2009) in such a way that for each AIRS golf
ball, three CALIPSO samples closest to the centres of each AIRS footprint are kept. These data are then
collocated with the vertical profiling of the NASA L2 Lidar CloudSat geometrical profiling
(GEOPROF) data (version P1_R04; Mace and Zhang, 2014). Each AIRS footprint includes thus
information on the vertical structure (cloud top and cloud base for each of the cloud layers) at the spatial
resolution of the radar footprints (1.4 km x 2.3 km) and in addition to cloud detection, cloud optical
depth, cloud top and apparent cloud base (corresponding to the real cloud base or to the height at which
the cloud reaches opacity) at the spatial resolution of the CALIPSO cloud data (5 km x 0.09 km). A
cloud feature flag indicates whether the cloud is opaque. The CALIPSO L2 cloud data also indicate at
which horizontal averaging the cloud was detected (1 km, 5 km or 20 km), which is a measure of the
optical thickness of the cloud. For a direct comparison with AIRS cloud data, we use clouds detected at
horizontal averaging over 5 km or less, corresponding to minimum particle backscatter coefficient of
about 0.0008 km$^{-1}$sr$^{-1}$ at night and about 0.0015 km$^{-1}$sr$^{-1}$ during day, for a cirrus with an altitude of
about 12 km (Fig. 4 of Winker *et al.*, 2009). This corresponds to clouds with VIS optical depth larger
than about 0.05 to 0.1 (Winker *et al.*, 2008). The scene over each AIRS footprint is estimated by using
the cloud detection of all three CALIPSO samples per AIRS golf ball as: clear sky, partly cloudy and
overcast.
For the evaluation of cloud height we determine the lidar CloudSat GEOPROF cloud layer which is
closest to $z_{cld}$ from AIRS. From the 5 km averaged CALIPSO data we also determine the height at which
the cloud reaches a certain optical depth, in particular 0.5, $z_{COD0.5}$. We then require that this height is
located within the corresponding cloud layer of the lidar CloudSat GEOPROF data.
Cloud optical depth (COD) determined from lidar backscatter depends on a correction for multiple
scattering which itself depends on COD and microphysics (e. g. Comstock and Sassen, 2001; Chen *et*
*al.*, 2002; Lamquin *et al.*, 2008). As CALIPSO assumes a constant multiple scattering coefficient of 0.6
in the retrieval (Winker, 2003), COD might be slightly underestimated, especially for larger COD. We
therefore estimate from Figure 3 in (Lamquin *et al.*, 2008) a correction factor and deduce that a COD of
0.50 should correspond to a COD given by CALIPSO of about 0.37. To determine the height within the
cloud at which COD reaches 0.5 we also use an assumption on the shape of the ice water content vertical
profile between cloud top and cloud base (Feofilov *et al.*, 2015b).
In the following, we analyze three years (2007-2009) of collocated AIRS-CALIPSO-CloudSat data,
separately for three latitude bands: tropical / subtropical latitudes (30°N-30°S), midlatitudes (30°N-60°N
and 30°S-60°S) and polar latitudes (60°N-90°N and 60°S-90°S).





### 3.2    Cloud detection

The a posteriori cloud detection leads to an agreement with the CALIPSO-CloudSat cloud detection in
about 85% (84%) over ocean, 82% (79%) over land and 70% (73%) over ice / snow, using atmospheric
and surface ancillary data, deduced from AIRS-NASA (ERA-Interim). Table 1 presents these
agreements separately for the three latitude bands. In general, these agreements are quite high,
considering that CALIPSO and GEOPROF data only sample a small area of the AIRS footprint. They
are slightly higher over ocean than over land. Compared to the AIRS-LMD cloud retrieval presented in
(Stubenrauch *et al.*, 2010), the agreement with CALIPSO-CloudSat has improved both over ocean and
land, but slightly decreased over sea ice. The latter can be explained by applying now only one test over
all surface types. In the earlier version we used an additional brightness temperature difference test
related to temperature inversions. A detailed analysis (not shown) indicated that it also introduced noise.
To further illustrate cloud amount (CA) uncertainties due to ancillary data, geographical maps of CA
differences between AIRS-CIRS based on ancillary data from AIRS-NASA and from ERA-Interim,
together with $T_{surf}$ differences, are shown in Figure 3. When using AIRS-NASA ancillary data, CA is
mostly smaller over land during night and larger over land in the afternoon. One might observe a positive
correlation with differences in $T_{surf}$: $T_{surf}$ of the ancillary data deduced from AIRS-NASA is slightly
smaller during night and larger during daytime over large parts of the continents. From the $T_{surf}$
comparison with ARSA in section 2.4, we deduced that over land AIRS-CIRS CA is slightly
underestimated during night when using AIRS-NASA ancillary data, while slightly underestimated in
the afternoon when using ERA-Interim ancillary data. Patterns of differences in atmospheric water
vapour are less reflected in those of CA (not shown), but slightly more atmospheric water vapour in the
ancillary data (as in the tropics for AIRS-NASA compared to ARSA and ERA-Interim) might lead to a
slight underestimation of CA.

### 3.3    Cloud height

Figure 4 presents normalized distributions of the difference between the height at which COD reaches a
value of about 0.5, $z_{COD0.5}$, determined from CALIPSO, and the retrieved cloud height from AIRS, $z_{cld}$,
as well as normalized distributions of the difference between the cloud top height from CALIPSO, $z_{top}$,
and $z_{cld}$. We compare results of the CIRS cloud retrieval, using ancillary data from AIRS-NASA and
ERA-Interim, separately for high-level clouds ($p_{cld} < 440$ hPa) and lower-lever clouds ($p_{cld} \times 440$ hPa).





The AIRS cloud height is compared to the CALIPSO-CloudSat cloud layer, which is the closest to $z_{cld}$.
This is justified, because CALIPSO and CloudSat sample only sparsely the AIRS footprint, and AIRS
could observe a mixture of several clouds. In general, all distributions of differences between $z_{COD0.5}$ and
$z_{cld}$ peak around 0 km and are slightly narrower for low-level clouds than for high-level clouds. Results
are similar for both ancillary data, with a slight cloud height overestimation for lower level clouds in the
tropics over ocean (not shown), when using ERA-Interim, and a height overestimation of some clouds in
polar regions over ocean (not shown), when using AIRS-NASA ancillary data. The latter might be
explained by the fact that in some of these regions surface temperature and atmospheric profiles of good
quality are only available in 10% of the situations. When comparing distributions of $z_{top}$ - $z_{cld}$, the peaks
for lower clouds are still around 0 km, whereas for high-level clouds $z_{cld}$ lies on average 1.5 km below
the cloud top, meaning that $T_{cld}$ is about 10 K warmer than the cloud top (Figure S1). The CIRS retrieved
cloud height coincides with the height of maximum lidar backscatter (Stubenrauch *et al.*, 2010), with
mid-height between cloud top and "apparent" cloud base (real cloud base for optically thin clouds or
cloud height at which the cloud reaches opacity), as shown in Figure S1, or with $z_{COD0.5}$, as shown in
Figure 4.
To investigate more in detail how the CIRS retrieved cloud height relates to the height of COD of about
0.5 and to cloud top ($z_{top}$), we analyze in Figure 5 their average difference as a function of AIRS cloud
emissivity, separately for high-level and lower level clouds. For this analysis we have selected cases for
which AIRS cloud height lies within the cloud borders from CALIPSO-CloudSat GEOPROF, leaving
about 82% / 73% / 57% and about 55% / 59% / 58% of the statistics of high-level and lower level clouds
over the tropics / midlatitudes / polar regions, respectively. In general, for low-level clouds, the AIRS
cloud height lies about 250 m – 500 m below the height at which the cloud reaches an optical depth of
about 0.5, independently of $\varepsilon_{cld}$, while $z_{cld}$ lies about 1 km below the cloud top. For high-level clouds the
$z_{cld}$ varies from 1 km above for $\varepsilon_{cld} = 0.1$ to 1 km below the height corresponding to COD of 0.5 for $\varepsilon_{cld}$
= 1, assuming that COD is accurately determined for all $\varepsilon_{cld}$. This means that for thin cirrus $z_{cld}$ from
AIRS corresponds to a height of COD < 0.5, while for opaque high clouds to a height of COD > 0.5. On
the other hand, $z_{cld}$ lies about 1.5 km to 2.5 km below the cloud top, the difference to cloud top increasing
with $\varepsilon_{cld}$. Since the apparent vertical extent also increases with $\varepsilon_{cld}$, the difference between $z_{top}$ and $z_{cld}$
scaled by apparent vertical extent does not depend on $\varepsilon_{cld}$, and it is about 0.5 for high-level and for low-
level clouds. Considering the normalized frequency distributions of $z_{top}$ – $z_{COD0.5}$ and $z_{top}$ - $z_{cld}$, as well as
these differences scaled by apparent cloud vertical extent, presented in Figure 6, we deduce that it needs
less geometrical thickness for opaque clouds than for semi-transparent clouds to reach COD of 0.5,



while the $\chi^2$ method determines a height within the cloud, which corresponds well to the middle between
cloud top and apparent cloud base, in dependent of $\varepsilon_{cld}$. This is important to take into account for the
determination of radiative fluxes and heating rates of upper tropospheric clouds, when using the cloud
height retrieved from IR sounder measurements. The broader distributions for high-level clouds
compared to low-level clouds in Figures 4 and 6 may be explained by the fact that high-level clouds
often have diffuse cloud tops (e. g. Liao *et al.*, 1995), especially in the tropics ($z_{top}$ - $z_{cld}$ is slightly larger
for the same $\varepsilon_{cld}$).
In order to see how well the distribution of clouds is represented within the atmosphere, we compare in
Figure 7 the normalized distributions of $z_{cld}$ from AIRS, using both sets of ancillary data, and of $z_{COD0.5}$
from CALIPSO, whenever clouds are detected (excluding subvisible cirrus, see section 3.1), separately
over land and over ocean in the three latitude bands. AIRS $z_{cld}$ distributions are very similar, with slightly
more low-level clouds over land using ERA-Interim and slightly more higher clouds over polar ocean
(which are mostly misidentifications as pointed out earlier). The $z_{COD0.5}$ distributions from CALIPSO
have a slightly larger part of high-level clouds in the tropics and AIRS $z_{cld}$ distributions show a slightly
larger part of low-level clouds in the tropics. The latter disappear if one considers only cases with all
three CALIPSO samples cloudy within an AIRS golf ball, so these low-level clouds are part of partly
cloudy fields for which it is difficult to compare results from samples of very different spatial resolution.
Thus the distributions look more similar when only mostly covered cloud fields are considered (three
CALIPSO samples cloudy within an AIRS golf ball). In the tropics, the peak of the AIRS $z_{cld}$
distributions for high-level clouds is still slightly broader towards lower heights than for CALIPSO (not
shown). Additional filtering out of multi-layer clouds ultimately leads to very similar distributions, as
also presented in Figure 7. A plausible interpretation is that in cases of multiple cloud layers the 15 km
footprints of AIRS often mix radiation from different cloud layers, when the upper cloud layer does not
fully cover the footprint, and thus determines a cloud height which might be slightly lower than the one
of the uppermost cloud layer. The distributions in the midlatitudes still peak at slightly lower heights, due
to the fact that high-level clouds in these latitudes are on average optically thicker (storm tracks) than in
the tropics, and as we have seen in Figure 5, in these cases $z_{cld}$ lies below $z_{COD0.5}$.
To summarize, the evaluation of cloud height has shown that IR sounders capture quite well the vertical
distribution of uppermost clouds in the atmosphere. The retrieval provides a cloud height of about 1 km
below cloud top in the case of low-level clouds and of about 1.5 km to 2.5 km below cloud top height in
the case of high-level clouds. In the latter case, the retrieved cloud height corresponds to a height of COD
< 0.5 for optically thin clouds and to a height of COD > 0.5 for optically thick clouds. On the other hand,





multiple scattering within optically thicker clouds is in general larger so that the correction we have
applied above, which was meant for clouds with a total COD of 0.5, was probably not enough. As
already shown by Stubenrauch *et al.* (2010), the CIRS retrieved cloud height coincides with the middle
between cloud top and apparent cloud base, and this for all cloud heights. Even though the spatial
resolution of 15 km may mix clear sky and cumulus clouds, or thin cirrus with optical thicker high
clouds, the cloud height is in general well determined within 1.5 km.
**4    Average Cloud properties and variability**
In this section we give a short overview of cloud properties obtained from the AIRS-CIRS and IASI-
CIRS cloud climatologies. Monthly L3 data, gridded at a spatial resolution of 1° latitude x 1° longitude,
have been produced in the same manner as for the GEWEX Cloud Assessment data base (Stubenrauch
*et al.*, 2013): in a first step, averages were determined per observation time over 1° latitude x 1°
longitude, and in a second step these cloud properties were averaged per month. In addition to monthly
averages, the data base also includes time variability and histograms. We have also added uncertainties
on $p_{cld}$ and $\varepsilon_{cld}$ deduced from the $\chi^2$ method.
Figure 8 compares normalized frequency distributions of $p_{cld}$ (CP) over 30° wide latitude bands during
boreal winter and boreal summer, separately over land and over ocean. The AIRS and IASI CP
distributions are very similar. The contribution of high-level clouds is slightly larger over land than over
ocean, especially in the tropics, and the contribution of low-level clouds is larger over ocean.
Considering seasonality, the strongest signature is the shift of the Intertropical Convergence Zone
(ITCZ) towards the summer hemisphere, especially over land.
Figure 9 presents global averages of total cloud amount (CA) and relative contributions of high-level,
mid-level and low-level clouds, determined by dividing these cloud amounts (CAH, CAM, CAL) by
CA. The sum of the relative contributions, CAHR, CAMR and CALR is equal to 1. Pressure limits for
high-level/mid-level and mid-level/low-level cloud classification are 440 hPa and 680 hPa,
corresponding to altitudes of about 6 km and 3 km, respectively. Relative cloud amount values give an
indication of how the detected clouds are vertically distributed in the atmosphere, when observed from
above. Compared to the absolute values, they are less influenced by differences in cloud detection
sensitivity and should be more useful for comparison with climate models (Stubenrauch *et al.*, 2013).
Global averages of AIRS-CIRS and IASI-CIRS are compared with those from selected cloud
climatologies of the GEWEX Cloud Assessment data base: the International Satellite Cloud
Climatology Project (ISCCP; Rossow and Schiffer, 1999), two cloud climatologies derived from
observations of the Moderate Resolution Imaging Spectroradiometer (MODIS) aboard the Aqua satellite





by the MODIS Science Team (MODIS-ST; Frey *et al.,* 2008) and by the MODIS CERES Science
Team (MODIS-CE; Minnis *et al.,* 2011) and two cloud climatologies derived from CALIPSO
observations by the CALIPSO Science Team (CALIPSO-ST; Winker *et al.,* 2009) and the GCM-
Oriented CALIPSO Cloud Products (CALIPSO-GOCCP; Chepfer *et al.*, 2010). The latter two use
vertical averaging (CALIPSO-GOCCP) and horizontal averaging (CALIPSO-ST) to reduce the noise of
the relatively small samples. The latter is more sensitive to thin layers of subvisible cirrus. ISCCP is
essentially using two atmospheric window channels (IR and VIS, the latter only during daytime). For the
GEWEX Cloud Assessment data base the eight-times-daily ISCCP results have been averaged to four
specific local observation times: 3:00 AM, 9:00 AM, 3:00 PM and 9:00 PM, and a day-night adjustment
on CA, which is included in the original data, has not been included to better illustrate the differences
between VIS-IR and IR-only results. We separately examine observations mostly during day,
corresponding to 1:30PM (3:00PM for ISCCP, 9:30AM for IASI), and mostly during night,
corresponding to 1:30AM (3:00AM for ISCCP and 9:30PM for IASI). Total cloud amount from the
GEWEX Cloud Assessment data base is about 0.68±0.03 (Stubenrauch *et al.*, 2013), while CALIPSO-
ST provides a cloud amount of 0.73, because it includes subvisible cirrus.
While all data agree quite well on the total cloud amount, with ISCCP and MODIS-CE providing
smaller CA during night (both including VIS information for cloud detection during daytime), CAHR
exhibits a large spread, essentially due to different sensitivity to thin cirrus : active lidar is the most
sensitive, followed by IR sounders, as confirmed in Figure 9. The CIRS results are very similar to the
results from the AIRS-LMD cloud climatology (Stubenrauch *et al.*, 2010). AIRS-CIRS results based on
different ancillary data are also very similar as well as IASI-CIRS and AIRS-CIRS results, day and
night. They present global averages of CA around 0.67 ó 0.70, formed by 40% high-level clouds, 20%
midlevel clouds and 40% low-level clouds as seen from above. This is in excellent agreement with the
results from CALIPSO. A slightly higher value in CAMR (20% instead of 14%) can be explained by the
fact that the distinction between high-level and mid-level clouds of CALIPSO is according to cloud top
height, whereas AIRS and IASI provide a cloud height which is about 1.5 km lower (see section 3.3).
When combining VIS and IR information, thin cirrus above low-level clouds tend to be misidentified as
mid-level clouds (ISCCP) or as low-level clouds (MODIS), leading to a not negligible underestimation
of CAHR (30% instead of 40%). During nighttime, for which only one IR channel is available, ISCCP
underestimates the height of all semi-transparent high-level clouds, so that CAHR drops to 15%. When
IR spectral information is available, as for MODIS, results are similar to those during daytime.





Differences between ocean and land, also presented in Figure 9, correspond to about 15% for total CA,
with about 20% more low-level clouds over ocean and about 10% more high-level and mid-level clouds
over land. The CIRS retrievals provide similar values during day and night. It is interesting to note that
during daytime the difference in CA shows a larger spread between the datasets, while during nighttime
the spread is larger for CALR. During nighttime, low-level clouds are more difficult to detect, especially
over land.
Table 2 summarizes averages of these cloud amounts over the whole globe, over ocean and over land,
also contrasting NH and Southern hemisphere (SH) midlatitudes (30°-60°) and tropics (15°N-15°S). The
largest fraction of high-level clouds is situated in the tropics and the largest fraction of single layer low-
level clouds in the SH midlatitudes. Only about 10% of all clouds in the tropics are single layer midlevel
clouds, compared to about 22% in the midlatitudes. As already discussed in sections 2.4 and 3.2,
uncertainty in CA as well as in CALR due to ancillary data is largest over land (about 5% and 10%,
respectively), linked to underestimation of low-level clouds during night with AIRS-NASA, and in the
afternoon with ERA-Interim. Uncertainties due to ancillary data are much smaller for high-level clouds.
When separating them into three distinct classes of opaque, thick cirrus and thin cirrus according to $\varepsilon_{cld}$
(see section 2.4), uncertainties are less than 5% at low latitudes, increasing to 10% at midlatitudes for
opaque clouds, while those for cirrus do not exceed 5%. This can be explained by the fact that in the case
of opaque clouds the ancillary data often have to be interpolated in time, and atmospheric profiles and
$T_{surf}$ have a larger variability in the midlatitudes than in the tropics. While high-level opaque clouds only
make out about 5.2% of all clouds, relative cloud amounts of thick cirrus and thin cirrus are about 21.5%
and 13%, respectively, with maximum appearance in the tropics, of 7.5%, 27.5% and 21.5%,
respectively. Their relative amounts are summarized in Table 3. The independent use of $p_{cld}$ and $\varepsilon_{cld}$
made it possible to construct a climatology of upper tropospheric cloud systems by i) applying a spatial
composite technique on adjacent $p_{cld}$ and ii) using $\varepsilon_{cld}$ to distinguish convective core, cirrus anvil and thin
cirrus of these systems. These data have revealed for the first time that the $\varepsilon_{cld}$ structure within tropical
anvils is related to the convective depth (Protopapadaki $et$ $al.$, 2017), which might have important
consequences on radiative feedbacks.
Figure 10 presents zonal averages of CA, CAH and CAL as well as effective cloud amount for total
(CAE) high-level (CAEH) and low-level (CAEL) clouds. The annual zonal averages are presented from
the three CIRS and the prior LMD cloud climatology. In addition, boreal winter and boreal summer
zonal averages are shown for AIRS-CIRS alone, but separately for each of the thirteen years to illustrate
the inter-annual spread. Effective cloud amount corresponds to the cloud amount weighted by cloud





emissivity and includes the IR radiative effect of the detected clouds. In general, CAE is about 0.2
smaller than CA. Maximum CAH and CAEH appear in the ITCZ, while maximum CAL and CAEL is
found in the SH midlatitudes. Interannual variability is largest in CA and CAL (CAE and CAEL) in the
NH polar region. One also observes that the midlatitude interannual variability of CAH is larger in
winter than in summer, most probably linked to storm track variability. When comparing the different
CIRS retrievals, all agree in general very well, with AIRS-CIRS and IASI-CIRS with ERA-Interim
being very close, while AIRS-CIRS with AIRS-NASA presents slightly more high-level clouds and less
low-level clouds around 60S and slightly less CA and CAL in the NH polar region.
Figures 11 and 12 present geographical maps of annual CAH and CAL, respectively, as well as seasonal
differences. Compared are AIRS-CIRS, ISCCP and CALIPSO-GOCCP, the latter two from the
GEWEX Cloud Assessment data base. In all datasets the most prominent feature in CAH is the ITCZ
and its shift towards the summer hemisphere. However, due to the better sensitivity to cirrus, the absolute
values and seasonal variations are more pronounced for AIRS-CIRS (IASI-CIRS, not shown) and
CALIPSO-GOCCP than for ISCCP. Due to the narrow nadir track of CALIPSO and the reduced
statistics of CALIPSO-GOCCP in the present GEWEX Cloud Assessment data base, these data look
noisier than AIRS-CIRS and ISCCP. In addition, jet streams and midlatitude storm tracks in winter, as
well as continental cirrus in summer can be distinguished. Considering CAL, AIRS-CIRS well captures
the stratocumulus regions off the West coasts of the continents and stratus decks in the subtropical
subsidence regions in winter, even if this type of cloud is easier to detect by using instruments including
VIS channels (during daytime, ISCCP) or active instruments (CALIPSO-GOCCP).
Time series of deseasonalized anomalies in global monthly mean CA, CAEH and CAEL of the three
CIRS data sets are shown in Figure 13 over the time period of 2004 ó 2016 for AIRS and 2008 ó 2016
for IASI. To illustrate the effect of the calibration for changes in atmospheric $CO_2$ concentration (section
2.4.2), a time series of AIRS-CIRS deseasonalized CA anomalies, without having applied this
correction, is added. Whereas the uncorrected CA anomalies increase by about 0.040 within a decade,
the magnitude of the calibrated CA and CAEL variations lie within 0.010 and of CAEH within 0.005,
being mostly stable within the uncertainty range. Indeed, global surface temperature did not increase
much over this period (not shown).
The seasonal cycle of different cloud properties is presented in Figure 14 for six 30° wide latitude bands
ranging from SH polar to NH polar, comparing results from CIRS data and those from the GEWEX
Cloud Assessment data base. As already acknowledged during the GEWEX Cloud Assessment
(Stubenrauch *et al.,* 2013), the seasonal cycles agree quite well between the different data sets, with



exception of the polar regions where passive remote sensing does not perform well and the CALIPSO
data are not conform with the other data sets in the GEWEX Cloud Assessment data base, because they
exclude measurements from 1:30PM during polar night (polar winter) and from 1:30AM during polar
day (polar summer). The most prominent features of the latitudinal seasonal cycles are i) the shift of the
ITCZ towards the summer hemisphere, seen as an amplitudinal signal of 0.1 in CA, 0.3 in CAH and 16
K in CT in the SH and NH tropical bands (mostly over land, not shown) and ii) less clouds in late
summer in the midlatitudes (mostly over ocean and stronger in NH, not shown). The seasonal cycle of
cloud temperature is largest in the polar regions (coherent for all data sets), followed by SH sub-tropical
band, NH midlatitudes, NH sub-tropical band and SH midlatitudes, with amplitudes ranging from 20 K
to 10 K. However, while the CT amplitude is linked to change in cloud height in the low latitudes, it is
more related to change in atmospheric temperature (and corresponding cloud temperature) at higher
latitudes.
**5    Applications**
After the comparisons to other datasets in sections 3 and 4, which have proven the reliability of the CIRS
upper tropospheric clouds, we present in the following two analyses on upper tropospheric cloud
variability with respect to changes in atmospheric conditions to illustrate the usefulness of the CIRS
cloud data for climate studies.
**5.1 Studying hemispheric differences in clouds**
While the NH and the SH reflect the same amount of sunlight within 0.2 Wm$^{-2}$ (Stephens *et al.*, 2015),
there is a small energy imbalance between both hemispheres of our planet, with slightly more energy
absorbed by the SH (0.9 Wm$^{-2}$), yielding more frequent precipitation in the SH and more intense
precipitation in the NH (Stephens *et al.*, 2016). The symmetry in planetary albedo is achieved by
increased reflection of SH midlatitude clouds offsetting the greater reflection of the NH land masses
(Stephens *et al.*, 2015).
The more intense precipitation in the NH is probably linked to the fact that on annual average the ITCZ
peak latitude is about 5°N, shown in Figure 15. On average, total CA is about 10% (0.06) smaller in the
NH than in the SH (excluding the polar regions), without a pronounced seasonal cycle (not shown). This
is linked to more clouds over ocean than over land, producing the increased reflection in the SH
midlatitudes as discussed in (Stephens *et al.,* 2015).  From Figure 15 we deduce that the annual
difference in CAH between NH and SH is 0.05, with a pronounced seasonal cycle of about 0.3 in
amplitude. Results from the three CIRS cloud climatologies, AIRS-LMD, CALIPSO-GOCCP, ISCCP
and MODIS-CE are similar. This seasonal cycle corresponds to the one of the ITCZ peak latitude, which





moves up to 12°N in July. It is also interesting to note that the width of the ITCZ is smaller in July /
August (10.5° ó 12.5°) than in January (17°) and CAH is about 10% larger in August than in January,
which would suggest even more intense precipitation in the NH in boreal summer. For this analysis, the
properties of the ITCZ have been determined by fitting the tropical peak of the latitudinal CAH
distributions per month and year (as in Figure 10). While all datasets agree on the ITCZ peak latitude and
mostly on the ITCZ width (with the Gaussian fit on the ITCZ maximum producing falsely a smaller
width for CALIPSO-GOCCP, because due to ubiquitous thin cirrus, the minima in the subtropics are not
as well pronounced as in the other data sets), MODIS-CE and ISCCP produce smaller absolute values of
maximum CAH because of smaller sensitivity to thin cirrus. The seasonal cycle of maximum CAH is
reduced for CALIPSO-GOCCP and AIRS-LMD due to the inclusion of thinner cirrus (for AIRS-LMD
clouds down to $\varepsilon_{cld} > 0.05$, compared to a threshold for CIRS clouds of 0.10). Figure 15 confirms and
extends the interpretation of the results of (Stephens *et al.*, 2016), by linking the difference in
hemispheric CAH to the shifting of the ITCZ and its stronger intensity in the NH during boreal summer
(smaller width and larger maximum).
**5.2 Studying El Niño-Southern Oscillation (ENSO) effects**
ENSO is the most dominant mode of interannual variability in the Earthøs climate system (e.g. Bjerknes,
1969). The trade winds, blowing from east to west, warm the water as they push it, which leaves warm
water in the West Pacific Maritime Continent (WPMC) and cool water in the tropical East Pacific.
While warm air is rising, building up convection and upper tropospheric clouds, air dries over the cooler
water in the east, thus this SST gradient is responsible for the Walker circulation. ENSO events, El Niño
(warm phase) and La Niña (cold phase), are characterized by large-scale SST anomalies in the tropical
Pacific, compared to the normal situation described above. El Niño events are initiated by a positive SST
anomaly in the equatorial eastern and central Pacific which reduces the east-west SST gradient and
hence the strength of the Walker circulation (Gill, 1980), resulting in weaker trade winds. The weaker
trade winds in turn drive the ocean circulation changes that further reinforce the SST anomaly. The
positive ocean-atmosphere feedback leads to the warm phase of ENSO, which is characterized by strong
rising motion in the central Pacific and a descending branch over the initially strong convective area over
the WPMC. After an El Niño reaches its mature phase, negative feedbacks are required to terminate
growth. According to Lloyd *et al.* (2012), the major source of this negative feedback stems from the
reduction in solar energy at the ocean surface by increased cloud cover over the warm water. Depending
on the location of maximum SST anomalies and associated atmospheric heating, El Niño events may be
distinguished as eastern and central Pacific warming events. A review is given by Wang *et al.* (2016).



The cold phase of ENSO (El Niña) starts with a cold SST anomaly in the tropical Pacific, increasing the
SST gradient and amplifying the Walker circulation, leading to stronger convection and more upper
tropospheric clouds over the WPMC.
To illustrate maximum climate variability patterns in the tropics, we contrast the strongest El Niño and
La Niña events during the AIRS observation period, with multivariate ENSO index of 2.1 in Dec. 2015
and -1.6 in Dec. 2010, respectively. Figure 16 presents geographical difference patterns between these
two ENSO modes in surface temperature and resulting atmospheric parameters, using AIRS-CIRS
cloud data, collocated ERA-Interim data and outgoing longwave radiation (OLR) from NASA-AIRS
(Susskind *et al.*, 2012). As described in the literature, and summarized in the paragraph above, Figure 16
confirms that during an El Niño event East and central Pacific strongly warm, while temperatures are
slightly cooler over the WPMC. The latter is warmer during La Niña. Higher SSTøs lead to more water
vapour in the atmosphere, while the WPMC with its lower SSTøs is drier. The vertical updraft (negative
difference in vertical wind) intensifies in a narrow band just north of the equator over the Pacific west of
the WPMC and a short branch to the South-East, in a typical pattern. The pattern differences in fraction
of opaque high clouds represents the ones of convection, very similar to the updraft pattern, while high-
level clouds increase over a wider part as outflowing anvils, in coherence with increasing water vapour,
while they decrease over the drier WPMC. Thin cirrus increase as parts of anvils in the two branches, but
also in the drier WPMC and North-west of the convective band. The OLR pattern is very similar to the
one of CAH, increasing over WPMC and decreasing where CAH increases over the Pacific. The pattern
of changes in high-level cloud temperature (CTH) shows some differences from the patterns of the other
variables. In general, CTH warms where there are also less high-level clouds and it is lower where the
updraft increases.
So far, the observational period of AIRS and IASI is too short to directly obtain long-term cloud
feedbacks. An alternative approach is to assess cloud feedback in response to interannual climate
variability like ENSO. Dessler (2010) demonstrated that as the surface warms, cloud changes lead to
trapping additional energy, i.e. the longwave cloud feedback is positive. Zelinka and Hartmann (2011)
investigated the response of tropical mean cloud parameters to the ENSO cycle and their effect on top of
atmosphere radiative fluxes. They found during El Niño periods a decrease of high-level cloud amount
as well as an increase in their height which would have opposite effects on the OLR, with a dominating
effect coming from the first. Susskind *et al.* (2012) have shown that global mean and tropical mean OLR
anomaly time series are strongly correlated with ENSO variability, with OLR change resulting primarily
from changes in mid-troposphere water vapour and cloud amount over the WPMC and the East Pacific.



Observed variability in cloud, atmospheric and surface patterns due to ENSO variability can be used to
constrain climate modelling and to understand the processes behind these changes (e. g. Stephens *et al.*,
2017). Though the ENSO related SST anomalies might not correspond to patterns of anthropogenic
climate warming, Zhou *et al.* (2015) have shown that interannual cloud feedback may be used to directly
constrain the long-term cloud feedback. Changes in the geographical pattern and amount of high-level
tropical clouds leads to variations in cloud radiative heating and cooling which then may influence the
large-scale circulation (e.g. Slingo and Slingo 1991, Tian and Ramanathan, 2003).
Since the radiative effects of high opaque clouds and thin cirrus are quite different, we investigate the
geographical patterns of cloud amount changes with respect to tropical mean surface temperature
changes, separately for high opaque, thick cirrus and thin cirrus ($p_{cld} < 330$ hPa, $\varepsilon_{cld} > 0.95$, $\varepsilon_{cld}$ between
0.5 and 0.95 and $\varepsilon_{cld} < 0.5$, respectively). By making use of the whole period between 2003 and 2015
(covering 156 months), we determine a change in upper tropospheric cloud amount as a function of
change in tropical mean surface temperature by a linear regression of their monthly time anomalies, at a
spatial resolution of 1° latitude x 1° longitude. Similar techniques were already used in other studies
related to ENSO and cloud feedback (e.g. Lloyd *et al.*, 2012; Zhou *et al.*, 2014, Yue *et al.*, 2017). Figure
17 presents the change in amount of high opaque cloud (mostly of convective origin), in thick cirrus
(often formed from convective outflow as anvils) and in thin cirrus (which might be formed as anvil or
via in situ freezing) per °C of warming in the tropics (20°N ó 20°S), obtained as the linear slopes of these
monthly time anomaly relationships. The cloud amounts are from AIRS-CIRS, while the surface
temperatures are from the ERA-Interim ancillary data. Results are very similar when using surface
temperatures from AIRS-NASA (Figure S2). Figure 17 also presents geographical patterns of relative
slope uncertainty. In general, large changes in cloud amount per °C of warming have smaller uncertainty
than small ones, indicating robust patterns. Even though the change in tropical mean temperature is
mostly linked to ENSO variability over the studied period and it is still uncertain how to relate these to
long-term patterns due to anthropogenic climate warming, it is very interesting to note that high opaque
clouds and thin cirrus show very different change patterns. While the high opaque clouds, linked to
strong precipitation (Protopapadaki *et al.*, 2017), increase in a narrow band in the tropics, there is a large
increase in thin cirrus around these regions, the latter hypothesized to affect directly the atmospheric
circulation through their radiative heating (e.g. Sohn, 1999; Lebsock *et al.*, 2010). To get a better
understanding on these feedback processes one has to consider the heating rates of these upper
tropospheric cloud systems and link them to the dynamics.



## 6 Conclusions

Two global climatologies of cloud properties have been presented, obtained from AIRS and IASI
observations by the CIRS cloud retrieval. This retrieval software package, developed at LMD, can be
easily adapted to any IR sounder. The retrieval method itself, based on a weighted $\chi^2$ method on
radiances around the 15 μm $CO_2$ absorption band, and the ʻa posterioriʼ cloud detection, based on the
spectral coherence of retrieved cloud emissivities, have already been evaluated in previous publications.
IR sounders are especially advantageous for the retrieval of upper tropospheric cloud properties. Their
good spectral resolution allows a reliable cirrus identification down to an IR optical depth of 0.1, day and
night. The CIRS retrieval applies improved radiative transfer and an original calibration method to adjust
simulated atmospheric spectral transmissivity profiles according to latitudinal, seasonal and interannual
atmospheric $CO_2$ variations. This $CO_2$ calibration method has removed an artificial CA trend of about
4% over the observation period 2004 to 2016, which was directly related to not having taken into
account the anthropogenic $CO_2$ increase in the spectral transmissivities simulated for a specific
atmospheric $CO_2$ concentration. The magnitude of calibrated cloud amount and effective low-level
cloud amount deseasonalized variations lie within 1% and of effective high-level cloud amount within
0.5% over this period.
Common ancillary data (surface temperature, atmospheric profiles) come from the meteorological
reanalyses ERA-Interim, which have been interpolated to the observation times of AIRS and IASI.
Additional application of retrieved AIRS-NASA ancillary data allowed to iteratively make adjustments
to both sets of ancillary data for optimal results in cloud properties and also to estimate uncertainties in
cloud amounts. Since the cloud detection depends on the coherence of spectral cloud emissivity, the
surface temperature influences only slightly the cloud amount (in particular the one of low-level clouds).
AIRS total cloud amount is 67% / 70%, high-level cloud amount 27% / 27% and low-level cloud
amount 27% / 29%, using AIRS-NASA / ERA-Interim, giving an estimate of 5% / 10% uncertainty on
global averages for CA / CAL, respectively. Uncertainties are larger over land and ice / snow than over
ocean, in particular because $T_{surf}$ of ERA-Interim is underestimated in the afternoon and $T_{surf}$ of AIRS-
NASA is underestimated during night due to cloud contamination. In the future, the CIRS cloud retrieval
might use ancillary data from ECMWF meteorological analyses or from the new reanalysis ERA5, both
also having a better temporal resolution.
Cloud / clear sky detection agrees with the one of CALIPSO in 85% / 84% over ocean, 79% / 82% over
land and 73% / 70% over ice and snow, for AIRS-NASA / ERA-Interim ancillary data, respectively.
Typical uncertainties in cloud pressure range from 30 hPa for high-level clouds to 120 hPa for low-level



clouds, coinciding with about 1.2 km in altitude. A comparison with CALIPSO-CloudSat has shown,
that the CIRS retrieved cloud height lies only about 1 km below cloud top in the case of low-level clouds
and about 1.5 km to 2.5 km below cloud top in the case of high-level clouds. The latter leads to retrieved
cloud temperatures which are about 10 K warmer than the cloud top. This has to be considered when
determining radiative effects or when evaluating climate models. The CIRS retrieved cloud height
coincides with the middle between cloud top and apparent cloud base (real cloud base for optically thin
clouds or cloud height at which the cloud reaches opacity), independently of $\varepsilon_{cld}$. When comparing to the
height at which the cloud reaches a VIS optical depth of about 0.5, the CIRS retrieved cloud height, in
the case of high-level clouds, lies about 1 km above for optically thin clouds and about 1 km below for
optically thick clouds. While for low-level clouds the apparent cloud vertical extent is about 1 km, for
high-level clouds it slightly increases with $\varepsilon_{cld}$, from 3 km to 4 km, with slightly higher values in the
tropics than in the midlatitudes, linked to diffusive cloud tops.
Total cloud amount consists of about 40% high-level clouds and about 40% low-level clouds and 20%
mid-level clouds, the latter two only detected when not hidden by upper clouds. Upper tropospheric
clouds are most abundant in the tropics, where high opaque clouds make out 7.5%, thick cirrus 27.5%
and thin cirrus 21.5% of all clouds. IASI values are very similar. The most prominent features of
latitudinal seasonal cycles are the shift of the ITCZ towards the summer hemisphere, with an amplitude
of 0.1 in CA, 0.3 in CAH and 16 K in CT in the SH and NH tropical bands, even stronger over land.
The asymmetry in CAH between Northern and Southern hemisphere with annual mean of 5% has a
pronounced seasonal cycle with a maximum of 25% in boreal summer, which can be linked to the shift
of the ITCZ peak latitude. The latter has an annual mean of 5°N, moving to 12°N with a slightly more
intense ITCZ (smaller width and larger maximum) in boreal summer.
To illustrate further the usefulness of the CIRS cloud data for climate studies, we have finally presented
ENSO effects and tropical geographical change patterns in high opaque clouds and thin cirrus with
respect to tropical mean surface temperature changes. Even though the change in tropical mean
temperature is mostly linked to ENSO variability over the studied period and it is still uncertain how to
relate these to long-term patterns due to anthropogenic climate warming, the large difference in
geographical change patterns of high opaque clouds and thin cirrus indicates that their response to
climate change may be different. which then has consequences on the atmospheric circulation.
To get a better understanding on these feedback processes one has to consider the heating rates of these
upper tropospheric cloud systems and link them to the dynamics. Therefore the AIRS-CIRS and IASI-
CIRS cloud data have been further used to build upper tropospheric cloud systems (based on $p_{cld}$) and




then to distinguish convective cores, cirrus anvil and thin cirrus according to $\varepsilon_{cld}$ (Protopapadaki *et al.*,
2017). These data are being further exploited, together with other data and modelling at different scales,
within the framework of the GEWEX PROcess Evaluation Study on Upper Tropospheric Clouds and
Convection (UTCC PROES, Stubenrauch and Stephens, 2017) to advance our understanding on upper
tropospheric cloud feedbacks.
The AIRS-CIRS and IASI-CIRS cloud climatologies will be made available at the French data centre
AERIS, which also will continue their production.
**7    Data availability**
AIRS L1 data are available at https://mirador.gsfc.nasa.gov/. The NASA Science Team L2 standard
products (Version 6; Olsen *et al.,* 2016) are available at https://mirador.gsfc.nasa.gov/. IASI L1 data are
available at the French Data centre IASI L2 data provided by NOAA, are available at the
Comprehensive Large Array-data Stewardship System (CLASS) center (https://www.
class.ncdc.noaa.gov). The ARSA database is available at : http://climserv.ipsl.polytechnique.fr/fr/les-
donnees/arsa-analyzed-radiosoundingsarchive .html. The operational version of the 4A radiative transfer
model (Scott and Chédin, 1981) is available at http://www.4aop.noveltis.com. The cloud climatologies
of the GEWEX Cloud Assessment data base are available at: http://ipsl.polytechnique.fr/gewexca. THE
AIRS-CIRS and IASI-CIRS cloud climatologies will be made available by the French Data Centre
AERIS.
**Acknowledgements**
This work has been financially supported by CNRS, by the ESA Cloud_cci project and by CNES. The
authors thank the members of the IASI, AIRS, CALIPSO and CloudSat science teams for their efforts
and cooperation in providing the data as well as the engineers and space agencies who control the quality
of the data. We thank the Aeris data infrastructure for providing access to the data used in this study and
for the continuation of the data production.

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



Table 1. Agreement in cloudy and clear sky scenes between CALIPSO and the AIRS-CIRS ʹa
posterioriøcloud detection. Statistics include three years (2007-2009) collocated observations at 1:30 LT.

| surface \ latitude | tropics | | mid- latitudes | | polar | |
|---|---|---|---|---|---|---|
| ancillary data | AIRS | ERA | AIRS | ERA | AIRS | ERA |
| ocean | 86.5% | 84.2% | 90.2% | 91.5% | 93.0% | 95.0% |
| land | 86.4% | 83.2% | 80.7% | 77.6% | 77.3% | 79.7% |
| sea ice | | | 71.5% | 82.0% | 71.2% | 81.2% |
| snow | 73.5% | 71.9% | 74.9% | 68.5% | 65.5% | 66.7% |

Table 2. Averages of a) CA, b) CAHR, c) CAMR and d) CALR (in %) from AIRS-LMD (2003-2009),
AIRS-CIRS (2003-2015, using AIRS-NASA / ERA-Interim ancillary data) and IASI-CIRS (2008-2015,
using ERA-Interim ancillary data).
a)           CA (%)

| latitude band | AIRS-LMD V1 | AIRS-CIRS | IASI-CIRS |
|---|---|---|---|
| globe | 67 | 67 / 70 | 67 |
| ocean | 72 | 71 / 74 | 72 |
| land | 56 | 57 / 59 | 56 |
| 60°N ó 30°N | 69 | 69 / 72 | 69 |
| 15°N ó 15°S | 67 | 63 / 66 | 62 |
| 30°S ó 60°S | 80 | 84 / 85 | 85 |

b)           CAHR (%)

| latitude band | AIRS-LMD V1 | AIRS-CIRS | IASI-CIRS |
|---|---|---|---|
| globe | 41 | 41 / 40 | 40 |
| ocean | 38 | 38 / 37 | 37 |
| land | 48 | 49 / 47 | 47 |
| 60°N ó 30°N | 40 | 40 / 40 | 40 |
| 15°N ó 15°S | 59 | 58 / 57 | 58 |
| 30°S ó 60°S | 28 | 30 / 30 | 29 |




c)                        CAMR (%)

| latitude band | AIRS-LMD V1 | AIRS-CIRS | IASI-CIRS |
|---|---|---|---|
| globe | 18 | 19 / 19 | 20 |
| ocean | 16 | 16 / 17 | 18 |
| land | 23 | 25 / 23 | 23 |
| 60°N ó 30°N | 22 | 23 / 22 | 22 |
| 15°N ó 15°S | 11 | 10 / 10 | 11 |
| 30°S ó 60°S | 21 | 23 / 22 | 23 |

d)                        CALR (%)

| latitude band | AIRS-LMD V1 | AIRS-CIRS | IASI-CIRS |
|---|---|---|---|
| globe | 41 | 40 / 41 | 40 |
| ocean | 47 | 45 / 46 | 44 |
| land | 29 | 27 / 30 | 30 |
| 60°N ó 30°N | 38 | 37 / 38 | 38 |
| 15°N ó 15°S | 30 | 32 / 33 | 31 |
| 30°S ó 60°S | 51 | 47 / 48 | 48 |

Table 3. Averages of relative amount (in %) of opaque ($\epsilon_{cld} > 0.95$), cirrus ($0.95 > \epsilon_{cld} > 0.5$) and thin
cirrus ($\epsilon_{cld} < 0.5$) from AIRS-CIRS (2003-2015, using AIRS-NASA / ERA-Interim ancillary data) /
IASI-CIRS (2008-2015, using ERA-Interim ancillary data).

| latitude band | opaque / tot CA | cirrus / tot CA | thin Cirrus / tot CA |
|---|---|---|---|
| globe | 5.3 / 5.0 / 5.4 | 21.7 / 21.5 / 20.9 | 13.4 / 13.0 / 12.9 |
| ocean | 5.0 / 4.5 / 4.9 | 20.0 / 19.9 / 19.2 | 12.5 / 12.0 / 12.1 |
| land | 6.1 / 5.9 / 6.6 | 25.8 / 25.3 / 24.9 | 15.6 / 15.2 / 14.7 |
| 60°N ó 30°N | 5.4 / 4.8 / 5.4 | 22.9 / 23.5 / 22.8 | 11.1 / 11.0 / 10.9 |
| 15°N ó 15°S | 7.3 / 7.0 / 7.7 | 28.2 / 27.5 / 26.8 | 21.6 / 21.3 / 22.1 |
| 30°S ó 60°S | 4.8 / 4.2 / 4.4 | 17.5 / 18.9 / 18.1 | 6.9 / 6.6 / 5.9 |





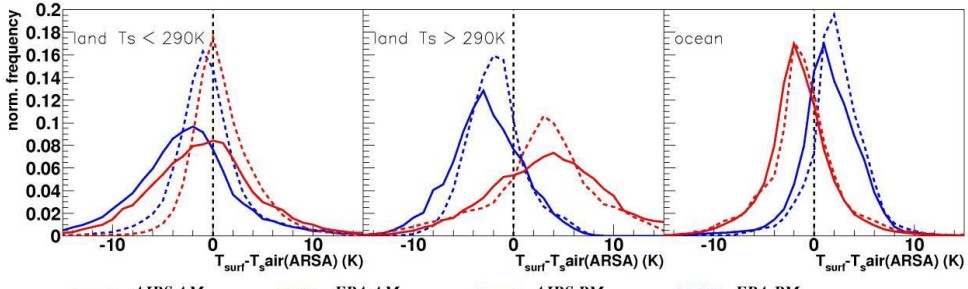

Figure 1. Normalized distributions of the difference between surface skin temperature, as used in the cloud retrieval, deduced from AIRS-NASA of good quality and from ERA-Interim, and collocated surface air temperature of the ARSA data base. Statistics includes January and July from 2003 ó 2015, separately over land for colder temperatures ($T_{surf}$ < 290 K), over land for warmer temperatures ($T_{surf}$ > 290 K) and over ocean.




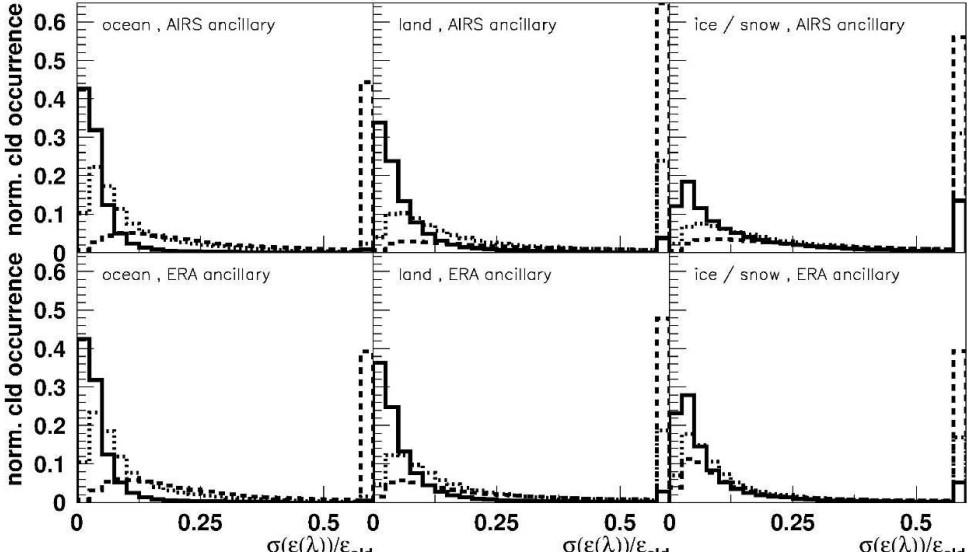

Figure 2. Normalized distributions of spectral variability of effective cloud emissivity over six
wavelengths between 9 and 12 μm divided by cloud effective emissivity retrieved by the $\chi^2$ method,
separately for scenes declared as cloudy (full line), as partly cloudy (dotted) and as clear sky (broken
line) by the CALIPSO samples. Statistics includes three years (2007-2009) of observations at 1:30 LT,
separately over ocean (left), over land (middle) and over ice / snow (right); top: with ancillary data
deduced from AIRS-NASA; bottom: with ancillary data from ERA-Interim.



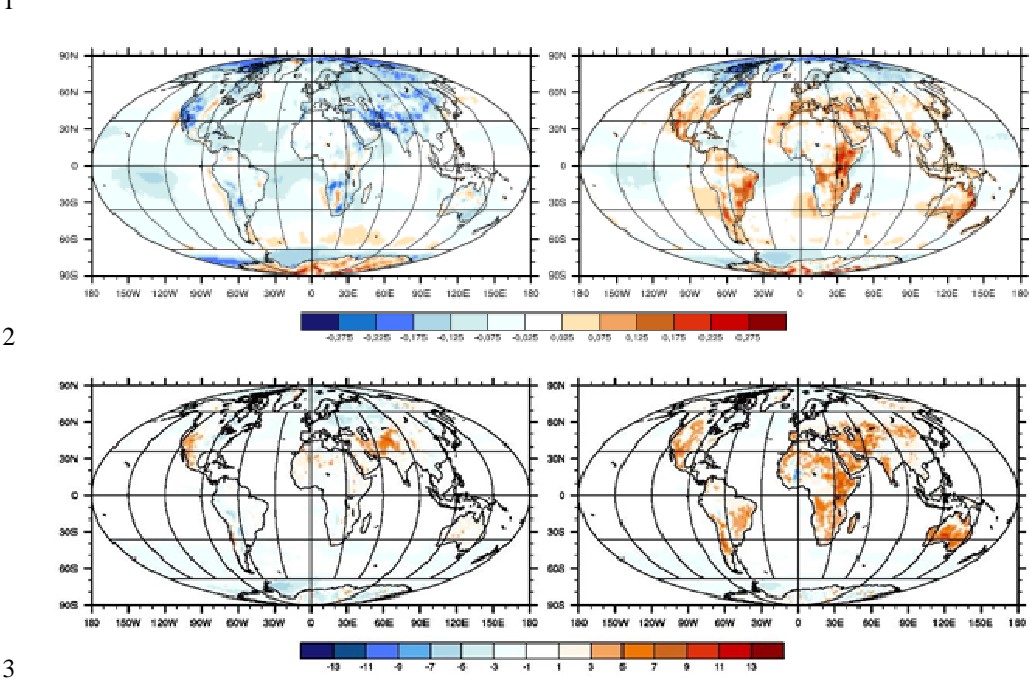

4    Figure 3. Geographical maps of difference in total CA (above) between the two AIRS-CIRS data sets,

5    based on ancillary data from AIRS-NASA and from ERA-Interim, and in $T_{surf}$ (below) between AIRS-

6    NASA and ERA-Interim as used in the retrieval, separately at 1:30AM (left) and at 1:30PM (right)



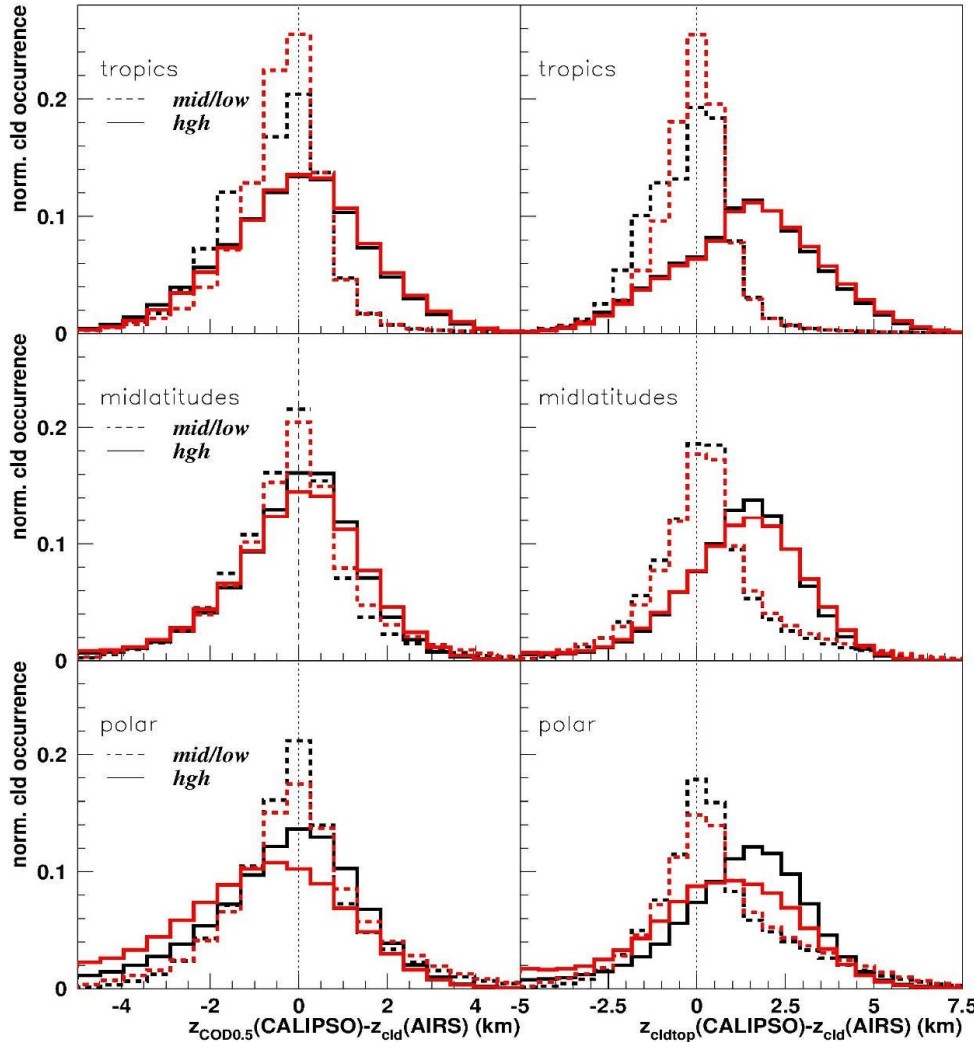

Figure 4. Normalized frequency distributions of the difference between the cloud height at which the
optical depth reaches a value of 0.5 from CALIPSO and $z_{cld}$ from AIRS (left) and between the cloud top
height from CALIPSO and $z_{cld}$ from AIRS (right); $z_{cld}$ from AIRS is compared to the cloud layer of
CALIPSO, coherent with CALIPSO-CloudSat GEOPROF, which is the closest to $z_{cld}$. Statistics
includes three years (2007-2009) of observations at 1:30 LT. AIRS-CIRS cloud retrievals using ancillary
data from AIRS-NASA in red and from ERA-Interim in black, separately for high-level clouds (full
line) and for clouds with $p_{cld} > 440$ hPa (broken line). Analysis over three latitude bands: 30°N-30°S
(upper panel), 30°-60° (middle panel) and 60°-85° (lower panel).

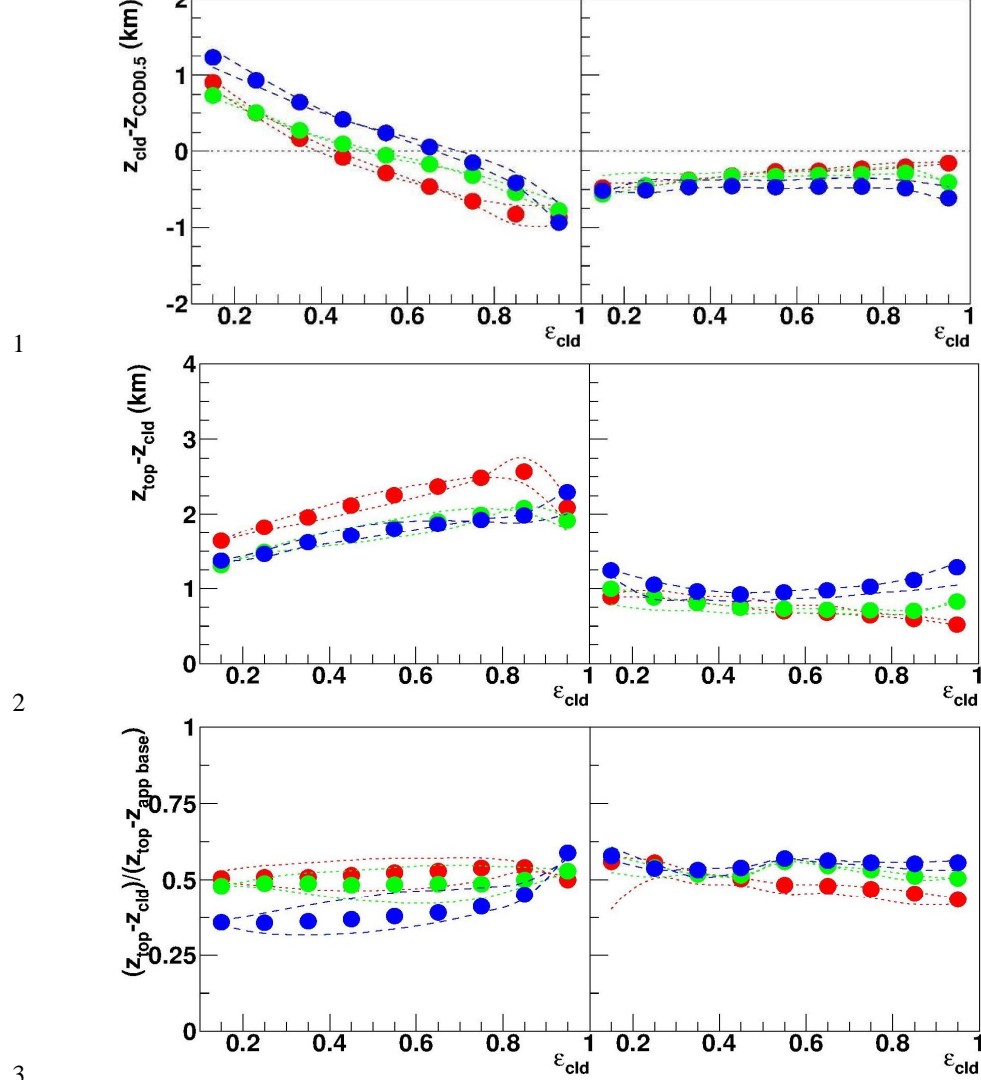

Figure 5. Average difference between $z_{cld}$ from AIRS-CIRS and CALIPSO height at which the COD
reaches about 0.5 (top), between $z_{top}$ from CALIPSO and $z_{cld}$ (middle) and between $z_{top}$ and $z_{cld}$ scaled by
'apparent' cloud vertical extent (bottom) as function of AIRS-CIRS cloud emissivity for clouds in the
tropics (red), midlatitudes (green) and polar latitudes (blue), separately for high-level clouds (left) and for
low-level (right) clouds. Three years of statistics for cases where $z_{cld}$ from AIRS and CALIPSO height
lie within vertical cloud borders determined from CloudSat-CALIPSO GEOPROF. Observations at 1:30
LT. Statistical errors are negligible and the broken lines indicate a range between single layer clouds and
multi-layer clouds.



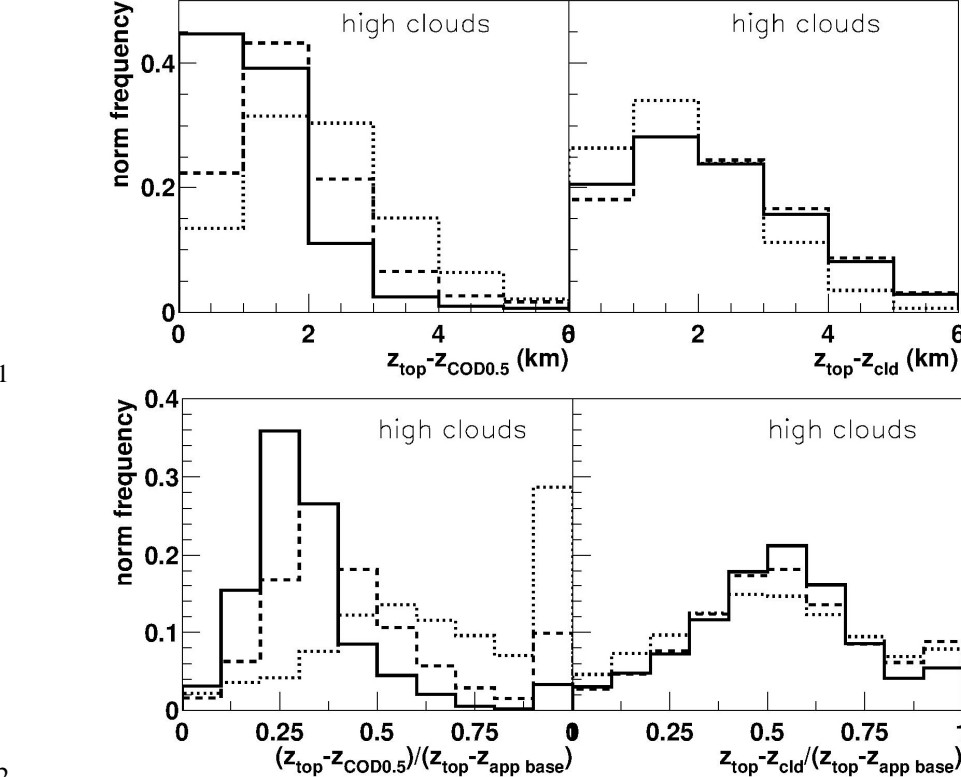

Figure 6. Normalized frequency distributions of differences between CALIPSO cloud top and height at which the COD reaches about 0.5 (left) and between CALIPSO cloud top and $z_{cld}$ from AIRS (right) for high-level clouds, in absolute values (top) and scaled by apparent vertical cloud extent (bottom). Distributions are compared for clouds with $\varepsilon_{cld} > 0.8$ (full line), $0.8 > \varepsilon_{cld} > 0.4$ (broken line) and $0.4 > \varepsilon_{cld} > 0.1$ (dotted line). Three years of statistics for cases where $z_{cld}$ from AIRS and CALIPSO height lie within vertical cloud borders determined from CloudSat-CALIPSO GEOPROF. Observations at 1:30 LT.





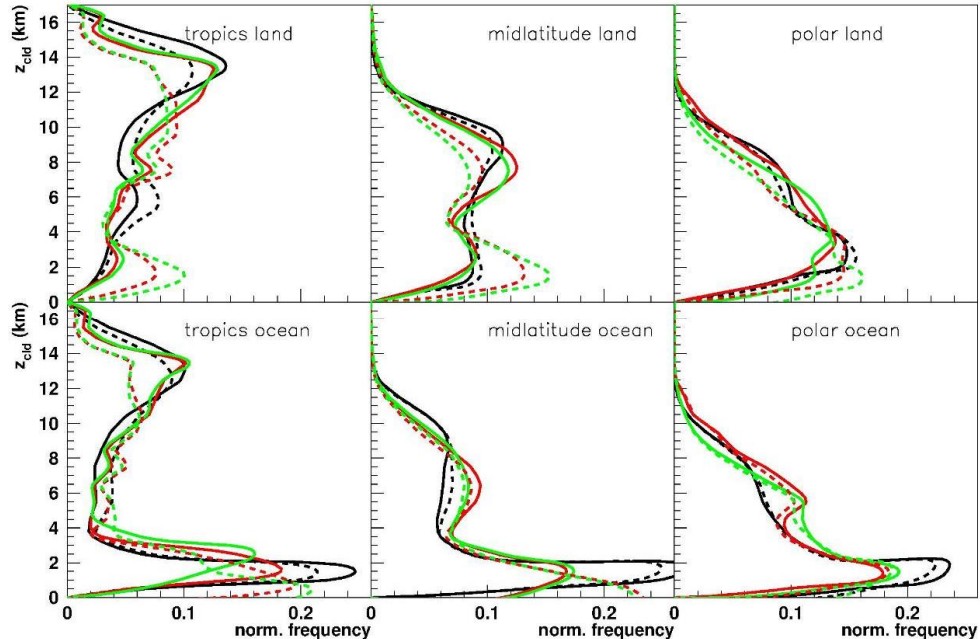

Figure 7. Normalized frequency distributions of $z_{cld}$ from AIRS, using ancillary data from AIRS-NASA
(red) and from ERA-Interim (green) and $z_{COD0.5}$ from CALIPSO (black), separately over land (top) and
over ocean (bottom), in the tropics (left), midlatitudes (middle) and polar latitudes (right). For each data
set two distributions are shown: for all detected clouds, except subvisible cirrus (broken line) and for
mostly cloudy fields of single layer clouds (full line).





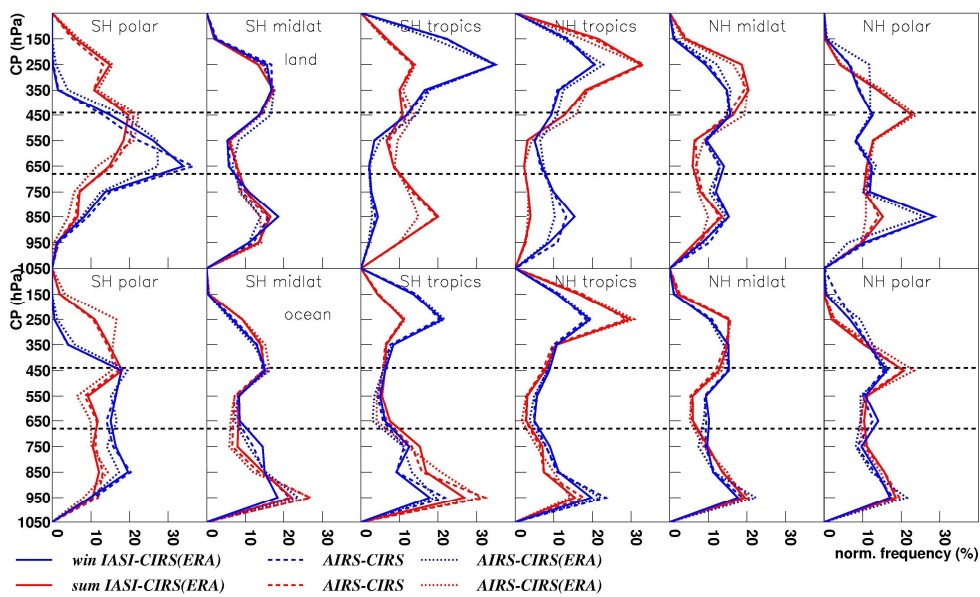

Figure 8. Normalized frequency distributions of $p_{cld}$ from AIRS-CIRS, using ancillary data from AIRS-NASA (dashed line) and from ERA-Interim (dotted line), as well as from IASI-CIRS (full line), separately over land (top) and over ocean (bottom) in six latitude bands of 30° from Southern hemisphere polar (left) to Northern hemisphere polar latitudes (right), in boreal winter (December, January, February; blue) and in boreal summer (June, July, August; red). Statistics from 2008.





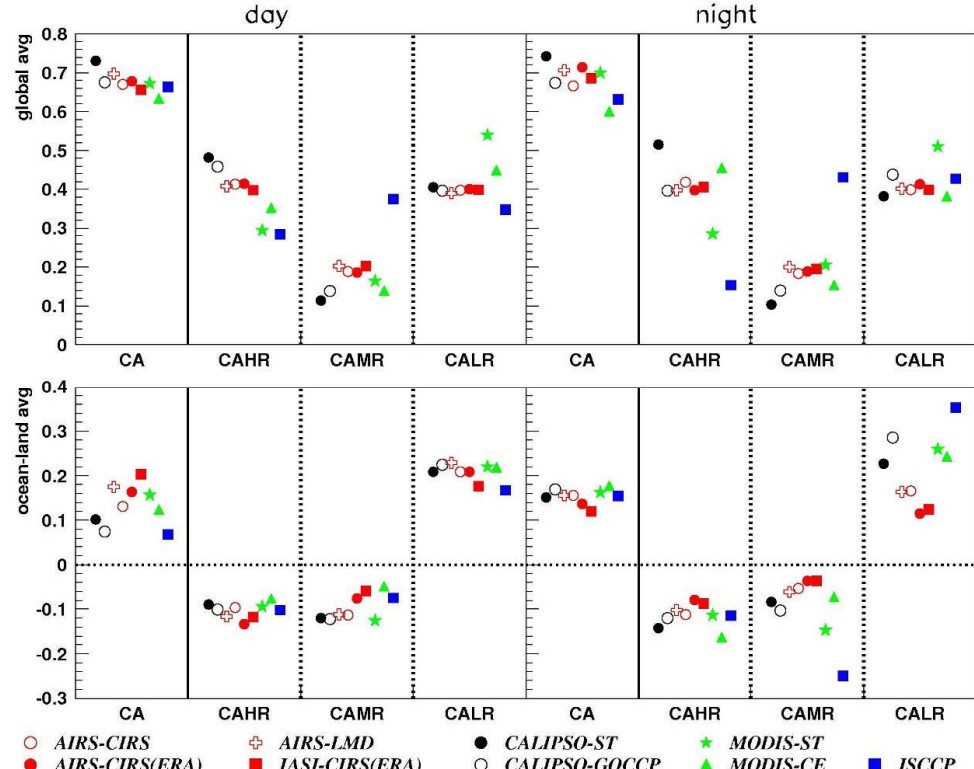

Figure 9. Top: Global averages of total cloud amount (CA), as well as of fraction of high-level, mid-level and low-level cloud amount relative to total cloud amount (CAHR + CAMR + CALR = 1). Comparisons of IR sounder cloud data (AIRS, IASI) with L3 data from the GEWEX Cloud Assessment data base, separately for observations mostly during day (left), corresponding to 1:30PM (3:00PM for ISCCP and 9:30AM for IASI), and mostly during night (right), corresponding to 1:30AM (3:00AM for ISCCP and 9:30PM for IASI). Bottom: Averages of ocean-land differences for the same parameters.



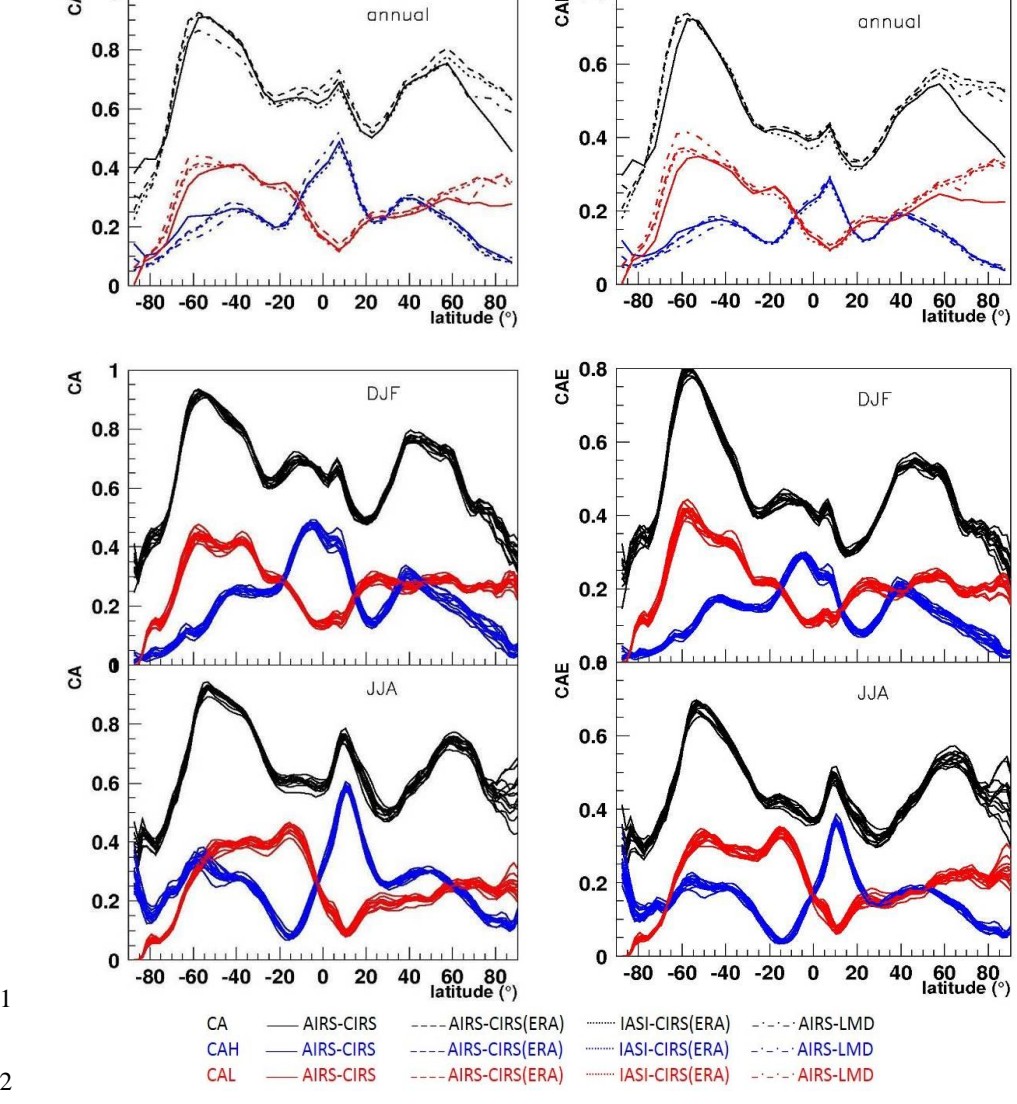

Figure 10. Zonal distributions of CA, CAH and CAL (left) and CAE, CAEH and CAEL (right),
separately as annual mean (top), in boreal winter (December, January, February; middle) and in boreal
summer (June, July, August; bottom). For the annual mean, cloud amounts are compared between
AIRS-CIRS, using ancillary data from AIRS-NASA (full line) and from ERA-Interim (broken line),
IASI-CIRS (dotted line) and AIRS-LMD (dash-dotted line. For boreal winter and boreal summer,
AIRS-CIRS (using AIRS-NASA ancillary data) is shown separately for each year between 2003 to
2015, illustrating inter-annual variability .



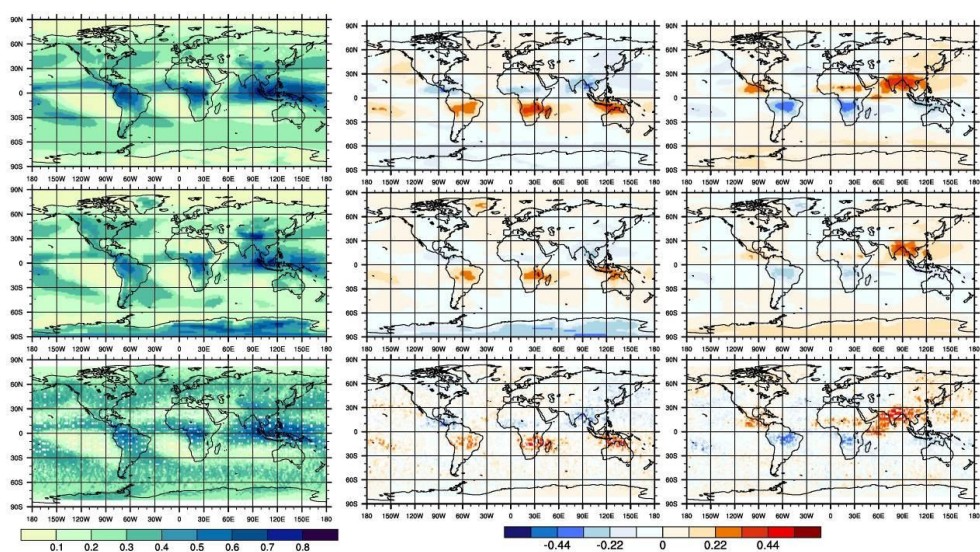

Figure 11. Geographical maps of annual CAH (left) of AIRS-CIRS (2003-2015, top) compared to
ISCCP (1984-2007, middle) and CALIPSO-GOCCP (2007-2008, bottom) from the GEWEX Cloud
Assessment data base, as well as seasonal anomalies of DJF (middle) and of JJA (right).

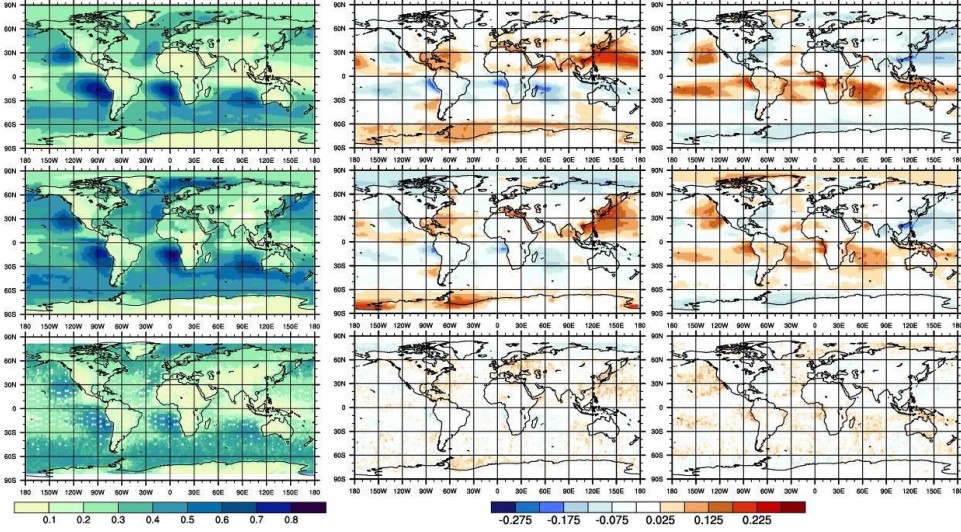

Figure 12. Geographical maps of annual CAL (left) of AIRS-CIRS (2003-2015, top) compared to
ISCCP (1984-2007, middle) and CALIPSO-GOCCP (2007-2008, bottom) from the GEWEX Cloud
Assessment data base, as well as seasonal anomalies of DJF (middle) and of JJA (right).



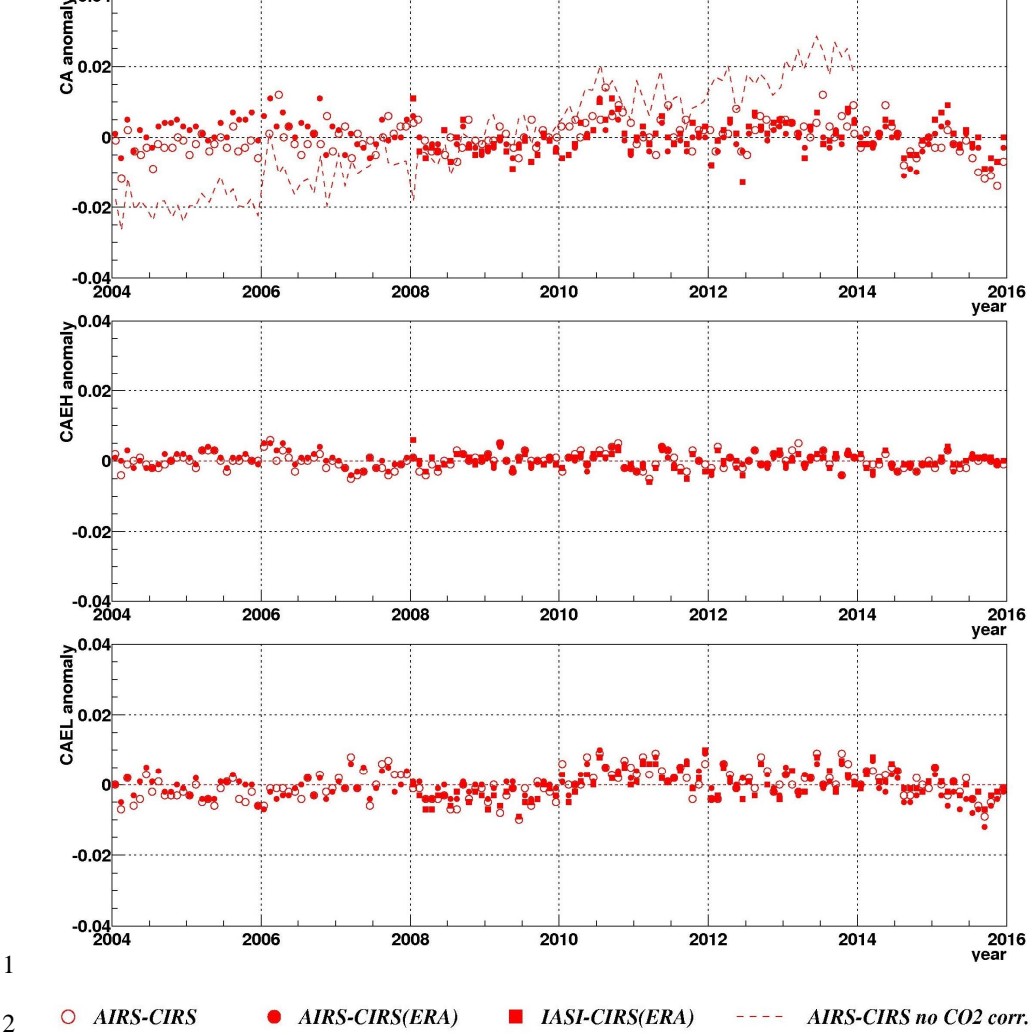

○ *AIRS-CIRS*    ● *AIRS-CIRS(ERA)*    ■ *IASI-CIRS(ERA)*    - - - - *AIRS-CIRS no CO2 corr.*
Figure 13. Time anomalies of deseasonalized CA, CAEH and CAEL over the globe. In the case of CA,
additional values are shown without calibration of spectral atmospheric transmissivities for changes in
atmospheric $CO_2$ concentration.





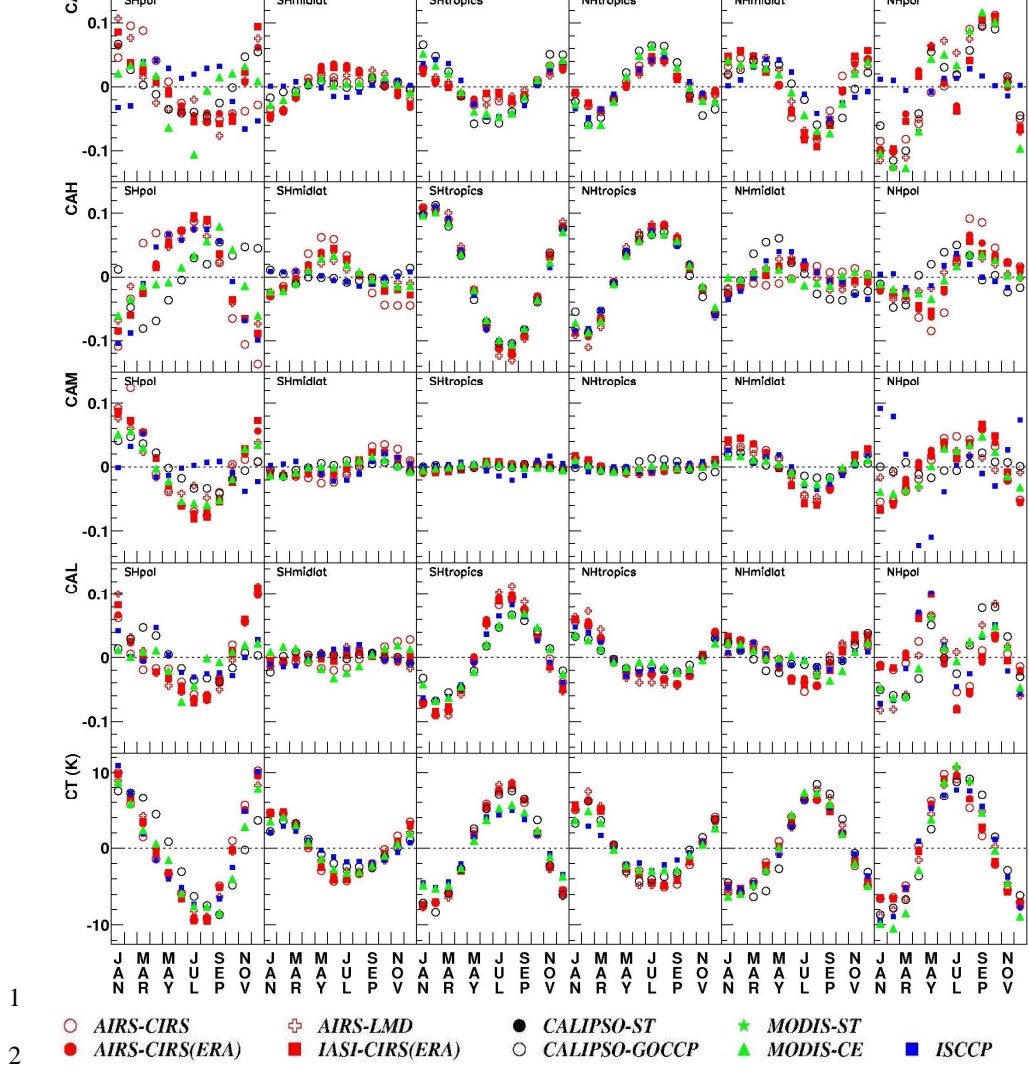

3    Figure 14. Seasonal cycle of CA, CAH, CAM, CAL and CT over 30° wide latitude bands from SH

4    polar to NH polar.





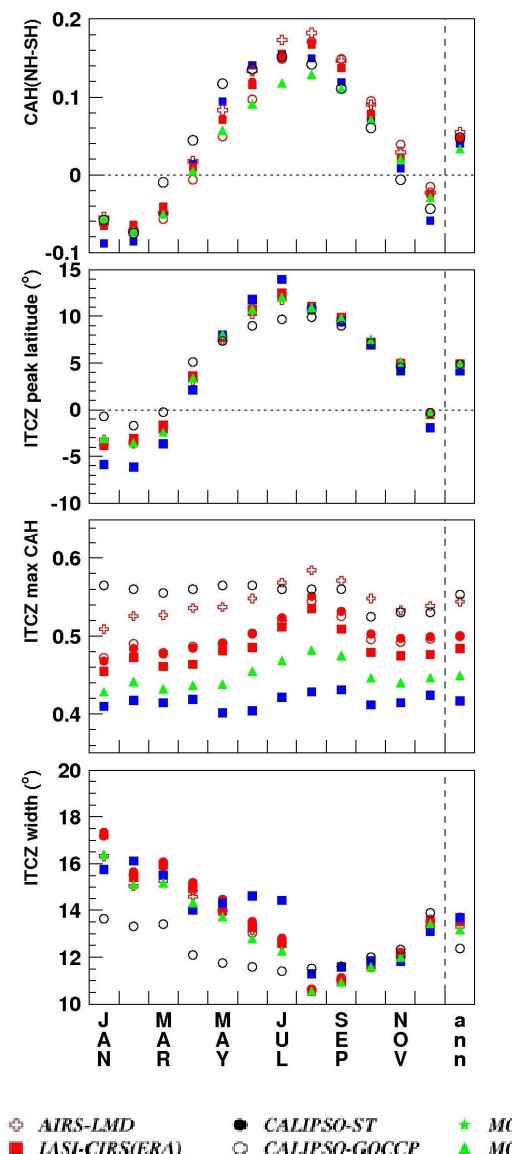

3    Figure 15 Seasonal cycle / annual average of CAH differences between NH hemisphere and SH

4    hemisphere (60N-60S); seasonal cycle / annual average of ITCZ peak latitude, maximum CAH within

5    ITCZ and ITCZ width.





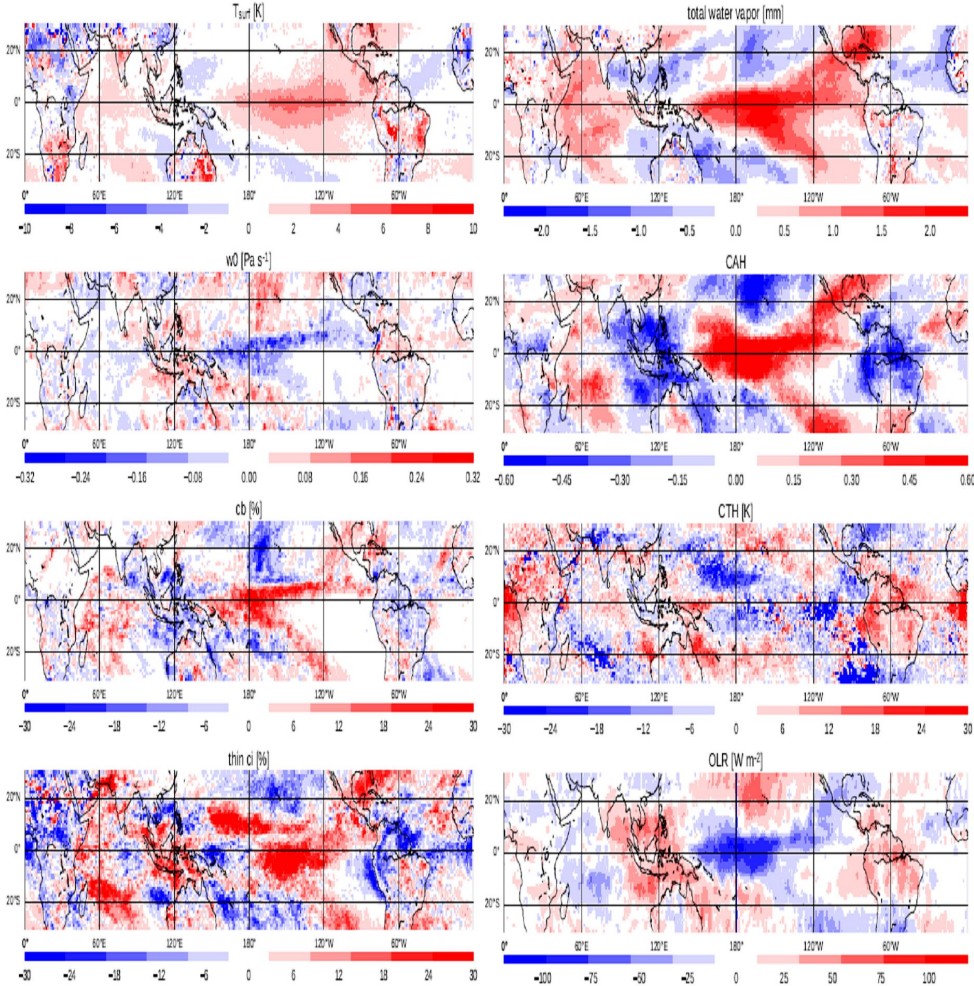

Figure 16. Differences between December 2015 and 2010, corresponding to El Niño and La Niña,
respectively, in T$_{surf}$ (1. Panel, left), total atmospheric water vapour (1. Panel, right) and vertical wind at
500 hPa (2. Panel, left) from ERA-Interim, in CAH (2. Panel, right), fraction of Cb (3. Panel, left), cloud
temperature of high-level clouds (3. Panel, right) and fraction of thin cirrus (4. Panel, left) from AIRS-
CIRS, and OLR (4. Panel, right) from AIRS- NASA.





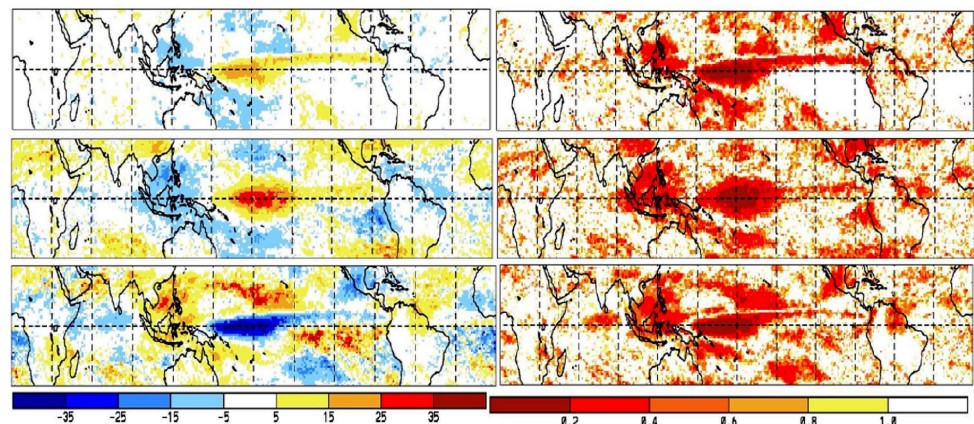

Figure 17. Left: Slopes of change in Cb (top), cirrus (middle) and thin cirrus (bottom) amount in % per
°C of tropical warming (20°N ó 20°S); right: relative slope uncertainty for Cb (top), cirrus (middle) and
thin cirrus (bottom) amount change per °C tropical warming. Results using upper tropospheric ($p_{cld}$ <
330 hPa) cloud type anomalies from AIRS-CIRS and surface temperature anomalies from ERA-Interim
of 156 months during the period 2003-2015.

