# Peer review of "Cloud climatologies from the InfraRed Sounders AIRS and IASI"

_Atmospheric Chemistry and Physics, 2017_

## Referee Comment (RC1) · Anonymous Referee #2 · 21 Jul 2017

Review of Atmospheric Chemistry and Physics manuscript by Stubenrauch et al. entitled

**"Cloud climatologies from the Infrared Sounders AIRS and IASI: Strengths, Weaknesses and Applications"**

**General impression and recommendations**

This manuscript presents in detail a cloud property retrieval method, CIRS (Clouds from IR Sounders), which has been adapted to recent hyperspectral sounding datasets from the AIRS and IASI sounding instruments and which has a capability to be applied also to other existing (e.g., CrIS) and planned sounders. Results have been compared to A-Train data (mainly CALIPSO and CloudSat data) showing very encouraging results. Finally, Level 3 products (monthly, seasonal and yearly means) are demonstrated and compared to other existing climate data records. The study also highlights the necessity to account for changing atmospheric $CO_2$ concentrations in the use of the method for generation of climate data records.

I have no particularly critical or serious points questioning the core method but there are still issues that need to be further explained and commented. Such issues are e.g. related to

- the role of CALIPSO-CALIOP data for tuning the method
- the exact description of the used CALIPSO dataset for tuning and for evaluation of cloud properties
- the consequence of using some unphysical assumptions in the retrieval
- the balance between finding spectral coherence in the solutions and still maintain physically reasonable emissivity differences
- justification of the statement of achieving successful cloud detection down to IR cloud optical thicknesses of 0.1

More details here (and also several minor issues) are given below in the list of Specific Comments.

Quite a lot of editorial suggestions are given and there is also a need to improve several of the figures to make them easier to understand.

A more serious issue is the length of the manuscript. 52 pages of text (36 pages) and figures (16 pages) appear to be too ambitious here and it will be difficult for

readers (and even reviewers!) to digest. I suggest that Section 5 on applications is removed (see specific comment number 25) in an attempt to shorten the paper.

In conclusion, I recommend publication of this manuscript after addressing the detailed comments below and after implementing the suggested shortening of the manuscript.

**Specific comments**

1. Page 1, Abstract, line 19, "to evaluate":

   The term "to evaluate" should be changed to "to design and evaluate".

   You used A-train data to find your 'a posteriori' cloud masking thresholds, right? Then you should be clear in your description that A-train data is not completely independent from your data/method. This is important for the reader to know.

2. Page 1, Abstract, line 23, "coincides":

   To use the term "coincides" here is a too strong conclusion from your results. Figure 6 (lower right panel) clearly shows a rather broad distribution of results where frequencies at the two extremes (0 and 1) are still about 20-25 % of the frequency for the value 0.5 (representing the middle of the defined layer).Therefore you can possibly only state that the cloud height can be "approximated" by the middle of the defined layer. Also "middle" could possibly be replaced by "the mean layer height" to make the description scientifically stricter.

3. Page 1, Abstract, line 27, "apparent vertical cloud extent":

   The explanation here is confusing, indicating that upper level clouds generally have higher cloud emissivities than lower level clouds. This cannot be true. I guess the authors mean something else. Please clarify!

4. Page 2, Abstract, lines 5-8, "response to climate change" + Page 3, Section 1, lines 23-25 and the entire section 5:

   The last sentence in the abstract, the sentence about Section 5 in Section 1 and the entire section 5 could possibly be removed for shortening the paper (see also comment 25!).

5. Page 2, Section 1, line 11, "70 % cloud cover":

   Although this is a widely used and accepted figure for global cloudiness, I would like to point out that a value of global cloud cover cannot be stated without first defining what you mean by a cloud. The figure 70 % is kind of representing clouds which have a significant impact on radiation budgets and it could possibly be relevant if you define that clouds should have at least a cloud optical thickness of approximately 0.2. But if including also the thinnest clouds (often called sub-visible clouds and so far

only observed by high sensitive instruments like CALIPSO-CALIOP) the figure may increase to values well above 80 %.
I think it would be appropriate to at least make a short statement on what clouds are considered when stating that global cloudiness is about 70 %.

6.  Page 3, Section 1, line 3: "optical depth less than 3"

    My impression is that the capability is better than that, i.e., the capability of having reasonable cloud optical depth estimations from CALIOP data covers the interval 0-5. Please check that the value of 3 is really justified.

7.  Page 7, Section 2.4, line 4, "emissivities larger than 1":

    I must say that it is quite disturbing to "be forced" to use unphysical values in the retrieval. I understand that uncertainties can lead to this but I am not sure that this is then the best way of handling these uncertainties. Why not restrict emissivities to 1 in the optimization/minimization process when knowing that this is physically correct? I can't see why your present method gives better uncertainty descriptions of the retrieved cloud pressures than when using a restricted emissivity value. Don't inconsistencies give rise to new inconsistencies? Please explain and motivate.

8.  Page 7, Section 2.4, lines 22-28, "a posteriori cloud detection":

    The "a posteriori cloud detection" has already been briefly introduced (page 4, lines 7-11). Why repeating this information here?  Delete these lines or move part of this to the relevant section 2.5.

9.  Page 9, Section 2.4.1, lines 18-20, "ocean cloud amounts larger during night":

    To find larger ocean cloud amounts at night than during day is found in many regions (e.g. over marine stratocumulus areas). What made you think this was a problem specifically for ERA-Interim? Please explain.

10. Page 10, Section 2.4.2:

    The $CO_2$ correction appears to be a very relevant change (also visualized nicely in Figure 13. This appears to be one of the most important improvements of the methodology. Should become mandatory in all sounding-based retrievals for climate datasets, in my opinion.

11. Page 11, Section 2.5, general comment on the "a posteriori cloud detection":

    The methodology appears a bit awkward compared to many other cloud retrieval methods in that cloud properties are first derived and then a determination whether a FOV is cloudy is carried out as a second step. Most common otherwise is that a cloud screening is done first and then followed by a cloud property retrieval. So, could you confirm that after having performed the cloud property retrieval, all FOVs are still assumed to be cloudy?  Does it mean that you will always find a solution to Equation 2? You have already mentioned some problems in finding a distinct minimum for lowlevel clouds (page 7, lines 2-3) but what happens in obviously cloud-free situations?

12. Page 11, Section 2.5, line 16 + lines 20-21, "meaning of spectral coherence":

I am a bit concerned about the concept indicating that, for a cloud to be identified, the differences between emissivities in the six infrared channels should be small. In this wavelength region we know that the refractive indices of water and ice, respectively, varies considerably. For example, this is one of the fundamental properties that allows separating water clouds from ice clouds in passive imagery (e.g. as introduced by Pavolonis et al., 2005, J. Appl. Meteorol.). This fact would also certainly introduce considerable differences in cloud emissivities depending on if it is a water or ice cloud in addition to variations in optical thickness or partial coverage within each FOV. So, isn't there a risk that the demand on spectral coherence is in conflict with reality? Or are you able to find a balanced and optimized method based on reference observations from CALIPSO-CALIOP data and still retain reasonable resulting emissivity differences? I guess that the access to CALIPSO-CALIOP data here is essential since it would be difficult otherwise (e.g. through detailed cloud model simulations) to find an optimal way here. Please comment.

13. Page 11, Section 2.5, line 25, "standard deviation":

How do you calculate the standard deviation here? Do you use all values in the "AIRS golf ball" (i.e., 9 values) for the calculation for each wavelength? The current description is not clear enough on this.

14. Page 11, Section 2.5, line 27, "CALIPSO samples":

Unfortunately, here you introduce the use of CALIPSO data without having described what data you actually used (this description comes later in Section 3.1). More clearly, it is not obvious to the reader that you will get three CALIPSO samples in the AIRS golf ball. For this, you need to know that you use 5 km CALIPSO data. Because of the importance of A-train data for your method and study, I am of the opinion that you should have introduced them already in Section 2 on "Data and Methods". Can you consider changing this?

15. Page 12, Section 2.5, lines 18-19, "minimum optical depth":

In the introduction section you mention that with IR vertical sounding data "reliable detection of cirrus with IR optical depths as low as 0.1" is possible indicating that this is much better than what can be achieved from other sensors (except from active sensors). I wonder what this restriction in order "to reduce noise" means in this context? Have you estimated further the minimum cloud optical depths being detected after introducing this restriction? CALIPSO-CALIOP offers the possibility to do such in-depth studies.

16. Page 13, Section 3.1, lines 16-19, "CALIPSO and CloudSat data":

This requirement should mean (?) that you require that both CloudSat and CALIPSO say it is cloudy. But what about the fact that CALIPSO sees much more of the very thin cirrus clouds being available? Does it mean that these cirrus cases are not

included in your evaluation study despite the fact that you several times have emphasized the capability of your method to detect very thin cirrus? Or is it different for studies of cloud amount (as indicated by description in lines 7-15) and cloud top height? Please comment!

17. Page 13, Section 3.1, line 23, "underestimated COD":

Just for your information: The latest version of the CALIPSO-CALIOP dataset (version 4.1) gives indeed higher CODs. This change can possibly be connected to what you write here (currently I do not know the details behind this change).

18. Page 14, Section 3.2, lines 2-3, "agreement":

I have to ask you to specify better what you mean by "agreement". There are so many skill scores around so you'd better be strict in describing exactly the measure you use. I guess you refer to what is normally called "Hit Rate" which is the number of correct cloudy AND clear cases divided by the total number of cases.

19. Page 14, Section 3.3, generally on results in Figure 4 (Page 40):

First, please revise the wording of the caption of this figure. The first sentence here is too complicated and the description should possibly be made more clear (the same is actually true for Figure 5). Also make clear (in all figures) what you mean by "1:30 LT" (AM or PM??).
The question raised in the previous comment 16 remains: Are thin cirrus detected by CALIPSO but not by CloudSat part of this study or not? If not, what can be said about the quality of these retrieved cloud heights (as compared to CALIPSO data alone)?

20. Page 15, Section 3.3, line 9, "coincides":

See previous comment 2.

21. Page 16, Section 3.3, lines 5-24, Figure 7:

Very interesting and impressive results shown here! Results for medium and high clouds are probably quite superior to those being presented from passive imagery in other CDRs. Only for low-level clouds we still see quite some discrepancies which is understandable for several reasons. This indicates that the best representation of the true vertical distribution of cloudiness in a climate sense could be a combination of sounding and passive imagery data. Do you agree? Maybe you should mention this. Interesting is that problems for low clouds for sounding applications is not showing up very clearly later in Figure 9, except possibly during night for the land-ocean difference. Maybe you should explain why?

22. Page 16, Section 3.3, line 32, "coincides":

See previous comment 2.

23. Page 18, Section 4, lines 15-16, "sensitivity of lidars":

You write that "active lidar is the most sensitive". Quite true but you haven't explained whether CALIPSO results in Figure 9 are already "filtered" (so that the thinnest clouds as given by the original CALIOP CLAY product are removed) or not. Has there been any filtering of 'sub-visible clouds' (I assume there has)?
This is a relevant question to ask also for the statement in the Conclusions section on page 25, line 25. We need to know exactly what is the used CALIPSO dataset used as reference!

24. Page 21, Section 4, line 4, Figure 14, "Seasonal cycle of cloud temperatures":

How come there is a rather large consensus between different methods when studying cloud temperatures for the polar areas (leftmost and rightmost columns) when the spread is very large when it comes to cloud amount (top row of the same columns)? I suspect it is an indication of that cloud temperatures and surface temperatures are very similar here. This implies (in my opinion) that the separation of cloudy and cloud-free areas is indeed not very accurate. So, where is really the truth as regards polar cloudiness?
Apart from this reflection, I consider Figure 14 as a very nice compilation of global cloudiness and its variation.

25. Pages 21-24, Section 5, "beyond scope??"

In my opinion, Section 5 feels like out of scope of this study. Although introducing highly interesting topics (especially section 5.2), this work would benefit from being presented as a separate (or companion) publication.
This manuscript is very, very long and it will put the readers (as it truly has for reviewers!) to a real test when digesting it. I would say that especially section 5.2 on the ENSO effects and its coupling to cloud/radiation feedbacks also requires a different category of expertize for reviewing it with more focus on modelling and studies of climate change and climate feedback effects. Consequently, I have not provided specific comments on this section and I suggest that it is removed for the shortening of this paper.

26. Page 26, Section 6, line 1, "coincides":

See previous comment 2.

27. Page 24-27, Section 6, general comment:

A very comprehensive and good summary of the content of the paper. However, it could be shortened (page 26, lines 14-32) as a consequence of comment 25 above.

**Technical corrections**

1. Page 1, Abstract, line 11-14:

The current introductory sentences assumes that the reader already knows about the LMD cloud retrieval scheme. I suggest a slight reformulation to make it less unclear,

e.g. like the following

"The Laboratoire de Météorologie Dynamique (LMD) cloud retrieval scheme CIRS (Clouds from IR Sounders) has been adapted to cope with any Infrared (IR) sounding instrument. This has been accomplished by applying improved radiative transfer calculations as well as by introducing an original method accounting for atmospheric spectral transmissivity changes associated with varying $CO_2$ concentrations".

2. Page 2, Abstract, line 3, "5 % asymmetry":

Please clarify better what you mean with asymmetry. Does it mean that there is generally 5 % more high clouds in the Northern Hemisphere? I assume this is what you mean (supported also by Figure 10) but you should make it crystal clear for the reader in the Abstract!

3. Page 2, Section 1, line 17, "properties":

Do you really mean "properties"? I would rather say "cloud detection".

4. Page 2, Section 1, line 32, "determine":

Like the previous comment, I am not sure about the correct wording here. The word "determine" is very strong and almost indicates that the CALIPSO and CloudSat satellites together are creating/defining the clouds (☺). Rather, you should express that they "are capable of observing the cloud vertical structure".

5. Page 3, Section 1, line 5, "the cloud retrieval method":

Be a bit more specific, e.g. write "the evolution of the original cloud retrieval method".

6. Page 3, Section 1, line 9, "radiative transfer":

I think you should write "radiative transfer calculations" or "radiative transfer modelling". To only write "radiative transfer" is too general and (I guess) just a shortening of more correct terms.

7. Page 3, Section 1, line 11, "initial": See 5 above (consider using same notation).

8. Page 3, Section 1, line 11, "radiative transfer": See 6 above (consider using same notation).

9. Page 4, Section 2.1, line 11, "The NASA Science team….":

I would recommend to start a new paragraph here to increase the readability.

10. Page 4, Section 2.1, line 15, "Susskind et al, 2003":

I see inconsequent reference formulations on several places in the manuscript. When you make a direct reference to other publications directly in the text (like here) you should (according to my experience) preferably write:

"The methodology is essentially unchanged from that described in Susskind et al. (2003)."

You have done this correctly in other places (e.g., Page 5, line 27). I think you should be consistent here. Use the formulation above when specifically discussing a publication and use reference in parenthesis when not making a direct statement of the referred publication (a "softer" reference).

Check also the following references for the same reason:

- Page 4, line 27
- Page 6, line 5

11. Page 4, Section 2.1, line 20, "shortwave window channels":

Please write "shortwave infrared window channels" since "shortwave" most often is reserved to define visible channels.

12. Page 4, Section 2.1, line 22, "partial cloud cover":

A better formulation is probably "under partially cloudy conditions".

13. Page 4, Section 2.1, line 24, "snow or ice":

Maybe a better formulation is "…snow or ice covered surfaces also provided by NASA L2 data".

14. Page 4, Section 2.1, line 26, "ideology":

I would suggest using the term "concept" rather than "ideology".

15. Page 4, Section 2.1, line 27, "and allow":

I suggest replacing this with "which allows".

16. Page 5, Section 2.2, line 1, "12 km":

Is the 12 km valid for each individual footprint or the 2x2 array?

17. Page 5, Section 2.2, line 9, "the cloud retrieval":

You should write "the CIRS cloud retrieval".

18. Page 5, Section 2.2, lines 9-10, "retrieved atmospheric profiles":

Be more specific. You should write "IASI-retrieved atmospheric profiles".

19. Page 5, Section 2.2, line 15, "Therefore":

You should not start a new paragraph here if you refer directly to what was written in the previous sentences. Make it also very clear that you never (well, not in time for your development) got access to EUMETSAT Version 6 data otherwise this statement appears rather strange.

20. Page 5, Section 2.2, line 21, "same source":

I guess you rather mean a "less instrument-dependent source"?

21. Page 6, Section 2.3, line 1, "proxy":

I don't like the word "proxy" in this context. It indicates that it is a kind of simulation or approximation of the real vertical velocity. The vertical pressure velocity $\omega$ is just another formulation of the vertical velocity which arises when you use pressure as your vertical coordinate instead of the standard geometrical height in meters. So, to my knowledge, it's the "real thing" and not a "proxy".
But I guess you refer to the fact that the direct calculation of $\omega$ is difficult without making approximations. The most common here is the geostrophic assumption leading to the so-called "$\omega$-equation". In this sense, I guess you may be correct in interpreting it as an approximation. But still, present day NWP models are capable of calculating $\omega$ so I just wonder what value you are using here? On the other hand, the approximated value at the 500 hPa level is probably quite accurate anyway (conditions here are largely quasi-geostrophic on the large scale) so perhaps this discussion is less important. Anyway, give it a thought.

22. Page 7, Section 2.4, line 12, "arise":

Maybe reformulate to "these cases occur in about 7 to 15 % of all cases"?

23. Page 8, Section 2.4.1, line 14, "less than ..?..":

Strange formulation. You'd better write "0.99 for wavelengths less than 10 μm and 0.98 for wavelengths larger than 10 μm".

24. Page 13, Section 3.1, line 6, "spatial resolution CALIPSO":

Shouldn't it be "5 km x 0.3 km"? I thought the basic FOV of CALIOP was 300 meter.

25. Page 15, Section 3.3, Figure 5 (Page 41):

I suggest that you try to include some additional explanatory features or legends in the figure (e.g., legend with the three coloured dots explained). To look for all explanations in the caption is not very reader-friendly. Try to speed up the correct interpretation of figures with the use of more graphical legends or marks. This remark is probably valid for many other figures in the manuscript.

26. Page 15, Section 3.3, line 27, "Considering…":

I suggest starting a new paragraph here in order to avoid too long chunks of text (unnecessary tiring for the reader).

27. Page 15, Section 3.3, line 28; Figure 6 (Page 42):

In the caption you describe one of the curves as "broken line". I am not sure whether this is the most common way of describing such a curve. More often the term "dashed line" is used. Consider changing to "dashed". This suggestion is valid for many other figures in the manuscript.

28. Page 16, Section 3.3, lines 28-29, "height of COD":

Semantically, it sounds strange (or even incorrect) to express COD as representing a height. Of course, I understand what you mean but it can actually be misinterpreted. Since you have already defined $z_{COD0.5}$ why not use this terminology here, e.g. "the retrieved cloud height exceeds $z_{COD0.5}$ for optically thin clouds while it is lower than $z_{COD0.5}$ for optically thick clouds".

29. Page 20, Section 4, line 17, "three CIRS datasets?"

It is not obvious what three datasets you mean (not explained in text)! Please clarify.

---

## Referee Comment (RC2) · Anonymous Referee #1 · 6 Aug 2017

Stubenrauch et al. present results for a unified retrieval approach of cloud top height and cloud amount for high spectral resolution infrared sounders called 'Clouds from IR Sounders' (CIRS). Results for the AIRS and IASI instruments are shown and compared to other cloud climatologies assembled as part of the GEWEX cloud assessment. Discussion on ancillary data sets as used for input in the radiative transfer calculations is included, such as the AIRS team T/q retrievals, ERA-Interim T/q profiles, and IASI surface emissivity. The previously developed sounder retrieval methodology, with modifications that resulted in the CIRS retrieval algorithm, is described in detail. Corrections accounting for increases in atmospheric $CO_2$ that has impacts on the interpretation of cloud trends is described and is a part of the CIRS retrieval package. An a posteri-

ori cloud detection process is described and placed in context with the cloud property retrieval approach. The cloud properties are thoroughly compared to the active Cloud-Sat and CALIPSO profilers in terms of cloud detection and cloud height accuracy. The climatological values of CIRS clouds to other A-train and ISCCP data sets are shown and described. Lastly, the paper ends with a few examples of application of the data to hemispheric differences and ENSO variability.

The AIRS and IASI sounders are extremely valuable observational tools for atmospheric sounding, including cloud properties. The authors have described a single approach that is applied to multiple sounders, and that is a valuable contribution for establishing the temporal continuity of cloud properties from these observations. A very thorough effort in comparing the cloud properties to other sensors has been accomplished. The reviewer appreciates the amount of effort over the years that it has taken to provide a multi-sensor cloud property record and the authors should be commended for that.

With that said, the paper is a slog to get through. It is long and overly detailed. There are many instances of over explaining things that could be easily covered with a reference or assuming some prior knowledge on the part of the reader. There are too many plots and subpanels of plots and at times repetitive information is presented. The two applications that are included don't have a lot of focus and the novelty of the results isn't entirely clear as currently written. With the overwhelming (and ultimately unnecessary) detail, it is hard to narrow down to the most salient points to be made and figures (or panels within the figures) that should definitely be retained and definitely should be removed. Given that this paper does need major revisions, it is mostly from the perspective of tightening up the presentation and making it easier on the reader. The material is there and seems mostly complete – it only needs to be tightened up more than its present form.

Following are some specific suggestions and comments on various parts of the manuscript:

[Figure]

Title: the authors should consider a better title that is punchier and emphasizes the great aspects of using sounders for cloud properties (and not have 'weaknesses' in the title)

Abstract: it is pretty long and not very specific. For instance, lines 24-28 has a single long sentence making multiple points about the apparent cloud top/base. Is the correction for co2 really that original and worth advertising in the abstract? On lines 23-24, the 'global cloud amount' is detected clouds, not effective emissivity?

p. 5, lines 20-21: did the authors try (or consider) using a SST data set independent of the IR sounders, say, RTG-SST or the optimal approach using microwave made available at www.remss.com?

p. 5, lines 23-24: 'quite different' is not quantitative and not useful in the context of this discussion. How different were they? Line 25: the IASI and AIRS sounders will not resolve the diurnal cycle but will capture aspects of it. Lines 26-27: if it is of any help, there is a paper that describes cloud type comparisons between AIRS and ECMWF T/q:

Yue, Q. et al. (2013), Cloud-state dependent sampling in AIRS observations based on CloudSat cloud classification, J. Climate, 26, 8357–8377.

p. 6, lines 9-12: the variables should be listed here (e.g., T, q, emissivity, sfc T, etc.)

p. 7, lines 11-12: here is a good example of over explaining. Why should 'for which temperature first increases with height before decreasing' be included? This is technically only true if ascending in the atmosphere. Line 12: 'moved to the inversion layer' is not clear. Is the cloud placed at the base of the inversion? Hopefully not the top because that would be impossible in reality. Line 14: '...about 7 to 15% of the time.'

p. 8, lines 20-23: this statement is unclear. How can 'clear sky' be 'not too cloudy'?

p. 9, line 7: there is a specific QC approach that filters based on a PBest or PGood pressure level. Was this done on a per profile basis? Or were the Level 3 gridded AIRS

[Figure]

Team products used?

p. 9, lines 8-9: there is a paper that describes AIRS surface temperature biases with respect to ship observations:

Dong, S., S. T. Gille, J. Sprintall, and E. J. Fetzer (2010), Assessing the potential of the Atmospheric Infrared Sounder (AIRS) surface temperature and specific humidity in turbulent heat flux estimates in the Southern Ocean, J. Geophys. Res., 115, C05013, doi:10.1029/2009JC005542.

p. 9, line 11: with respect to what is the land more complex?

p. 10, line 23: is the artifact in cloud amount causing more clouds? Less clouds? Higher clouds? Lower clouds?

p. 11, line 3: base of the inversion?

p. 11, line 25: is 'not cloudy' the same as 'clear'? or something else?

p. 12, line 10: 'explainable' should be 'explained'. The paper could use a good thorough editing for clarity of English.

p. 12, lines 14-16: are there three different sigmas for the three different emissivity_i values? It appears that some of the clear will be selected as cloudy, and vice-versa. Is this correct?

p. 12, lines 20-21: the emis < 0.1 threshold is very conservative. The IR sounders will capture a lot of optically thinner clouds than that. Are the authors arguing the point that below that threshold some clear values could leak in? The paper by Kahn et al. (2008) seems to argue that the emis threshold could be lower than that:

Kahn, B. H. et al. (2008), Cloud-type comparisons of AIRS, CloudSat, and CALIPSO cloud height and amount, Atmos. Chem. Phys., 8, 1231–1248.

p. 12, section 3.1: this is where the paper starts to be a real grind. Wasn't the methodology of the AIRS and C/C comparison described in a previous paper(s) by the lead author? There must be a way to tighten this up and make it more concise, but I am lacking any good suggestions for that.

p. 14, start of Section 3.2: it is really nice to see that the level of agreement is very similar to the AIRS Team cloud retrievals in Kahn et al. (2008) with a finer breakdown of surface type and ancillary data. Is the fact that the percentage is slightly higher over ice/snow indicative of a loss of skill at sounding T/q over these surfaces, and Era-Interim is superior? What is different about these profiles over ice/snow? Better detection of inversions and isothermal layers in ERA-Interim?

p. 14-16: section 3.3: this section is extremely long and detailed. A lot of it seems consistent with previous paper by the first author. Around lines 31-32 on p. 15 there is one quite interesting point about opaque clouds and a reduced geometrical thickness. Could this be because the IWC is larger in these clouds and thus leads to a smaller difference between the sounders and CALIOP? This reminds me of a paper by Sherwood et al. discussing these types of discrepancies:

Sherwood, S. C., J.-H. Chae, P. Minnis, and M. McGill (2004), Underestimation of deep convective cloud tops by thermal imagery, Geophys. Res. Lett., 31, L11102, doi:10.1029/2004GL019699.

p. 17-21, Section 4: another really long section with figures 8-14 that have a combined total of over 80 sub-panels. A lot of these figures are known from previous papers or are common knowledge. Some of these panels appear to show some redundant information. I would suggest trying to trim this down as much as possible and try and keep the information to the most interesting and novel bits.

p. 19, lines 25-27: I don't see why 'which might have important consequences on radiative feedbacks' should be there. Since the SW and LW budgets are not shown with respect to the different cloud types described in the paper, this is speculative. I would further emphasize that there are many other interesting things about these

particular clouds, including the hydrological cycle, not just radiation and its feedbacks.

p. 20, lines 27-28: Are the authors suggesting that the global cloud amount should be related to the global surface temperature? Is there a previous reference that argues for this? Most studies show a relationship of the patterns of global cloud distributions, height, types, etc. can change with respect to global averaged surface temperature, but I've never seen an argument for an average global cloud amount. Also, another point here regarding surface temperature that it did not increase much. If the authors are referring to the alleged 'hiatus', I think that is basically proven that there was no hiatus (a recent paper by T. Karl at NOAA).

http://science.sciencemag.org/content/348/6242/1469

p. 21, lines 28-29: what is the justification to relate infrared derived cloud amount to SW reflected radiation? Are there any previous papers that have shown a correlation? The infrared derived cloud amount saturates around an optical depth of 5 or so, but the SW does not. How can the infrared derived cloud products be used to infer consistency with SW results?

p. 22, lines 1-3: how can the CAH be used as a proxy for precipitation rate? Because the ITCZ is narrower in the CIRS data, one can infer a more intense precipitation rate? I'm not sure I understand the logic used here.

p. 22, first paragraph of Section 5.2: there is no reason to have a basic tutorial on ENSO in the paper. The authors should just get to the results and describe what is novel and delete that part.

Figure 3: numbers are too small and blurry for reading

Figure 4: why bother with the right column? Weren't these differences previously described by the lead author?

Figure 6: three figures in a row describing apparent cloud top and biases with CALIOP. Need to emphasize the novel results and parts of figures that support them. The

numbers are overlapping on the x-axis at the edges of the subpanels too.

Figure 13: can't tell the difference between open and closed red circle, red square, and red dashed line

Figure 14: the seasonal variability in latitude bands is well understood. What is new in this figure? Are there new insights between different instruments and inferences of the seasonal cycle?

---

## Author Response (AR1)

**Response to reviews**

We want to thank the referees for appreciating our work and for the thoughtful comments and suggestions. Most of them have been taken into account to improve the manuscript. We apologize for the difficulties associated with the length of the manuscript and excessively long sentences. We have re-worked the manuscript and addressed each comment. In the text below, the reviewer's comments are marked in italics blue and our answers are given in normal font.

As the referee has correctly pointed out, the method itself is not new (the first author developed it in the 1990's for TOVS), and there exist several publications, which are referenced in the article. Indeed it was a difficult task to select what should be presented and what left out, which is reflected in differing opinions of the 2 referees.

Since both referees suggested to shorten the manuscript, we have done our best to do it without losing the message we wanted to deliver to the community. Here is the list of actions performed

1) shortening section 2 'Data and methods' and moving a shortened version of section 3.1 'Collocated AIRS-CALIPSO-CloudSat data' to this section

2) simplifying Table 2, taking out 5 figures / 22 figure panels (3 figures moved to supplement)

3) taking out the ENSO discussion in section 5 (together with Fig. 16) and

4) revising the remaining applications in section 5

We do not agree with the suggestion of a complete removal of section 5 'Applications', as the presented method is not new and one of the goals of this article was to present scientific applications (as indicated in the title).

Since the results similar to those presented in new Fig. 12 have recently been published for other data sets, it would be difficult to use the presented material in a separate publication. We compare our results to one of them and point out an interesting extension. We plan to work on a more complex analysis to pursue this subject further, but we think it's important to present already these results in the current publication.

**Response to Referee#1**

*Title: the authors should consider a better title that is punchier and emphasizes the great aspects of using sounders for cloud properties (and not have 'weaknesses' in the title)*

the title was changed to : **Cloud climatologies from the InfraRed Sounders AIRS and IASI:**

**Strengths and Applications**

*Abstract: it is pretty long and not very specific. For instance, lines 24-28 has a single long sentence making multiple points about the apparent cloud top/base. Is the correction for co2 really that original and worth advertising in the abstract? On lines 23-24, the 'global cloud amount' is detected clouds, not effective emissivity?*

We have substantially re-worked the abstract, to be more specific.

Rewritten to: The global cloud amount is estimated to 0.67 ó 0.70, for clouds with IR optical depth larger than about 0.1. The spread of 0.03 is associated with ancillary data.

It is really the amount of detected clouds; it is interesting to mention that global effective cloud emissivity of detected clouds is very similar: 0.65-0.66;

This leaves global effective cloud amount (detected clouds weighted by cloud emissivity) to about 0.46-0.48.

*p. 5, lines 20-21: did the authors try (or consider) using a SST data set independent of the IR sounders, say, RTG-SST or the optimal approach using microwave made available at www.remss.com?*

There are two philosophies in creating cloud climatologies : 1) ancillary data are also taken from observations, and 2) ancillary data are taken from model forecast or meteorological reanalyses. The advantage of the first is that these climatologies are independent of model input, however the problem is that the ancillary data might have biases due to faults in clear sky detection and due to interpolation when no good quality data are available.

In this article we compare these approaches; for the first one, we preferred to stay with data which include the same instrument (the ancillary data come from a combined IR sounder ó microwave retrieval). For IASI, at the time of the development, the available ancillary data did not have the quality needed. Therefore we switched to the second approach.

A separate SST data set would not help, as we also need surface temperature over land, and both are needed at the satellite observation times. I addition they also should be coherent with the retrieved atmospheric profiles.

*p. 5, lines 23-24: ÷quite different ø is not quantitative and not useful in the context of this discussion. How different were they?*

Rewritten to: The comparison with collocated temperature profiles of the Analyzed RadioSoundings Archive (ARSA, available at the French data centre AERIS) has shown that, while AIRS-NASA and ERA-Interim (section 2.3) temperature profiles do agree in general with the ARSA profiles within 1 K, differences between IASI-NOAA and ARSA profiles were often larger than 1 K in the lower troposphere (not shown).

In the following plots we present differences in T profiles between NASA AIRS V6 and ARSA, ERA Interim and ARSA and NOAA IASI and ARSA, separately for different latitude bands over land (above) and ocean (below). Whereas AIRS and ERA agree in general within 1K, NOAA IASI differs from ARSA often more than 1 K in the lower troposphere.

[Figure]

*Line 25: the IASI and AIRS sounders will not resolve the diurnal cycle but will capture aspects of it.*

We agree that the diurnal cycle is difficult to resolve with data given in temporal intervals with 4h ó 8h ó 4h ó 8h, but as one can see in the cited conference proceeding (publication is under preparation), by using appropriate analysis techniques, both the amplitude and phase of the diurnal cycle of upper tropospheric clouds can be obtained, especially due to the fact that IR sounders provide unbiased day-night results.

Rewritten to :

This brought us to the conclusion, that ancillary data from the same source are necessary to make use of the AIRS ó IASI synergy for exploring cloud diurnal variability in a coherent way.

*Lines 26-27: if it is of any help, there is a paper that describes cloud type comparisons between AIRS and ECMWF T/q:*

*Yue, Q. et al. (2013), Cloud-state dependent sampling in AIRS observations based on*

*CloudSat cloud classification, J. Climate, 26, 8357ó8377.*

Unfortunately this very interesting article refers to NASA AIRS L2 data of Version 5 ;

We have used in our revised cloud climatology NASA AIRS L2 data of Version 6

*p. 6, lines 9-12: the variables should be listed here (e.g., T, q, emissivity, sfc T, etc.)*

the whole paragraph was taken out, as this issue was already partly discussed in 2.2.

*p. 7, lines 11-12: here is a good example of over explaining. Why should ÷for which temperature first increases with height before decreasingø be included? This is technically*

*only true if ascending in the atmosphere. Line 12: ÷moved to the inversion layerø is not clear. Is the cloud placed at the base of the inversion? Hopefully not the top because that would be impossible in reality. Line 14: ÷…about 7 to 15% of the time.ø*

Text improvements taken into account.

In the case of an inversion, the cloud height is set to the level at which the temperature starts to decrease with height.

*p. 8, lines 20-23: this statement is unclear. How can ÷clear skyø be ÷not too cloudyø?*

text in parentheses taken out; the synergy of IR sounder and microwave also leads to retrievals for party cloudy scenes. In that case, a ÷cloud clearingø is performed before the retrieval.

*p. 9, line 7: there is a specific QC approach that filters based on a PBest or PGood pressure level. Was this done on a per profile basis? Or were the Level 3 gridded AIRS Team products used?*

We used the quality criteria on a per profile basis, as we work with L2 data ; reference added

*p. 9, lines 8-9: there is a paper that describes AIRS surface temperature biases with respect to ship observations:*

*Dong, S., S. T. Gille, J. Sprintall, and E. J. Fetzer (2010), Assessing the potential of the Atmospheric Infrared Sounder (AIRS) surface temperature and specific humidity in turbulent heat flux estimates in the Southern Ocean, J. Geophys. Res., 115, C05013, doi:10.1029/2009JC005542.*

Again, the problem is that this paper refers to AIRS V5 ; we had problems to find published results for the AIRS V6 version, apart from the V6 L2 Performance and Test Report.

*p. 9, line 11: with respect to what is the land more complex?*

We meant that there was not a clear bias found as over ocean ; clarified in the text to :

Since differences over land might be positive or negative (Fig. 2), we left the AIRS-NASA surface temperature ($T_{surf}$) values as they are.

*p. 10, line 23: is the artifact in cloud amount causing more clouds? Less clouds? Higher clouds? Lower clouds?*

Global cloud amount is increasing, when the CO2 increase is not taken into account in the computation of atmospheric spectral transmissivities (new Fig. 10);

When splitting into low-level and high-level cloud amounts, the artefact led to increasing CAL and slightly decreasing CAH.

*p. 11, line 3: base of the inversion?*

In the case of atmospheric temperature inversions, the cloud height is moved to the level at which the temperature starts to decrease with height, and $\varepsilon_{cld}$ is scaled accordingly.

*p. 11, line 25: is ¬not cloudyø the same as ¬clearø? or something else?*

As the IR sounder footprint size is large, it is difficult to distinguish between completely clear sky and cloudy. Even the evaluation with CALIPSO-CloudSat stays approximate as the sampling is only about 1.5 km x 2.5 km, which corresponds to a sampling of about 2 %.

*p. 12, line 10: ¬explainableø should be ¬explainedø The paper could use a good thorough editing for clarity of English.*

Unfortunately, all authors are non-native English speakers; we tried however to improve the readability of the present version to the best of our abilities.

*p. 12, lines 14-16: are there three different sigmas for the three different emissivity_i values? It appears that some of the clear will be selected as cloudy, and vice-versa. Is this correct?*

The thresholds were chosen separately for 1) ocean, 2) land and 3) snow/ice, as the distributions in new Fig S1 (original Fig. 2 moved to the supplement) showed slightly different distributions. Indeed, all methods using thresholds include misidentifications. These are difficult to estimate because of the sampling (2% of CALIPSO-CloudSat per AIRS footprint). The cloud detection includes 80 (over ice) to 92% (over ocean) cases for which CloudSat-lidar GEOPROF and CALIPSO at 5 km resolution (excluding subvisible cirrus) have identified at least one cloud layer, and 30% cases for which the samples did not include a cloud layer. The latter might look at first as a large misidentification of clear sky as cloudy, but the very small coverage of the CloudSat-CALIPSO samples (2%) certainly includes
partly cloudy fields.

Results in section 3 show that by using these thresholds the overall agreement with CloudSat-CALIPSO
is 70% (over ice) to 85% (over ocean), given as hit rates.

*p. 12, lines 20-21: the emis < 0.1 threshold is very conservative. The IR sounders will capture a lot of*
*optically thinner clouds than that. Are the authors arguing the point that below that threshold some clear*
*values could leak in? The paper by Kahn et al. (2008) seems to argue that the emis threshold could be*
*lower than that:*

*Kahn, B. H. et al. (2008), Cloud-type comparisons of AIRS, CloudSat, and CALIPSO*

*cloud height and amount, Atmos. Chem. Phys., 8, 1231ó1248.*

Indeed, the AIRS-LMD climatology (Stubenrauch et al. 2010) went down to an $\varepsilon_{cld}$ of 0.05.

Considering the large footprint and a comparison of $\varepsilon_{cld}$ distributions for cloudy and clear sky CloudSat-
CALIPSO scenes (see below), we decided to exclude scenes with $\varepsilon_{cld} < 0.1$.

We made the sentence more explicit : To reduce misidentification of clear sky as high-level clouds, only
clouds with $\varepsilon_{cld} \times 0.10$ are considered.

Indeed, this came out of a study with CALIPSO-CloudSat :

[Figure]

The above figures present normalized $\varepsilon_{cld}$ distributions of high-level clouds, after multi-spectral cloud
detection, but leaving clouds with $0.05 < \varepsilon_{cld} < 0.10$ as clouds, separately for cloudy scenes defined by
GEOPROF and CALIPSO (full line) and for all scenes (dotted line). The first bin includes scenes with
$0.05 < \varepsilon_{cld} < 0.10$; in the tropics this bin has more clear sky than high-level clouds. Therefore we have
moved the threshold to 0.1. As the contribution of the first bin is small compared to the integral, this
seemed a reasonable choice.

*p. 12, section 3.1: this is where the paper starts to be a real grind. Wasnøt the methodology of the AIRS*
*and C/C comparison described in a previous paper(s) by the lead author? There must be a way to*
*tighten this up and make it more concise, but I am lacking any good suggestions for that.*

Indeed, part of the description of the collocated dataset was already published before, though not the
computation of the cloud height corresponding to a specific optical depth. Referee #2 finds that this
section is not detailed enough.

We have rewritten this section and moved it to section 2.4, hoping that in this way the paper gains clarity.
It also allows the reader who is only interested in the results, directly to go to sections 3-5.

*p. 14, start of Section 3.2: it is really nice to see that the level of agreement is very similar to the AIRS Team cloud retrievals in Kahn et al. (2008) with a finer breakdown of surface type and ancillary data.*

We don∉t completely agree with the statement about the level of agreement with the AIRS cloud data from NASA V5: one important difference is that while the AIRS NASA V5 cloud data agree well for high-level clouds, they have a very large height bias for low-level clouds. This is stated by the Kahn paper: a bias which reaches about 5 km ! Actually, this was the reason for adapting the $\chi^2$ retrieval method to AIRS. Our comparison with the NASA V5 AIRS cloud height was published (Fig. 12) in 2008. Our goal was to build a cloud climatology which is reliable for all clouds. If this is not the case, there will be many cloud type misidentifications. Though the retrieved properties of low-level clouds might be noisier, it was important that their height is not biased, so that they are not confounded with higher level clouds.

Kahn et al. have published a new version of the NASA AIRS cloud climatology, but as unfortunately the team does not yet participate in the GEWEX cloud assessment (though invited), a direct comparison is difficult.

*Is the fact that the percentage is slightly higher over ice/snow indicative of a loss of skill at sounding T/q over these surfaces, and Era-Interim is superior? What is different about these profiles over ice/snow? Better detection of inversions and isothermal layers in ERA-Interim?*

The frequency of retrievals with good quality decreases over ice/snow, probably also because clouds over these surfaces are more difficult to detect. In addition, polar regions might oft be covered by clouds (especially in SH ocean). We show a map of relative frequency of good quality retrievals of $T_{surf}$ for December 2007, at 1:30AM LT (criteria described in 2.5.1). When only 10% of the time during a month, data are available and the meteorological situation is very variable during the month, the interpolation gets to its limits, whereas ERA-Interim data are always available. ERA-Interim also detects twice more inversions than AIRS (though we do not know which of the dataset is closer to the reality).

[Figure]

*Rel. frequency of good quality Tsurf, Dec 2007, 1:30AM*

*p. 14-16: section 3.3: this section is extremely long and detailed. A lot of it seems consistent with previous paper by the first author. Around lines 31-32 on p. 15 there is one quite interesting point about opaque clouds and a reduced geometrical thickness. Could this be because the IWC is larger in these clouds and thus leads to a smaller difference between the sounders and CALIOP?*

*This reminds me of a paper by Sherwood et al. discussing these types of discrepancies:*

*Sherwood, S. C., J.-H. Chae, P. Minnis, and M. McGill (2004), Underestimation of*

*deep convective cloud tops by thermal imagery, Geophys. Res. Lett., 31, L11102,*

*doi:10.1029/2004GL019699.*

We have substantially shortened this section, also by taking out Fig. 6 and taking out 3 panels of Fig 4 and moving 3 panels of Fig 5 to the supplement. We also tried to be more concise. Compared to Stubenrauch et al. 2010, the estimation of the height at which the cloud reaches a COD of 0.5 is new, though one has to keep in mind that it depends on several assumptions (section 2.4). Concerning the slight drop in difference between $z_{cld}$ and $z_{top}$ for $\varepsilon_{cld}$ close to 1, it probably means that for these clouds opacity is reached within a smaller vertical extent, as for those clouds $z_{cld}$ also corresponds to the mean between top and height at which clouds gets opaque. We cited the Sherwood paper in Stubenrauch et al. 2010, where we had already shown that $z_{top} - z_{cld}$ increases with $z_{top} - z_{(app\ base)}$, reaching up to 3 km.

*p. 17-21, Section 4: another really long section with figures 8-14 that have a combined total of over 80 sub-panels. A lot of these figures are known from previous papers or are common knowledge. Some of these panels appear to show some redundant information. I would suggest trying to trim this down as much as possible and try and keep the information to the most interesting and novel bits.*

We took out 16 panels of Fig. 10-13 and 6 panels of Fig. 14, which we also moved to the supplement. This leaves 5 Figs, and we shortened the discussion. On the other hand, we want to show the quality of the new climatologies, so we have to show some comparisons, even if they might not be novel.

*p. 19, lines 25-27: I don₀t see why ₃which might have important consequences on radiative feedbacks₀ should be there. Since the SW and LW budgets are not shown with respect to the different cloud types described in the paper, this is speculative. I would further emphasize that there are many other interesting things about these particular clouds, including the hydrological cycle, not just radiation and its feedbacks.*

We agree with this suggestion so we took this part out and shortened the sentences to:

The independent use of $p_{cld}$ and $\varepsilon_{cld}$ made it possible to build a climatology of upper tropospheric cloud systems, using $\varepsilon_{cld}$ to distinguish convective core, cirrus anvil and thin cirrus of these systems. These data have revealed for the first time that the $\varepsilon_{cld}$ structure of tropical anvils is related to the convective depth (Protopapadaki *et al.*, 2017).

*p. 20, lines 27-28: Are the authors suggesting that the global cloud amount should be related to the global surface temperature? Is there a previous reference that argues for this? Most studies show a relationship of the patterns of global cloud distributions, height, types, etc. can change with respect to global averaged surface temperature, but I₀ve never seen an argument for an average global cloud amount. Also, another point here regarding surface temperature that it did not increase much. If the authors are referring to the alleged ₃hiatus₀, I think that is basically proven that there was no hiatus (a recent paper by T. Karl at NOAA).*

*http://science.sciencemag.org/content/348/6242/1469*

Thank you for the interesting article. We just wanted to make the point that global cloud amount stays stable during this period; we have removed the sentence about surface temperature.

*p. 21, lines 28-29: what is the justification to relate infrared derived cloud amount to SW reflected radiation? Are there any previous papers that have shown a correlation? The infrared derived cloud amount saturates around an optical depth of 5 or so, but the SW does not. How can the infrared derived cloud products be used to infer consistency with SW results?*

We talk here about total CA, which we have shown in section 4 to be consistent with all other climatologies. Also CAH, CAM and CAL are reliably identified, as all discussions in section 4 have shown ! Indeed the effective cloud emissivity saturates at 1 (corresponding to visible COD of about 10), while VIS COD continues to increase. However, the paper of Stephens et al. 2015 is relating the planetary albedo to cloud amount.

*p. 22, lines 1-3: how can the CAH be used as a proxy for precipitation rate? Because the ITCZ is*
*narrower in the CIRS data, one can infer a more intense precipitation rate? I∅m not sure I understand*
*the logic used here.*

We understand the InterTropical Convergence Zone as the zone with strong convection which then
produces large cirrus anvils. The latter stay longer in the atmosphere than the convective towers
themselves. It is also seen in all maps that the ITCZ has a strong occurrence of high-level clouds (which
are mostly cirrus anvils, see for example (Protopapadaki *et al.* 2017)). Hence, we assume that the ITCZ
can be determined by the latitude with a peak in CAH (new Fig. 8). We have partly rewritten this section
and hope that the motivation and analysis are easier to follow.

*p. 22, first paragraph of Section 5.2: there is no reason to have a basic tutorial on ENSO in the paper.*
*The authors should just get to the results and describe what is novel and delete that part.*

we have taken out the introduction and Figure 16 and its discussion.

*Figure 3: numbers are too small and blurry for reading*

fixed in new Figure 2

*Figure 4: why bother with the right column? Weren∅t these differences previously described*

*by the lead author?*

Right column taken out

*Figure 6: three figures in a row describing apparent cloud top and biases with CALIOP. Need to*
*emphasize the novel results and parts of figures that support them. The numbers are overlapping on the*
*x-axis at the edges of the subpanels too.*

Figure taken out and added quartiles to Fig 4, so that the width of the distributions are shown together
with the medians; this makes the discussion more concise

*Figure 13: can∅t tell the difference between open and closed red circle, red square, and red dashed line*

fixed in new Figure 10

*Figure 14: the seasonal variability in latitude bands is well understood. What is new in this figure? Are*
*there new insights between different instruments and inferences of the seasonal cycle?*

Panels with CAM taken out and Figure moved to supplement (new Figure S4); there is nothing new, it is just to show the quality of the new cloud climatologies, compared to other datasets.

**Response to Referee #2**

*5 particular issues that need further explanation:*

*- the role of CALIPSO-CALIOP data for tuning the method*

The cloud property retrieval was originally developed for TOVS data (Stubenrauch et al. 1996, 1999,
2006); at that time the cloud detection, which indeed was applied before the cloud retrieval, was
essentially based on interchannel regression tests using a combination of IR sounder and microwave
(MSU) brightness temperatures.

When we adapted the cloud retrieval to AIRS, channel 7 of AMSU did not work, so we could not adapt
the cloud detection. However the retrieval itself provides cloud pressure and emissivity for each
measurement (only about 5% of the data do not give a solution, these are declared immedeately as clear
sky). We then considered it more interesting to develop a cloud detection which could be applied after
the retrieval. The idea was to test the reliability of the results to decide if a footprint is cloudy. By
comparing clear sky and cloudy scenes determined within time synchroneous samples from CALIPSO
L2 5km cloud data, provided by NASA, we found that the relative spectral spread of cloud emissivities
determined at atmospheric window wavelengths is small if the footprint contains a cloud for which the
cloud height and emissivity are well determined (both are used in the computation of the spectral
emissivities), while most clear sky scenes lead to very large values. These distributions have been
published in Stubenrauch et al. 2010, and for the retrievals with new ancillary data in Fig. S1. These
distributions show a nice distinction between clear and cloudy, but the thresholds themselves have been
determined by examining many different aspects, like maps and comparison with other datatsets,
distributions separately over tropics, midlatitudes and polar regions. One important aspect was also to
test that AIRS, using two different ancillary data sets, together with IASI gave coherent answers, day and
night.

**So, the CALIPSO-CloudSat data have been essential to guide us in the cloud detection, but they were**
**not used to tune it.**

*- the exact description of the used CALIPSO dataset for tuning and for evaluation of cloud properties*

Again we want to stress that we did not use CALIPSO for tuning.

We have moved the section of the collocated AIRS-CALIPSO-CloudSat data forward, so that the
description is placed before the description of the cloud detection. It was well written that we used
version 3 of the NASA CALIPSO L2 cloud data averaged over 5 km (Winker et al. 2009) ; and we
explained the procedures how we used the data (for example excluding subvisible cirrus). By the way,
we published comparisons with lidar already in 2005, when we compared TOVS Path B cloud
properties with LITE (Stubenrauch et al. 2005) where we also investigated subvisible cirrus. In this paper
we just wanted to show that the CIRS cloud data are of slightly better quality than the AIRS-LMD cloud
climatology, and the effect of ancillary data, which in our opinion has not been stressed with other cloud
climatologies.

*- the consequence of using some unphysical assumptions in the retrieval*

We accept cloud emissivities up to a value of 1.5, due to noise. This is explained in the reference
Stubenrauch *et al.* 1999, which is cited :

As in Eq 2 the denominator includes two terms (Icld and Iclr) which get very close to each other in the
case of low-level clouds, the cloud emissivity can get larger than 1 when taking into account
uncertainties. In Stubenrauch et al. (1999), it was shown that the original method, which excluded values
larger than 1, underestimated the amount of low-level clouds considerably.

The limit larger than 1 has been chosen to compensate for radiation noise and ancillary data uncertainties
and this leads to a better identification of low-level clouds.

*- the balance between finding spectral coherence in the solutions and still maintain physically*
*reasonable emissivity differences*

The multi-spectral cloud detection is indeed based on wavelengths in an interval which is sensitive to
thermodynamical phase and ice crystal sizes. As can be seen in Fig. 3 of Guignard et al. (2012), the
relative cloud emissivity difference between 9 $\mu$m and 12 $\mu$m can go up to 0.3 for small IWP and ice
crystal size. However, instead of using a spectral difference, we use a standard deviation between 6
wavelengths, divided by retrieved cloud emissivity. This should be always smaller than 0.15, even in the
case of small IWP and ice crystal sizes which produce the largest slope (we have studied that in detail
when developing the method in 2010). In this empirical method, the error one makes, if the used cloud
pressure does not correspond to the real pressure, is larger, and Fig. S1 (of the supplement) illustrates
nicely, that this relative standard deviation is larger than 0.3 for clear sky scenes, while for cloudy scenes
distributions the distributions are really narrow, using CALIPSO-GEOPROF to separate cloudy and
clear sky scenes.

*- justification of the statement of achieving successful cloud detection down to IR cloud optical*
*thicknesses of 0.1*

optical thickness can be deduced from cloud emissivity as $COD = -\ln(1-\varepsilon_{cld})$

As we present clouds with $\varepsilon_{cld} > 0.1$, this corresponds to clouds with IR COD > 0.1 (or with VIS COD >
0.2 as VIS COD $= -2\ln(1-\varepsilon_{cld})$).

To reduce misidentification of clear sky as high-level clouds, only clouds with $\varepsilon_{cld} \times 0.10$ are considered.

Indeed, this came out of a study with CALIPSO-CloudSat :

[Figure]

The above figures present normalized $\varepsilon_{cld}$ distributions of high-level clouds, after multi-spectral cloud
detection, but leaving clouds with $0.05 < \varepsilon_{cld} < 0.10$ as clouds, separately for cloudy scenes defined by
GEOPROF and CALIPSO (full line) and for all scenes (dotted line). The first bin includes scenes with
$0.05 < \varepsilon_{cld} < 0.10$; in the tropics this bin has more clear sky than high-level clouds. Therefore we have
moved the threshold to 0.1. As the contribution of the first bin is small compared to the integral, this
seemed a reasonable choice.

***Specific comments***

*1. Page 1, Abstract, line 19, õto evaluateö:*

*The term õto evaluateö should be changed to õto design and evaluateö. You used A-train data to find*
*your -a posterioriøcloud masking thresholds, right? Then you should be clear in your description that A-*
*train data is not completely independent from your data/method. This is important for the reader to*
*know.*

We do not quite agree with this comment; the cloud retrieval was originally developed for TOVS
data (Stubenrauch et al. 1996, 1999, 2006); at that time the cloud detection, which indeed was applied
before the cloud retrieval, was essentially based on interchannel regression tests using a combination of
IR sounder and microwave (MSU) brightness temperatures.

When we adapted the cloud retrieval to AIRS, channel 7 of AMSU did not work, so we could not adapt
the cloud detection. However the retrieval itself provides cloud pressure and emissivity for each
measurement (only about 5% of the data do not give a solution, these are declared immedeately as clear
sky). We then considered it more interesting to develop a cloud detection which could be applied after
the retrieval. The idea was to test the reliability of the results to decide if a footprint is cloudy. By
comparing clear sky and cloudy scenes determined within time synchroneous samples from CALIPSO
L2 5km cloud data, provided by NASA, we found that the relative spectral spread of cloud emissivities
determined at atmospheric window wavelengths is small if the footprint contains a cloud for which the
cloud height and emissivity are well determined (as both are used in the computation), while most clear
sky scenes lead to very large values. These distributions have been published in Stubenrauch et al. 2010,
and for the retrievals with new ancillary data in Fig. S1. These distributions show a nice distinction
between clear and cloudy, but the thresholds themselves have been determined by examining many
different aspects, like maps and comparison with other datatsets, distributions separately over tropics,
midlatitudes and polar regions. One important aspect was also to test that AIRS, using two different
ancillary data sets, together with IASI gave coherent answers, day and night.

**So, the CALIPSO-CloudSat data have been essential to guide us in the cloud detection, but they were**
**not used to tune it.**

*2. Page 1, Abstract, line 23, õcoincidesö:*

*To use the term õcoincidesö here is a too strong conclusion from your results. Figure 6 (lower right*
*panel) clearly shows a rather broad distribution of results where frequencies at the two extremes (0 and*
*1) are still about 20-25 % of the frequency for the value 0.5 (representing the middle of the defined*
*layer).Therefore you can possibly only state that the cloud height can be õapproximatedö by the middle*
*of the defined layer. Also õmiddleö could possibly be replaced by õthe mean layer heightö to make the*
*description scientifically stricter.*

*3. Page 1, Abstract, line 27, õapparent vertical cloud extentö:*

*The explanation here is confusing, indicating that upper level clouds generally have higher cloud*
*emissivities than lower level clouds. This cannot be true. I guess the authors mean something else.*
*Please clarify!*

Rewritten as :

CIRS cloud height can be approximated by the mean layer height (for optically thin clouds) or the mean
between cloud top and the height at which the cloud reaches opacity. For high-level clouds, especially in
the tropics, this height lies on average 1 km to 3 km below cloud top.

*4. Page 2, Abstract, lines 5-8, õresponse to climate changeö + Page 3, Section 1, lines 23- 25 and the*
*entire section 5: The last sentence in the abstract, the sentence about Section 5 in Section 1 and the entire*
*section 5 could possibly be removed for shortening the paper (see also comment 25!).*

We have considerable shortened section 5, but have left two main studies, which have been described in
a more concise manner. The latter study is also compared to recent results using other data.

Changed last part of abstract to :

The 5% annual mean excess in high-level cloud amount in the Northern compared to the Southern
hemisphere has a pronounced seasonal cycle with a maximum of 25% in boreal summer, in accordance
with the moving of the ITCZ peak latitude, with annual mean of 4°N, to a maximum of 12°N. This
suggests that this excess is mainly determined by the position of the ITCZ. Considering interannual
variability, tropical cirrus are more frequent relative to all clouds when the global (or tropical) mean
surface gets warmer. Changes in relative amount of tropical high opaque and thin cirrus with respect to
mean surface temperature show different geographical patterns, suggesting that their response to climate
change might differ.

*5. Page 2, Section 1, line 11, õ70 % cloud coverö:*

*Although this is a widely used and accepted figure for global cloudiness, I would like to point out that a*
*value of global cloud cover cannot be stated without first defining what you mean by a cloud. The figure*
*70 % is kind of representing clouds which have a significant impact on radiation budgets and it could*
*possibly be relevant if you define that clouds should have at least a cloud optical thickness of*
*approximately 0.2. But if including also the thinnest clouds (often called sub-visible clouds and so far*
*only observed by high sensitive instruments like CALIPSO-CALIOP) the figure may increase to values*
*well above 80 %. I think it would be appropriate to at least make a short statement on what clouds are*
*considered when stating that global cloudiness is about 70 %.*

Indeed, in the GEWEX Cloud Assessment we found out that global cloud amount is about 0.68±0.03
when considering clouds with VIS optical depth of larger than 0.2, and additional 0.06 arise from
subvisible clouds detected by CALIPSO (Stubenrauch et al. 2013), which brings it to 0.74. This is
written in Section 4.

It seems for us appropriate to leave the about 70%, as this sentence is the first  in the introduction and is
just meant to bring up the importance of clouds because of their large coverage. 7 lines further the reader
finds more detail on the threshold (IR optical depth > 0.1).

*6. Page 3, Section 1, line 3: õoptical depth less than 3ö*

*My impression is that the capability is better than that, i.e., the capability of having reasonable cloud*
*optical depth estimations from CALIOP data covers the interval 0-5. Please check that the value of 3 is*
*really justified.*

The optical depth at which clouds are opaque is difficult to determine. In an earlier publication (Lamquin
*et al.* 2008), we wrote that the upper limit lies between 3 and 5. One should not forget that the uncertainty
is easily 20% due to uncertainty in multiple scattering contributions (Lamquin *et al.* 2008).

We have rewritten this in accordance :

Whereas the lidar can detect sub-visible cirrus, its beam can only penetrate the cloud down to optical
depth of about 3 to 5 (in visible range). For optically thicker clouds, the radar provides the cloud base.

*7. Page 7, Section 2.4, line 4, õemissivities larger than 1ö:*

*I must say that it is quite disturbing to õbe forcedö to use unphysical values in the retrieval. I understand*
*that uncertainties can lead to this but I am not sure that this is then the best way of handling these*

*uncertainties. Why not restrict emissivities to 1 in the optimization/minimization process when knowing that this is physically correct? I can¢t see why your present method gives better uncertainty descriptions of the retrieved cloud pressures than when using a restricted emissivity value. Don¢t inconsistencies give rise to new inconsistencies? Please explain and motivate.*

The reason is explained in the reference Stubenrauch *et al.* 1999 which is cited:

As in Eq 2 the denominator includes two terms (Icld and Iclr) which get very close to each other in the case of low-level clouds, the cloud emissivity can easily get unphysical when taking into account uncertainties. In Stubenrauch et al. (1999), it was shown that the original method, which excluded values larger than 1, underestimated the amount of low-level clouds considerably.

The limit larger than 1 has been chosen to compensate for radiation noise and ancillary data uncertainties and this leads then to a better identification of low-level clouds.

*8. Page 7, Section 2.4, lines 22-28, õa posteriori cloud detectionö:*

*The õa posteriori cloud detectionö has already been briefly introduced (page 4, lines 7- 11). Why repeating this information here? Delete these lines or move part of this to the relevant section 2.5.*

deleted

*9. Page 9, Section 2.4.1, lines 18-20, õocean cloud amounts larger during nightö:*

*To find larger ocean cloud amounts at night than during day is found in many regions (e.g. over marine stratocumulus areas). What made you think this was a problem specifically for ERA-Interim? Please explain.*

The problem is not that the cloud amount is larger during night than during day, but that results are different when using two different sets of ancillary data ; we had to find out which dataset had a problem, and after some time we found that the amplitude of the ERA-Interim SST diurnal cycle is not in agreement with observations. It is reassuring that after applying a correction, this had a positive effect on the cloud amounts, as now the diurnal variation of cloud amount is more similar.

Rewritten to : Without this correction, the cloud amount (CA) at night / early afternoon was 78% / 71%, compared to 71% / 71% when using AIRS ancillary data. The correction led to 76% / 73%, closer to the results using AIRS ancillary data.

*10. Page 10, Section 2.4.2:*

*The CO2 correction appears to be a very relevant change (also visualized nicely in Figure 13. This appears to be one of the most important improvements of the methodology. Should become mandatory in all sounding-based retrievals for climate datasets, in my opinion.*

Thank you for the compliment ☺ In our case this was necessary, as the spectral transmissivities came from look-up tables computed for a fixed $CO_2$ concentration.

Actually, Menzel *et al.* (2016) also use a varying $CO_2$ concentration adjustment, for a 35-year HIRS cloud climatology.

*11. Page 11, Section 2.5, general comment on the õa posteriori cloud detectionö:*

*The methodology appears a bit awkward compared to many other cloud retrieval methods in that cloud properties are first derived and then a determination whether a FOV is cloudy is carried out as a second*

 *Most common otherwise is that a cloud screening is done first and then followed by a cloud property retrieval. So, could you confirm that after having performed the cloud property retrieval, all FOVs are still assumed to be cloudy? Does it mean that you will always find a solution to Equation 2? You have already mentioned some problems in finding a distinct minimum for lowlevel clouds (page 7, lines 2-3) but what happens in obviously cloud-free situations?*

Actually, we see this method as an advantage, because the method tests if the retrieved values are coherent, whereas most cloud detection methods use many different threshold tests, mostly based on brightness temperatures. We would have liked to adapt the cloud detection which was based on the comparison of temperatures (after correction for water vapour effects) obtained from HIRS to those of the microwave sounding unit MSU (developed for TOVS) to AIRS. Unfortunately, the AMSU channel which sounded closest to the surface did not work from the beginning. Therefore we have developed this method. Indeed, the $\chi^2$ method provides in most cases (95%) a solution. The cloud detection is based on the coherence of spectral emissivities which are calculated using the retrieved cloud pressure. If the retrieved cloud pressure does not correspond to reality (as for clear sky or partly cloudy situations), the spectral variability gets large, as illustrated in Fig. S1.

We have now moved section 2.5 to section 2.4.3 and have rewritten part of the text.

*12. Page 11, Section 2.5, line 16 + lines 20-21, õmeaning of spectral coherenceö:*

*I am a bit concerned about the concept indicating that, for a cloud to be identified, the differences between emissivities in the six infrared channels should be small. In this wavelength region we know that the refractive indices of water and ice, respectively, varies considerably. For example, this is one of the fundamental properties that allows separating water clouds from ice clouds in passive imagery (e.g. as introduced by Pavolonis et al., 2005, J. Appl. Meteorol.). This fact would also certainly introduce considerable differences in cloud emissivities depending on if it is a water or ice cloud in addition to variations in optical thickness or partial coverage within each FOV. So, isnøt there a risk that the demand on spectral coherence is in conflict with reality? Or are you able to find a balanced and optimized method based on reference observations from CALIPSO-CALIOP data and still retain reasonable resulting emissivity differences? I guess that the access to CALIPSO-CALIOP data here is essential since it would be difficult otherwise (e.g. through detailed cloud model simulations) to find an optimal way here. Please comment.*

The multi-spectral cloud detection is indeed based on wavelengths in an interval which is sensitiv to thermodynamical phase and ice crystal sizes. As can be seen in Fig. 3 of Guignard et al. (2012), the relative cloud emissivity difference between 9 µm and 12 µm can go up to 0.3 for small IWP and ice crystal size. However, instead of using a spectral difference, we use a standard deviation between 6 wavelengths, divided by retrieved cloud emissivity. This should be always smaller than 0.15, even in the case of small IWP and ice crystal sizes which produce the largest slope (we have studied that in detail when developing the method in 2010). In this empirical method, the error one makes, if the used cloud pressure does not correspond to the real pressure, is larger, and Fig. S1 (of the supplement) illustrates nicely, that this relative standard deviation is larger than 0.3 for clear sky scenes, while for cloudy scenes distributions the distributions are really narrow, using CALIPSO-GEOPROF to separate cloudy and clear sky scenes.

*13. Page 11, Section 2.5, line 25, õstandard deviationö:*

*How do you calculate the standard deviation here? Do you use all values in the õAIRS golf ballö (i.e., 9 values) for the calculation for each wavelength? The current description is not clear enough on this.*

It is a standard deviation over all 6 emissivities per AIRS footprint.

*14. Page 11, Section 2.5, line 27, õCALIPSO samplesö:*

*Unfortunately, here you introduce the use of CALIPSO data without having described what data you actually used (this description comes later in Section 3.1). More clearly, it is not obvious to the reader that you will get three CALIPSO samples in the AIRS golf ball. For this, you need to know that you use 5 km CALIPSO data. Because of the importance of A-train data for your method and study, I am of the opinion that you should have introduced them already in Section 2 on õData and Methodsö. Can you consider changing this?*

Section 3.1 now moved to section 2.4

*15. Page 12, Section 2.5, lines 18-19, õminimum optical depthö:*

*In the introduction section you mention that with IR vertical sounding data õreliable detection of cirrus with IR optical depths as low as 0.1ö is possible indicating that this is much better than what can be achieved from other sensors (except from active sensors). I wonder what this restriction in order õto reduce noiseö means in this context? Have you estimated further the minimum cloud optical depths being detected after introducing this restriction? CALIPSO-CALIOP offers the possibility to do such in-depth studies.*

We made this sentence more explicit : To reduce misidentification of clear sky as high-level clouds, only clouds with $\varepsilon_{cld} \times 0.10$ are considered.

Indeed, this came out of a study with CALIPSO-CloudSat :

[Figure]

The above figures present normalized $\varepsilon_{cld}$ distributions of high-level clouds, after multi-spectral cloud detection, but leaving clouds with $0.05 < \varepsilon_{cld} < 0.10$ as clouds, separately for cloudy scenes defined by GEOPROF and CALIPSO (full line) and for all scenes (dotted line). The first bin includes scenes with $0.05 < \varepsilon_{cld} < 0.10$; in the tropics this bin has more clear sky than high-level clouds. Therefore we have moved the threshold to 0.1. As the contribution of the first bin is small compared to the integral, this seemed a reasonable choice.

*16. Page 13, Section 3.1, lines 16-19, õCALIPSO and CloudSat dataö:*

*This requirement should mean (?) that you require that both CloudSat and CALIPSO say it is cloudy. But what about the fact that CALIPSO sees much more of the very thin cirrus clouds being available? Does it mean that these cirrus cases are not included in your evaluation study despite the fact that you several times have emphasized the capability of your method to detect very thin cirrus? Or is it different for studies of cloud amount (as indicated by description in lines 7-15) and cloud top height? Please comment!*

We use CloudSat-lidar GEOPROF data, which detect a cloud layer when either CALIPSO or CloudSat detect a cloud layer (footprint 2.5 km x 1.5 km), and to add a different sampling (and because we needed a few other variables like COD) we use the CALIPSO 5km cloud data. In the latter we exclude subvisible cirrus (admitting only clouds detected with horizontal averaging < =5 km) for the evaluation, as we know that IR sounders are not sensitive to those. This corresponds to clouds with COD > 0.05 to 0.1, according to Winker *et al.* (2008).

Then, we require that both samplings detect a cloud, just to be sure that the sampling is coherent. These data are then used for all studies in this paper. We have tried to explain it better in the new section 2.4 :

í .The CALIPSO cloud data also indicate at which horizontal averaging along the track the cloud was detected (1 km, 5 km or 20 km), which is a measure of the COD. As in Stubenrauch et al. (2010), for a direct comparison with AIRS cloud data, we use clouds detected at **horizontal averaging over 5 km or less**. **This corresponds to clouds with visible COD larger than about 0.05 to 0.1** (Winker *et al.*, 2008). The scene type of an AIRS footprint is estimated as cloudy when the CALIPSO sample as well as the GEOPROF sample include at least one cloud layer. Clear sky is defined by cloud-free CALIPSO and GEOPROF samples within the AIRS footprint.

*17. Page 13, Section 3.1, line 23, õunderestimated CODö:*

*Just for your information: The latest version of the CALIPSO-CALIOP dataset (version 4.1) gives indeed higher CODs. This change can possibly be connected to what you write here (currently I do not know the details behind this change).*

Thanks for this information!

*18. Page 14, Section 3.2, lines 2-3, õagreementö:*

*I have to ask you to specify better what you mean by õagreementö. There are so many skill scores around so youdd better be strict in describing exactly the measure you use. I guess you refer to what is normally called õHit Rateö which is the number of correct cloudy AND clear cases divided by the total number of cases.*

Indeed, it is the hit rate which we have calculated. We have changed this in the text :

The hit rates between the ­a posterioriø cloud detection and the CALIPSO-CloudSat cloud detection are 85% (84%) over ocean, 82% (79%) over land and 70% (73%) over ice / snow.

*19. Page 14, Section 3.3, generally on results in Figure 4 (Page 40):*

*First, please revise the wording of the caption of this figure. The first sentence here is too complicated and the description should possibly be made more clear (the same is actually true for Figure 5). Also make clear (in all figures) what you mean by õ1:30 LTö (AM or PM??). The question raised in the previous comment 16 remains: Are thin cirrus detected by CALIPSO but not by CloudSat part of this study or not?*

*If not, what can be said about the quality of these retrieved cloud heights (as compared to CALIPSO data alone)?*

:30 is 1 :30AM, as defined in section 2.1 (1 :30 and 13 :30) ; however, as this leads to confusion with American readers, we will change this in the whole paper to 1 :30AM and 1 :30PM etcí

As explained before, for this comparison CALIPSO cloud data with COD > 0.05 to 0.1 are used.

The other referee suggested to take out the right panels of Figure 4 (which look very similar to the results
published in Subenrauch et al. 2010). We have worked on all figure captions ;

Compared to the publication of Kahn et al. 2008 about the NASA AIRS Science team results of cloud
height from Version 5, we show that in both cases, high-level clouds as well as mid- and low-level
clouds the height is determined without bias, if one consideres the cloud height given by AIRS as the
height of maximum lidar backscatter (Stubenrauch *et al.*, 2010), by the mean layer height (for optically
thin clouds) or the mean between cloud top and the height at which the cloud reaches opacity, as shown
in Figure S2 (considering mid-$p_{cld}$), or by $z_{COD0.5}$ (Figure 3).

*21. Page 16, Section 3.3, lines 5-24, Figure 7:*

*Very interesting and impressive results shown here! Results for medium and high clouds are probably*
*quite superior to those being presented from passive imagery in other CDRs. Only for low-level clouds*
*we still see quite some discrepancies which is understandable for several reasons. This indicates that the*
*best representation of the true vertical distribution of cloudiness in a climate sense could be a*
*combination of sounding and passive imagery data. Do you agree? Maybe you should mention this.*
*Interesting is that problems for low clouds for sounding applications is not showing up very clearly later*
*in Figure 9, except possibly during night for the land-ocean difference. Maybe you should explain why?*

Indeed, a combination of IR sounder and passive imagery would increase the quality during day. During
night, sounding provides better results, though the large footprints are a handicap for the identification of
low-level cloud fields (as shown in the analysis of new Fig. 5). The concept of the CIRS retrieval was
guided by the goal to create a cloud climatology with small biases, also for low-level clouds. Indeed, the
noise is much larger for low-level clouds than for high-level clouds, but the biases are small compared to
other IR sounder cloud climatologies. The comparison with CALIPSO-CloudSat comes to its limit in
the analysis of new Fig 5, as the size of the footprints is very different.

*20. Page 15, Section 3.3, line 9, õcoincidesö:*

*See previous comment 2.*

*22. Page 16, Section 3.3, line 32, õcoincidesö:*

*See previous comment 2.*

*26. Page 26, Section 6, line 1, õcoincidesö:*

*See previous comment 2.*

Replaced by ÷can be approximatedø

*23. Page 18, Section 4, lines 15-16, õsensitivity of lidarsö:*

*You write that õactive lidar is the most sensitiveö. Quite true but you havenøt explained whether*
*CALIPSO results in Figure 9 are already õfilteredö (so that the thinnest clouds as given by the original*
*CALIOP CLAY product are removed) or not. Has there been any filtering of ÷sub-visible cloudsø (I*
*assume there has)? This is a relevant question to ask also for the statement in the Conclusions section on*
*page 25, line 25. We need to know exactly what is the used CALIPSO dataset used as reference!*

In section 4, the CALIPSO L3 data of the GEWEX Cloud Assessment data base are used ; two teams
have provided their data, with the main difference by vertical (CALIPSO-GOCCP) or horizontal
averaging (CALIPSO-ST), as mentioned in the text. The details of the GEWEX Cloud Assessment data
base are found in (Stubenrauch et al. 2013) and especially in the WCRP report (Stubenrauch et al. 2012), where each team gave details how they created the L3 data. As I remember, CALIPSO-ST includes subvisible cirrus, which explains the larger CA, compared to all other datasets.

In section 3, L2 products have been used, as descibed in the new section 2.4.

*24. Page 21, Section 4, line 4, Figure 14, "Seasonal cycle of cloud temperatures":*

*How come there is a rather large consensus between different methods when studying cloud temperatures for the polar areas (leftmost and rightmost columns) when the spread is very large when it comes to cloud amount (top row of the same columns)? I suspect it is an indication of that cloud temperatures and surface temperatures are very similar here. This implies (in my opinion) that the separation of cloudy and cloud-free areas is indeed not very accurate. So, where is really the truth as regards polar cloudiness? Apart from this reflection, I consider Figure 14 as a very nice compilation of global cloudiness and its variation.*

This actually shows that cloud amount, depending on thesholds, might be different by 10%, while the averages of retrieved cloud properties, which only can be given when a cloud is detected, are more similar. (Missing 10% does not mean that the average properties of the clouds are completely different). In addition the polar regions are to be considered with care, as written in the discussions : the CALIPSO data does not conform with the other data sets in the GEWEX Cloud Assessment data base, because they exclude measurements from 1:30PM during polar night (polar winter) and from 1:30AM during polar day (polar summer).

As a similar figure was already published in Stubenrauch *et al.* (2013) (though not CT), we moved this Fig. to the supplement, in order to shorten the paper, and as suggested by referee#1.

*25. Pages 21-24, Section 5, "beyond scope??"*

*In my opinion, Section 5 feels like out of scope of this study. Although introducing highly interesting topics (especially section 5.2), this work would benefit from being presented as a separate (or companion) publication. This manuscript is very, very long and it will put the readers (as it truly has for reviewers!) to a real test when digesting it. I would say that especially section 5.2 on the ENSO effects and its coupling to cloud/radiation feedbacks also requires a different category of expertize for reviewing it with more focus on modelling and studies of climate change and climate feedback effects. Consequently, I have not provided specific comments on this section and I suggest that it is removed for the shortening of this paper.*

We do not agree with the suggestion of a complete removal of section 5 "Applications", as the presented method is not new and one of the goals of this article was to present scientific applications (as indicated in the title).

However, we have considerably shortened the section by removing the introduction on ENSO and the discussion about Fig. 16 as well as Fig. 16 itself.

Since the results similar to those presented in new Fig. 12 have recently been published using other data sets, it would be difficult to use the presented material in a separate publication. We plan to work on a more complex analysis to pursue this subject further, but we think it's important to present these results in the current publication.

*27. Page 24-27, Section 6, general comment:*

*A very comprehensive and good summary of the content of the paper. However, it could be shortened (page 26, lines 14-32) as a consequence of comment 25 above.*

Thank you ! We have revised the part considering section 5.

**Technical corrections**

*1. Page 1, Abstract, line 11-14:*

*The current introductory sentences assumes that the reader already knows about the LMD cloud retrieval scheme. I suggest a slight reformulation to make it less unclear, e.g. like the following*

*õThe Laboratoire de Météorologie Dynamique (LMD) cloud retrieval scheme CIRS (Clouds from IR Sounders) has been adapted to cope with any Infrared (IR) sounding instrument. This has been accomplished by applying improved radiative transfer calculations as well as by introducing an original method accounting for atmospheric spectral transmissivity changes associated with varying CO2 concentrationsö.*

This is not fully correct, as the cloud retrieval developed in the 1990øs did not have the name ÷CIRSø; this name corresponds to the adapted version.

We have rewritten the beginning as:

Global cloud climatologies have been built from 13 years of Atmospheric IR Sounder (AIRS) and 8 years of IR Atmospheric Interferometer (IASI) observations, using an updated Clouds from IR Sounders (CIRS) retrieval. The CIRS software can handle any Infrared (IR) sounder data. Compared to the original retrieval, it uses improved radiative transfer modelling, accounts for atmospheric spectral transmissivity changes associated with $CO_2$ concentration and incorporates the latest ancillary data (atmospheric profiles, surface temperature and emissivities).

*2. Page 2, Abstract, line 3, õ5 % asymmetryö:*

*Please clarify better what you mean with asymmetry. Does it mean that there is generally 5 % more high clouds in the Northern Hemisphere? I assume this is what you mean (supported also by Figure 10) but you should make it crystal clear for the reader in the Abstract!*

Rewritten as :

The 5% annual mean excess in upper tropospheric cloud amount in the Northern compared to the Southern hemisphere has a pronounced seasonal cycle with a maximum of 25% in boreal summer, in accordance with the moving of the ITCZ peak latitude to a maximum of 10°N.

*3. Page 2, Section 1, line 17, õpropertiesö:*

*Do you really mean õpropertiesö? I would rather say õcloud detectionö.*

Yes : we meant here that in addition to identification (which means detection), also their properties (height and emissivity) are well determined (even better than those for low-level clouds)

*4. Page 2, Section 1, line 32, õdetermineö:*

*Like the previous comment, I am not sure about the correct wording here. The word õdetermineö is very strong and almost indicates that the CALIPSO and CloudSat satellites together are creating/defining the clouds. Rather, you should express that they õare capable of observing the cloud vertical structureö.*

Changed according to suggestion

*5. Page 3, Section 1, line 5, õthe cloud retrieval methodö:*

*Be a bit more specific, e.g. write õthe evolution of the original cloud retrieval methodö.*

changed

*6. Page 3, Section 1, line 9, õradiative transferö:*

*I think you should write õradiative transfer calculationsö or õradiative transfer modellingö. To only write õradiative transferö is too general and (I guess) just a shortening of more correct terms.*

changed

*7. Page 3, Section 1, line 11, õinitialö: See 5 above (consider using same notation).*

Changed to original

*8. Page 3, Section 1, line 11, õradiative transferö: See 6 above (consider using same notation).*

changed

*9. Page 4, Section 2.1, line 11, õThe NASA Science teamí .ö:*

*I would recommend to start a new paragraph here to increase the readability.*

done

*10. Page 4, Section 2.1, line 15, õSusskind et al, 2003ö:*

*I see inconsequent reference formulations on several places in the manuscript. When you make a direct reference to other publications directly in the text (like here) you should (according to my experience) preferably write: õThe methodology is essentially unchanged from that described in Susskind et al. (2003).ö You have done this correctly in other places (e.g., Page 5, line 27). I think you should be consistent here. Use the formulation above when specifically discussing a publication and use reference in parenthesis when not making a direct statement of the referred publication (a õsofterö reference). Check also the following references for the same reason:*

*- Page 4, line 27*

*- Page 6, line 5*

Thanks, all changed

*11. Page 4, Section 2.1, line 20, õshortwave window channelsö:*

*Please write õshortwave infrared window channelsö since õshortwaveö most often is reserved to define visible channels.*

changed

*12. Page 4, Section 2.1, line 22, õpartial cloud coverö:*

*A better formulation is probably õunder partially cloudy conditionsö.*

changed

*13. Page 4, Section 2.1, line 24, õsnow or iceö:*

*Maybe a better formulation is õí snow or ice covered surfaces also provided by NASA L2 dataö.*

changed

*14. Page 4, Section 2.1, line 26, õideologyö:*

*I would suggest using the term õconceptö rather than õideologyö.*

changed

*15. Page 4, Section 2.1, line 27, õand allowö:*

*I suggest replacing this with õwhich allowsö.*

Rewritten to : The CIRS cloud retrieval allows cloud levels up to 30 hPa above the tropopause.

*16. Page 5, Section 2.2, line 1, õ12 kmö:*

*Is the 12 km valid for each individual footprint or the 2x2 array?*

For each individual footprint, clarified in text

*17. Page 5, Section 2.2, line 9, õthe cloud retrievalö:*

*You should write õthe CIRS cloud retrievalö.*

changed

*18. Page 5, Section 2.2, lines 9-10, õretrieved atmospheric profilesö:*

*Be more specific. You should write õIASI-retrieved atmospheric profilesö.*

changed

*19. Page 5, Section 2.2, line 15, õThereforeö:*

*You should not start a new paragraph here if you refer directly to what was written in the previous*
*sentences. Make it also very clear that you never (well, not in time for your development) got access to*
*EUMETSAT Version 6 data otherwise this statement appears rather strange.*

We could have gotten access after the development and evaluation of the cloud climatologies were
nearly at the end. Since it would have taken another year to build the ancillary data from this data set and
evaluate again the IASI cloud climatology (also in combination with AIRS), we opted for ERA-Interim
ancillary data to build the combined AIRS-IASI cloud climatologies.

As the sentence about V6 EUMETSAT retrievals seems to cut the flow, we took it out.

*20. Page 5, Section 2.2, line 21, õsame sourceö:*

*I guess you rather mean a "less instrument-dependent source"?*

We think it is more "retrieval quality-dependent source", but this would be difficult to write, as the different Science Teams are doing the best with the fundings they have available. (In the case of NOAA for example, the team had to move working on CrIS).

*21. Page 6, Section 2.3, line 1, "proxy":*

*I don't like the word "proxy" in this context. It indicates that it is a kind of simulation or approximation of the real vertical velocity. The vertical pressure velocity is just another formulation of the vertical velocity which arises when you use pressure as your vertical coordinate instead of the standard geometrical height in meters. So, to my knowledge, it's the "real thing" and not a "proxy".*

*But I guess you refer to the fact that the direct calculation of is difficult without making approximations. The most common here is the geostrophic assumption leading to the so-called " - equation". In this sense, I guess you may be correct in interpreting it as an approximation. But still, present day NWP models are capable of calculating so I just wonder what value you are using here? On the other hand, the approximated value at the 500 hPa level is probably quite accurate anyway (conditions here are largely quasi-geostrophic on the large scale) so perhaps this discussion is less important. Anyway, give it a thought.*

We needed the vertical velocity for the interpretation in the ENSO analysis. Since Fig. 16 and its interpretation is taken out according to the referees suggestion, this sentence is also taken out.

*22. Page 7, Section 2.4, line 12, "arise":*

*Maybe reformulate to "these cases occur in about 7 to 15 % of all cases"?*

Changed to : these cases occur in about 7 to 15 % of all cloudy cases

*23. Page 8, Section 2.4.1, line 14, "less than ..?..":*

*Strange formulation. You'd better write "0.99 for wavelengths less than 10 m and 0.98 for wavelengths larger than 10 m".*

Changed to : the surface emissivity is set to 0.99 for $\lambda_i < 10$ $\mu m$ and 0.98 for $\lambda_i \times 10$ $\mu m$

*24. Page 13, Section 3.1, line 6, "spatial resolution CALIPSO":*

*Shouldn't it be "5 km x 0.3 km"? I thought the basic FOV of CALIOP was 300 meter.*

I have understood that the diameter of the spots is 90m, and the sampling along track is 333 m.

For example : https://calipso.cnes.fr/en/CALIPSO/lidar.htm or Winker *et al.* (2009), p. 2312

*25. Page 15, Section 3.3, Figure 5 (Page 41):*

*I suggest that you try to include some additional explanatory features or legends in the figure (e.g., legend with the three coloured dots explained). To look for all explanations in the caption is not very reader-friendly. Try to speed up the correct interpretation of figures with the use of more graphical legends or marks. This remark is probably valid for many other figures in the manuscript.*

We have taken into account the referee's suggestion and revised all figures accordingly.

*26. Page 15, Section 3.3, line 27, "Considering" :*

*I suggest starting a new paragraph here in order to avoid too long chunks of text (unnecessary tiring for the reader).*

This whole paragraph has been rewritten (as Fig. 6 has been taken out, and Fig. 5 has been rebuilt with medians and interquartiles to show the width of the distributions within the same figure). We hope that it is now much easier to read.

*27. Page 15, Section 3.3, line 28; Figure 6 (Page 42):*

*In the caption you describe one of the curves as "broken line". I am not sure whether this is the most common way of describing such a curve. More often the term "dashed line" is used. Consider changing to "dashed". This suggestion is valid for many other figures in the manuscript.*

Thanks ; changed everywhere ; though dashed lines seems also to exist, at least according to google ;)

*28. Page 16, Section 3.3, lines 28-29, "height of COD":*

*Semantically, it sounds strange (or even incorrect) to express COD as representing a height. Of course, I understand what you mean but it can actually be misinterpreted. Since you have already defined zCOD0.5 why not use this terminology here, e.g. "the retrieved cloud height exceeds zCOD0.5 for optically thin clouds while it is lower than zCOD0.5 for optically thick clouds".*

This is obvious from the figure, but we want to stress the following :

In that case, $z_{cld}$ of thin cirrus should be approximated to a height at which COD reaches a value < 0.5 and $z_{cld}$ of opaque high clouds to a height at which COD reaches a value > 0.5.

*29. Page 20, Section 4, line 17, "three CIRS datasets?"*

*It is not obvious what three datasets you mean (not explained in text)! Please clarify.*

three CIRS climatologies (AIRS, using AIRS-NASA and ERA-Interim ancillary data, as well as IASI, using ERA-Interim ancillary data)

**Abstract**

Global cloud climatologies have been built from 13 years of Atmospheric IR Sounder (AIRS) and 8 years of IR Atmospheric Interferometer (IASI) observations, using an updated Clouds from IR Sounders (CIRS) retrieval. Thee CIRS software can handle any Infrared (IR) sounder data. Compared to the original retrieval, it uses improved radiative transfer modelling, accounts for atmospheric spectral transmissivity changes associated with $CO_2$ concentration and incorporates the latest ancillary data (atmospheric profiles, surface temperature and emissivities). ~~applies improved radiative transfer, as well as an original method accounting for atmospheric spectral transmissivity changes associated with $CO_2$ concentration. The latter is essential when considering long term time series of cloud properties. For the 13 year and 8 year global cloud climatologies of cloud properties from observations of the Atmospheric IR Sounder (AIRS) and of the IR Atmospheric Interferometer (IASI), respectively., we used the,Tonlyable onlyed when not hidden bylatest ancillary data (atmospheric profiles, surface emissivities and atmospheric spectral transmissivities).such as cloud amount and height as well as to explore the vertical structure of different cloud types.~~

ment with CALIPSO-CloudSat is about 84% - 85% over ocean, 79% - 82% over land and 70% - 73% over ice / snow, depending on atmospheric ancillary data. Global cloud amount is estimated to 67% - 70%. CIRS cloud height  can be approximated either by  the mean layer height (for optically thin clouds) or by the mean  between  cloud top and the  cloud base  height at which the cloud reaches opacity, This is valid for high-level as well as for low-level clouds identified by CIRS. This height  lies on average about 1 km  slightly increasing  the apparent vertical cloud extent  IR sounders are  particular advantageous to retriev 
[revised manuscript text omitted]
 a posteriori AIRS-CIRS cloud detection leads to an agreement with and the lidar-radar CALIPSO-CloudSat cloud detection (section 2.4) from GEOPROF and CALIPSO in about are 85% (84%) over ocean, 82% (79%) over land and 70% (73%) over ice / snow. Values in parantheses correspond to using atmospheric and surface ancillary data, deduced from AIRS-NASA (ERA-Interim) ancillary data. Table 1 presents separate these agreements comparisons separately for the three latitude bands. CALIPSO-CloudSat cloud detection is defined by at least one cloud layer from GEOPROF and from CALIPSO and clear sky is defined by three CALIPSO clear sky samples within one golf ball (section 2.4). In general, these agreements hit rates are quite high, considering that CALIPSO and GEOPROF data only sample a small area of the

AIRS footprints. They are slightly higher over ocean than over land. Compared to the AIRS-LMD cloud retrieval presented in (Stubenrauch et al., (2010), the agreement with CALIPSO-CloudSat has improved both over ocean and land, but slightly decreased over sea ice. The latter can be explained by applying now only one test over all surface types. In the earlier version we used an additional brightness temperature difference test related to temperature inversions. A detailed analysis (not shown) indicated that it also introduced noise.

To further illustrate cloud amount (CA) uncertainties due linked to ancillary data, we investigate, in

Figure 2, presents geographical maps of differences in CA differences and $T_{surf}$ between , using AIRS-

CIRS based on ancillary data from AIRS-NASA and from ERA-Interim, together with $T_{surf}$ differences, are shown in Figure 3. When using With AIRS-NASA ancillary data, CA over land is mostly often smaller over land during night and larger over land in the afternoon, with. One might observe a positive correlation with differences in $T_{surf}$: $T_{surf}$ of the ancillary data deduced from AIRS-NASA is slightly also smaller during night and larger in the afternoon during daytime over large parts of the continents. From tTConsidering the $T_{surf}$ comparison with ARSA in (section 2.5),4 leads then to the conclusion, this means we deduced that over land AIRS-CIRS CA is slightly underestimated during night when using with

AIRS-NASA ancillary data, while slightly underestimated in the afternoon when using with ERA-

Interim ancillary data. Patterns of differences in atmospheric water vapour are less reflected in those of

CA (not shown), but slightly more atmospheric water vapour in the ancillary data (as in the tropics for

AIRS-NASA compared to ARSA and ERA-Interim) might lead to a slight underestimation of CA.

**3.23 Cloud height**

For the evaluation of cloud height we determine the lidar CloudSat GEOPROF cloud layer which is closest to $z_{cld}$ from AIRS. From the 5 km averaged CALIPSO data we also determine the height at which the cloud reaches a certain optical depth, in particular 0.5, $z_{COD0.5}$. We then require that this height is located within the corresponding cloud layer of the lidar CloudSat GEOPROF data.

Cloud optical depth (COD) determined from lidar backscatter depends on a correction for multiple scattering which itself depends on COD and microphysics (e. g. Comstock and Sassen, 2001; Chen et al., 2002; Lamquin et al., 2008). As CALIPSO assumes a constant multiple scattering coefficient of 0.6

in the retrieval (Winker, 2003), COD might be slightly underestimated, especially for larger COD. We therefore estimate from Figure 3 in (Lamquin et al., 2008) a correction factor and deduce that a COD of

0.50 should correspond to a COD given by CALIPSO of about 0.37. To determine the height within the cloud at which COD reaches 0.5 we also use an assumption on the shape of the ice water content vertical profile between cloud top and cloud base (Feofilov et al., 2015b).

Figure 34 presents normalized distributions of the difference between from CALIPSO (section 2.4)  and

$z_{cld}$, from AIRS for the three latitude bands

We compare results for $p_{cld}$

< 440 hPa and $p_{cld} \times 440$ hPa, separately for AIRS-NASA and ERA-Interim ancillary data

In general, all distributions  peak around 0 km and are slightly narrower for lower-level clouds than for high-level clouds. Results are similar for both ancillary data, with a slight cloud height overestimation  of lower level clouds  over tropical  ocean for ERA-Interim (not shown). and a height overestimation of some clouds  over polar  ocean  for AIRS-NASA ancillary data (not shown). The latter can be explained by the fact that in some of these regions $T_{surf}$ and atmospheric profiles of good quality are only available in 10% of the time. When comparing distributions of

$z_{top}$ - $z_{cld}$, the peaks for lower clouds are still around 0 km, whereas for high-level clouds $z_{cld}$ lies on average 1.5 km below the cloud top (not shown),  very similar to results in Stubenrauch et al. (2010).

This mean that $T_{cld}$ is about 10 K warmer than the cloud top (Figure S2 of the supplement). The broader distributions for high-level clouds compared to low-level clouds may be explained by the fact that high-level clouds often have diffuse cloud tops (e. g. Liao et al., 1995), especially in the tropics ($z_{top}$ -

$z_{cld}$ is slightly larger for the same $\varepsilon_{cld}$, as shown in Figure 5). To summarize, $z_{cld}$, can be approximated by i) the height of maximum lidar backscatter (Stubenrauch et al., 2010), ii) $z_{COD0.5}$ (Figure 3), or iii) the mean layer height ( for optically thin clouds) or the mean between cloud top and the height at which the cloud reaches opacity), as shown in Figure S2 (considering mid-$p_{cld}$), or ( Figure 3)4.

For a more detailed investigate relates to the height of COD of about 0.5 and to cloud top ($z_{top}$), we analyze~~

Figure 45 compares median values of $z_{cld}$ - $z_{COD0.5}$, $z_{top}$ - $z_{cld}$ and ($z_{top}$ - $z_{cld}$)/($z_{top}$  - $z_{app\ base}$)

as a function of $\varepsilon_{cld}$,  for high-level clouds

For this analysis we have selected cases for which $z_{cld}$ lies between top and base of the closest  GEOPROF cloud layer.

This, leavesing about 82% / 73% / 57% and about 55% / 59% / 58% of the statistics of high-level and lower level clouds over thein tropics / midlatitudes / polar regions, respectively. In general, for low-level clouds, the AIRS cloud height lies about 250 m ó 500 m below the height at which the cloud reaches an optical depth of about 0.5, independently of $\varepsilon_{cld}$, while $z_{cld}$ lies about 1 km below the cloud top. For high-level clouds the $z_{cld}$ varies from 1 km above for $\varepsilon_{cld} = 0.1$ to 1 km below $z_{COD0.5}$ the height corresponding to COD of 0.5 for $\varepsilon_{cld} = 1$, assuming that $z_{COD0.5}$ COD is accurately determined estimated for all $\varepsilon_{cld}$ (section 2.4). In that case, This means that for thin cirrus $z_{cld}$ from AIRSof thin cirrus should be approximatedcorresponds to by a height of at whichwith COD reaches a value $< 0.5$, while forand $z_{cld}$ of opaque high clouds to by a height at whichwith of COD reaches a value $> 0.5$. On the other hand, $z_{cld}$ lies about 1.5 km to 2.5 km below $z_{top}$ $z_{top}$ the cloud top, the difference to cloud top increasing with $\varepsilon_{cld}$ (except for $\varepsilon_{cld}$ close to 1). Since $z_{top}$ $z_{top} - z_{app\,base}$ the apparent vertical extent also increases with $\varepsilon_{cld}$, (not shown), the $(z_{top} - z_{cld})/(z_{top}$ $z_{top} - z_{app\,base})$ difference between $z_{top}$ and $z_{cld}$ scaled by apparent vertical extent does not depend on $\varepsilon_{cld}$, and it is about 0.5 for high-level and for low level clouds. Considering the normalized frequency distributions of $z_{top}$ ó $z_{COD0.5}$ and $z_{top} - z_{cld}$, as well as these differences scaled by apparent cloud vertical extent, presented in Figure 6, Wwe deduce that it probably needs less geometrical thicknessvertical extent for opaque clouds than for semi-transparent clouds cirrus to reach a COD of 0.5, while the $\chi^2$ method determines a height within the cloud, which corresponds well to the middle mean between cloud top and apparent cloud base or the height at which the cloud reaches opacity, independent of $\varepsilon_{cld}$. This is important to take into account for the determination of radiative fluxes and heating rates of upper tropospheric clouds, when using the CIRS cloud heights retrieved from IR sounder measurements. We want to stress that also for low-level clouds $(z_{top} - z_{cld})/(z_{top}$ $z_{top} - z_{app\,base})$ is about 0.5 (0.4 to 0.6), while The broader distributions for high-level clouds compared to low-level clouds in Figures 4 and 6 may be explained by the fact that high-level clouds often have diffuse cloud tops (e. g. Liao et al., 1995), especially in the tropics ($z_{top} - z_{cld}$ is slightly larger for the same $\varepsilon_{cld}$). $z_{cld}$ of low-level clouds lies only about 0.1400 tom ó 1000.4 km below $z_{COD0.5}$, while $z_{cld}$ liesand about 500 0.5 km below $z_{top}$ $z_{top}$ 
[revised manuscript text omitted]
-The slightly higher smaller value in CALIPSO CAMR (2014% instead of 1420%) can be explainedis due by the factto that the different distinction between high-level and mid-level clouds: ofCALIPSO is according touses cloud top height, whereas AIRS and IASI provide-use a cloud height which is about 1.5 km lower than the top (see-section 3.23). When combining VIS and IR information, thin cirrus above low-level clouds tend to be misidentified as mid-level clouds (ISCCP) or as low-level clouds (MODIS), leading to a not negligible underestimation of CAHR (30% instead of 40%). During At nighttime, for whichwhen only one the IR channel is available, ISCCP underestimates the height of all semi-transparent high-level clouds, so that CAHR drops to 15%. When IR spectral information is available, as for IR sounders and MODIS, results are similar to those during daytime.

Differences between ocean and land, also presented in Figure 7, correspond to about 0.15 in

CA, with about 20% more low-level clouds over ocean and about 10% more high-level and mid- level clouds over land. The CIRS retrievals provide similar values during day and night. It is interesting to note that during daytime the difference in CA shows a larger spread between the datasets, while at night the spread is larger for CALR. At night, low-level clouds are more difficult to detect, especially over land.

Table 2 summarizes averages of these cloud amounts over the whole globe, over ocean and over land, also contrasting NH and Southern hemisphere (SH) midlatitudes (30°-60°) and tropics (15°N-15°S). The largest fraction of high-level clouds is situated in the tropics, while the largest fraction of single layer low-level clouds in the SH midlatitudes. Only about 10% of all clouds in the tropics are single layer midlevel clouds, compared to about 22% in the midlatitudes. As already discussed in sections 2.5 and

3.1, the uncertainty due to ancillary data in CA, as well as in CALR,  is largest over land (about 5% and 10%, respectively), because low-level clouds are underestimated  with AIRS-NASA during night and  with ERA-Interim in the afternoon. Uncertainties  are much smaller for high-level clouds.

Considering further three distinct high-level cloud classes, opaque, thick cirrus and thin cirrus  (see section 2.5), uncertainties for opaque clouds increase to 10% at midlatitudes might be due to interpolation of ancillary data and atmospheric profiles and $T_{surf}$  having 
[revised manuscript text omitted]

[Figure]

Figure 68. Normalized frequency distributions of $p_{cld}$ , separately over land and over ocean in six latitude bands of 30° from SH polar (left) to NH polar latitudes (right), in boreal winter (December, January, February; blue) and in boreal summer (June, July, August; red). Compared are results from AIRS-CIRS using two sets of ancillary data from AIRS-CIRS, using ancillary data from(-AIRS-NASA, (dashed line) and from (ERA-Interim, (dotted line), as well as from IASI-CIRS (full line), separately over land (top) and over ocean (bottom) in six latitude bands of 30° from Southern hemisphere polar (left) to Northern hemisphere polar latitudes (right), in boreal winter (December, January, February; blue) and in boreal summer (June, July, August; red). Statistics from 2008.

[Figure]

Figure 79. Top: Global averages of total cloud amount (CA) and fraction of high-level, mid-level and low-level cloud amount, relative to total cloud amount, (CAHR + CAMR + CALR = 1). Comparisons of IR sounder cloud data (AIRS, IASI) with L3 data from the GEWEX Cloud Assessment data base, separately for observations mostly during day ( 1:30PM; (3:00PM for ISCCP and 9:30AM for IASI, left), and mostly during night ( 1:30AM; (3:00AM for ISCCP and 9:30PM for IASI). Compared to the original ISCCP data, the day-night adjustment on CA has not been included to better illustrate the differences between VIS-IR and IR-only results. Bottom: Averages of ocean-land differences for the same parameters and data sets.

[Figure]

[Figure]

| CA | —— AIRS-CIRS | ---- AIRS-CIRS(ERA) | ········ IASI-CIRS(ERA) | –·–· AIRS-LMD |
|  |  |  |  |  |
| CAL | —— AIRS-CIRS | ---- AIRS-CIRS(ERA) | ········ IASI-CIRS(ERA) | –·–· AIRS-LMD |

Figure 8. Annual mean zonal distributions of CA, CAH and CAL (left) and CAE, CAEH and

CAEL (right)

. Results  are compared between AIRS-CIRS, using ancillary data from AIRS-NASA  and from ERA-

Interim  dashed line), IASI-CIRS  and AIRS-LMD .

For boreal winter and boreal summer, AIRS-CIRS (using AIRS-NASA ancillary data) is shown separately for each year between 2003 to 2015, illustrating inter-annual variability .

Figure 911. Top: Geographical maps of annual CAH (left) and CAL (right), from of AIRS-CIRS (2003-

2015, top),(2003-2015, top)  compared to ISCCP (20031984-2007, middle) and CALIPSO-GOCCP

(2007-2008, bottom),- the latter two from the GEWEX Cloud Assessment data base, as well as seasonal anomalies of DJF (middle) and of JJA (right).

[Figure]

○ *AIRS-CIRS*   ○ *AIRS-CIRS(ERA)*   ○ *IASI-CIRS(ERA)*   - - - - *AIRS-CIRS no CO2 corr.*

[Figure]

○ *AIRS-CIRS* ● *AIRS-CIRS(ERA)* ■ *IASI-CIRS(ERA)* —— *AIRS-CIRS no CO2 corr.*

Figure 1$\underline{0}$$\underline{3}$. Time anomalies of deseasonalized CA, CAEH and CAEL over the globe. In the case of CA, additional values are shown without calibration of spectral atmospheric transmissivities for changes in atmospheric $CO_2$ concentration.

[Figure]

Figure 12.5 Seasonal cycle / annual average of (1) CAH differences between NH hemisphere (0°-60N)

and SH hemisphere (0°-60S)(2) ITCZ peak latitude, (3)

maximum CAH within ITCZ and (4) width of ITCZ.

[Figure]

Figure 16. Differences between December 2015 and 2010, corresponding to El Niño and La Niña, respectively, in $T_{surf}$ (1. Panel, left), total atmospheric water vapour (1. Panel, right) and vertical wind at 500 hPa (2. Panel, left) from ERA-Interim, in CAH (2. Panel, right), fraction of Cb (3. Panel, left), cloud temperature of high-level clouds (3. Panel, right) and fraction of thin cirrus (4. Panel, left) from AIRS-CIRS, and OLR (4. Panel, right) from AIRS-NASA.

[Figure]

Figure 1237. Left: Geographical maps of linear regression sSlopes of between change monthly mean anomalies in amount of Cb ($\varepsilon_{cld} > 0.95$, top row), cirrus Ci ($0.95 > \varepsilon_{cld} > 0.4$, middle row) and thin cirrus Ci ($0.4 > \varepsilon_{cld} > 0.1$, bottom row) amount from AIRS-CIRS in % per °C of tropicaland global mean surface temperature anomalies warming (20°N ó 20°S)from ERA-Interim; left: $p_{cld} < 440$ hPa, middle: relative cloud amount,; right: $p_{cld} < 330$ hPa and relative cloud amount. Results using slope uncertainty for Cb (top), cirrus (middle) and thin cirrus (bottom) amount change per °C of tropical warming. Results using upper tropospheric ($p_{cld} < 330$ hPa) cloud type anomalies from AIRS-CIRS and surface temperature anomalies from ERA-Interim of 156 months during the period 2003-2015.